



**Influence of basement heterogeneity on the architecture of low subsidence rate Paleozoic**
**intracratonic basins (Ahnet and Mouydir basins, Central Sahara)**
Paul Perron[1], Michel Guiraud[1], Emmanuelle Vennin[1], Isabelle Moretti[2], Éric Portier[3], Laetitia
Le Pourhiet[4], Moussa Konaté[5]
[1]Université de Bourgogne Franche-Comté, Centre des Sciences de la Terre, UMR CNRS
6282 Biogéosciences, 6 Bd Gabriel, 21000 Dijon, France.
[2]ENGIE, Département Exploration & Production, 1, place Samuel de Champlain, Faubourg
de l'Arche, 92930 Paris La Défense, France.
[3]NEPTUNE Energy International S.A., 9-11 Allée de l'Arche – Tour EGEE – 92400
Courbevoie, France.
[4]Sorbonne Université, CNRS-INSU, Institut des Sciences de la Terre Paris, ISTeP UMR
7193, F-75005 Paris, France.
[5]Département de Géologie, Université Abdou Moumouni de Niamey, BP :10662, Niamey,
Niger.
Corresponding author: paul.perron@u-bourgogne.fr, paul.perron@hotmail.fr
**Abstract**
The Paleozoic intracratonic North African Platform is characterized by an association of
arches (ridges, domes, swells or paleo-highs) and low subsidence rate syncline basins of
different wavelengths (75–620 km). The structural framework of the platform results from the
accretion of Archean and Proterozoic terranes during the Pan-African orogeny (750–580 Ma).
The Ahnet and Mouydir basins are successively delimited from east to west by the Amguid El
Biod, Arak-Foum Belrem, and Azzel Matti arches, bounded by inherited Precambrian sub-
vertical fault systems which were repeatedly reactivated or inverted during the Paleozoic.



Major unconformities are related to several tectonic events such as the Cambrian–Ordovician
extension, Ordovician–Silurian glacial rebound, Silurian–Devonian "Caledonian"
extension/compression, late Devonian extension/compression, and "Hercynian" compression.
The deposits associated with these arches and syncline basins exhibit thickness variations and
facies changes ranging from continental to marine environments. The arches are characterized
by thin amalgamated deposits with condensed and erosional surfaces, whereas the syncline
basins exhibit thicker and well-preserved successions. In addition, the vertical facies
succession evolves from thin Silurian to Givetian deposits into thick Upper Devonian
sediments. Synsedimentary deformations are evidenced by wedges, truncations, and divergent
onlaps. Locally, deformation is characterized by near-vertical planar normal faults responsible
for horst and graben structuring associated with folding during the Cambrian–Ordovician–
Silurian period. These structures may have been inverted or activated during the Devonian
compression and the Carboniferous. The sedimentary infilling pattern and the nature of
deformation result from the slow Paleozoic reactivation of Precambrian terranes bounded by
vertical lithospheric fault zones. Alternating periods of tectonic quiescence and low-rate
subsidence acceleration associated with extension and local inversion tectonics correspond to
a succession of Paleozoic geodynamic events (i.e. far-field orogenic belt, glaciation).
Keywords: intracratonic basin, Paleozoic, arches, low-rate subsidence, tectonic heritage,
terranes, Central Sahara
**1    Introduction**
Paleozoic deposits fill numerous intracratonic basins, which may also be referred to as
"cratonic basins", "interior cratonic basins", or "intracontinental sags". Intracratonic basins
are widespread around the world (see Fig. 6 from Heine et al., 2008) and exploration for non-
conventional petroleum has revived interest in them. They are located in "stable" lithospheric



areas and share several common features (see Allen and Armitage, 2011 and references
therein) such as their geometries (i.e. large circular, elliptical, saucer-shaped to oval), their
stratigraphy (i.e. filled with continental to shallow-water sediments), their low rate of
sedimentation (an average of 7 m/Myr), their long-term subsidence (sometimes more than 540
Myr), and their structural framework (reactivation of structures and emergence of arches also
referred to in the literature as "ridges", "paleo-highs", "domes", and "swells"). Multiple
hypotheses and models have been proposed to explain how these slowly subsiding, long-lived
intracratonic basins formed and evolved (see Allen and Armitage, 2011 and references therein
or Hartley and Allen, 1994). However, their tectonic and sedimentary architectures are often
poorly constrained.
The main specificities of intracratonic basins are found on the Paleozoic North Saharan
Platform. The sedimentary infilling during c. 250 Myr is relatively thin (i.e. around a few
hundred to a few thousand meters), of great lateral extent (i.e. 16 million km$^2$), and is
separated by major regional unconformities (Beuf et al., 1971; Carr, 2002; Eschard et al.,
2005, 2010; Fabre, 1988, 2005; Fekirine and Abdallah, 1998; Guiraud et al., 2005; Legrand,
2003). Depositional environments were mainly continental to shallow-marine and
homogeneous. Very slow and subtle lateral variations occurred over time (Beuf et al., 1971;
Carr, 2002; Fabre, 1988; Guiraud et al., 2005; Legrand, 2003). The Paleozoic North Saharan
Platform is arranged (Fig. 1) into an association of long-lived broad synclines (i.e. basins) and
anticlines (i.e. arches swells, domes, highs, or ridges) of different wavelengths (λ: 75–620
km). Burov and Cloetingh, (2009) report deformation wavelengths of the order of 200–600
km when the whole lithosphere is involved and of 50–100 km when the crust is decoupled
from the lithospheric mantle. This insight suggests that the inherited basement fabric
influences basin architecture at a large scale. Intracratonic basins are affected by basement
involved faults which are often reactivated in response to tectonic pulses (Beuf et al., 1971;





Boote et al., 1998; Eschard et al., 2010; Fabre, 1988; Frizon de Lamotte et al., 2013; Galeazzi
et al., 2010; Guiraud et al., 2005; Wendt et al., 2006).
In this study of the Ahnet and Mouydir basins, a multidisciplinary workflow involving
various tools (e.g. seismic profiles, satellite images) and techniques (e.g. photo-geology,
seismic interpretation, well correlation, geophysics, geochronology) has enabled us to (1)
make a tectono-sedimentary analysis, (2) determine the spatial arrangement of depositional
environments calibrated by biostratigraphic zonation, (3) characterize basin geometry, and (4)
ascertain the inherited architecture of the basement and its tectonic evolution. We propose a
conceptual coupled model explaining the architecture of the intracratonic basins of the North
Saharan Platform. This model highlights the role of basement heritage heterogeneities in an
accreted mobile belt and their influence on the structure and evolution of intracratonic basins.
It is a first step towards a better understanding of the factors and mechanisms that drive
intracratonic basins.
**2   Geological setting: The Paleozoic North Saharan Platform and the Ahnet and**
**Mouydir basins**
The Ahnet and Mouydir basins (Figs 1 and 3) are located in south-western Algeria, north-
west of the Hoggar massif (Ahaggar). They are N–S oval depressions filled by Paleozoic
deposits. The basins are bounded to the south by the Hoggar massif (Tuareg Shield), to the
west by the Azzel Matti arch, to the east by the Amguid El Biod arch and they are separated
by the Arak-Foum Belrem arch.
Figure 2 synthetizes the lithostratigraphy, the large-scale sequence stratigraphic framework
delimited by five main regional unconformities (A to E), and the tectonic events proposed in
the literature (cf. references under Fig. 2) affecting the Paleozoic North Saharan Platform.



During the Paleozoic, the Ahnet and Mouydir basins were part of a set of the super-continent
Gondwana (Fig. 1). This super-continent resulted from the collision of the West African
Craton (WAC) and the East Saharan Craton (ESC), sandwiching the Tuareg Shield (TS)
mobile belt during the Pan-African orogeny (Craig et al., 2006; Guiraud et al., 2005;
Trompette, 2000). This orogenic cycle followed by the chain's collapse (c. 1000–525 Ma)
was also marked by phases of oceanization and continentalization (c. 900–600 Ma) giving rise
to the heterogeneous terranes in the accreted mobile belt (e.g. Trompette, 2000). Then, there is
evidence of a complex and polyphased history throughout the Paleozoic (Fig. 2), with
alternating periods of quiescence and tectonic activity, individualizing and rejuvenating
ancient NS, NE–SW, or NW–SE structures in arch and basin configurations (Badalini et al.,
2002; Bennacef et al., 1971; Beuf et al., 1968a, 1971; Boote et al., 1998; Boudjema, 1987;
Chavand and Claracq, 1960; Coward and Ries, 2003; Craig et al., 2006; Eschard et al., 2005,
2010, Fabre, 1988, 2005; Frizon de Lamotte et al., 2013; Guiraud et al., 2005; Logan and
Duddy, 1998; Lüning, 2005; Wendt et al., 2006). The Paleozoic successions of the North
Saharan Platform are predominantly composed of siliciclastic detrital sediments (Beuf et al.,
1971; Eschard et al., 2005). They form the largest area of detrital sediments ever found on
continental crust (Burke et al., 2003), dipping gently NNW (Beuf et al., 1969, 1971; Fabre,
1988, 2005; Fröhlich et al., 2010; Gariel et al., 1968; Le Heron et al., 2009). Carbonate
deposits are observed from the Mid–Late Devonian to the Carboniferous (Wendt, 1995, 1988,
1985; Wendt et al., 2009b, 2006, 1997, 1993; Wendt and Kaufmann, 1998). From south to
north, the facies progressively evolve from continental fluviatile to shallow marine (i.e. upper
to lower shoreface) and then to offshore facies (Beuf et al., 1971; Carr, 2002; Eschard et al.,
2005, 2010, Fabre, 1988, 2005; Fekirine and Abdallah, 1998; Guiraud et al., 2005; Legrand,
1967a).
**3   Data and methods**



A multidisciplinary approach has been used in this study integrating new data in particular
from the Ahnet and Mouydir basins (see supplementary data 1):
- Geographic Information System analysis (GIS);
- The basins and the main geological structures were identified from Landsat satellite images;
- Seismic section interpretation;
- Sedimentological and well-log analysis;
- Biostratigraphy and sequence stratigraphy;
- Geochronology and geophysical data.
The Paleozoic series of the Ahnet and Mouydir basins are well-exposed over an area of
approximately 170,000 km$^2$ and are well observed in satellite images (Google Earth and
Landsat from USGS). Furthermore, a significant geological database (i.e. wells, seismic
records, field trips, geological reports) has been compiled in the course of petroleum
exploration since the 1950s. The sedimentological dataset is based on the integration and
analysis of cores, outcrops, well-logs, and of lithological and biostratigraphic data. Facies
described from cores and outcrops of these studies were grouped into facies associations
corresponding to the main depositional environments observed on the Saharan Platform (table
1). Characteristic gamma-ray patterns (electrofacies) are proposed to illustrate the different
facies associations. The gamma-ray (GR) peaks are commonly interpreted as the maximum
flooding surfaces (MFS) (e.g. Catuneanu et al., 2009; Galloway, 1989; Milton et al., 1990;
Serra, 2009). Time calibration of well-logs and outcrops is based on palynomorphs
(essentially Chitinozoans and spores), conodonts, goniatites, and brachiopods (Wendt et al.,

143  2006).



Synsedimentary extensional and compressional markers are characterized in this structural
framework based on the analyses of satellite images (Figs 4 and 5), seismic profiles (Fig. 6),
21 wells (W1 to W21), and 12 outcrop cross-sections (O1 to O12). Wells and outcrop sections
are arranged into three E–W sections (Figs 9, 10 and supplementary data 2) and one N–S
section (supplementary data 3). Satellite images (Figs 4 and 5) and seismic profiles (Fig. 6)
are located at key areas (i.e. near arches) illustrating the relevant structures (Fig. 3). The
calibration of the key stratigraphic horizon on seismic profiles (Figs 9 and 10) was settled by
sonic well-log data using PETREL and OPENDTECT software. Nine key horizons easily
extendable at the regional scale are identified and correspond to major unconformities: top
Infra-Cambrian, top Ordovician, top Silurian, top Pragian, top Givetian, top mid-Frasnian, top
Famennian, top Quaternary and top Hercynian unconformities (Figs 9 and 10). The
stratigraphic layers are identified by the integration of satellite images (Google Earth and
Landsat USGS: https://earthexplorer.usgs.gov/) and the 1:20,000 geological map of Algeria
(Bennacef et al., 1974; Bensalah et al., 1971).
Subsidence analysis characterizes the vertical displacements of a given sedimentary
depositional surface by tracking its subsidence and uplift history (Van Hinte, 1978). The
resulting curve details the total subsidence history for a given stratigraphic column (Allen and
Allen, 2005; Van Hinte, 1978). Backstripping is also used to restore the initial thicknesses of
a sedimentary column (Allen and Allen, 2005; Angevine et al., 1990). Lithologies and
paleobathymetries have been defined using facies analysis or literature data. Porosity and the
compaction proxy are based on experimental data from Sclater and Christie, (1980). In this
study, subsidence analyses were performed on sections using OSXBackstrip software
performing 1D Airy backstripping (after Allen and Allen, 2005; Watts, 2001; available at:
http://www.ux.uis.no/nestor/work/programs.html).
**4    Structural framework and tectono-sedimentary structure analyses**





The structural architecture of the North Saharan Platform (Fig. 1) is characterized by an
association of syncline basins and anticlines (i.e. arches, domes, etc.). The basins (or sub-
basins) are mostly circular to oval. They are bounded by arches which correspond to the
mainly N–S Azzel-Matti, Arak-Foum Belrem, Amguid El Biod, and Tihemboka arches, the
NE–SW Bou Bernous, Ahara, and Gargaf arches, and the NW–SE Saoura and Azzene arches
(Fig. 1). The basins are structured by major faults frequently associated with broad
asymmetrical folds displayed by three main trends (Fig. 1): (1) near-N–S, varying from N0°
to N10° or N160°, (2) from N40° to 60°, and (3) N100° to N140° directions (Figs 1, 3A, and
4). These fault zones are about 100 km (e.g. faults F1 and F2, Fig. 4) to tens of kilometers
lengths  (e.g. faults  F3 to F8, Fig. 4).

### 4.1    Synsedimentary extensional markers

Extensional markers are characterized by the settlement of steeply west- or eastward-dipping
basement normal faults associated with colinear syndepositional folds of several kilometers in
length (e.g. Fig. 5A-A', 5B-B', 5C-C', 5E-E' and 6A), represented by footwall anticline and
hanging wall syncline-shaped forced folds. They are located in the vicinity of different arches
(Fig. 3) such as the Tihemboka arch (Figs 4BB', 5A-A' and 5B-B'), Arak-Foum Belrem arch
(Figs 4A-A', 5C-C' to 5F-F' and 6A, 6C), Azzel Matti arch (Fig. 6B), and Bahar El Hamar
area intra-basin arch (Fig. 6D). These tectonic structures can be featured by basement blind
faults (e.g. fault F5 in Fig. 5C-C', fault F1 in Figs 5D-D', 5F-F', and 6A). The deformation
pattern is mainly characterized by brittle faulting in Cambrian–Ordovician series down to the
basement and fault-damping in Silurian series (e.g. fault F2 in Fig. 5AA', faults F1 to F6 in
Fig. 6B). The other terms of the series (i.e. Silurian to Carboniferous) are usually affected by
folding except (see F1 faults in Figs 5F-F', 6B, 6D and 6C) where the brittle deformation can
be propagated to the Upper Devonian (due to reactivation and/or inversion as suggested in the
next paragraph).



In association with the extensional markers, thickness variations and tilted divergent onlaps of
the sedimentary series (i.e. wedge-shaped units, progressive unconformities) in the hanging
wall syncline of the fault escarpments are observed (Figs 5 and 6). These are attested using
photogeological analysis of satellite images (Fig. 5) and are marked by a gentler dip angle of
the stratification planes away from the fault plane (i.e. fault core zone). The markers of
syndepositional deformation structures are visible in the hanging-wall synclines of
Precambrian to Upper Devonian series (Figs 5 and 6).
The footwall anticline and hanging-wall syncline-shaped forced folds recognized in this study
are very similar to those described in the literature by Schlische (1995), Withjack et al. (1990,
2002), Withjack and Callaway (2000), Khalil and McClay (2002), and Grasemann et al.
(2005). The wedge-shaped units (DO0 to DO3; Figs 4, 5 and 6) associated with the hanging-
wall synclines are interpreted as synsedimentary normal fault-related folding. The whole
tectonic framework forms broad extensional horsts and graben related to synsedimentary
forced folds controlling basin shape and sedimentation (Figs 4, 5 and 6).
Following Khalil and McClay (2002), Lewis et al. (2015), Shaw et al. (2005), and Withjack et
al. (1990), we use the ages of the growth strata (i.e. wedge-shaped units) to determine the
timing of the deformation. The main four wedge-shaped units identified (DO0 to DO3) are
indicative of the activation and/or reactivation of the normal faults (extensional settings)
during Neoproterozoic, Cambrian–Ordovician, Early to Mid Silurian and Mid to Late
Devonian times.
In planar view, straight (F1 in Fig. 4A-A') and sinuous faults (F2, F3, F3', F4, F4', and F5 in
Fig. 4AA') can be identified. The sinuous faults are arranged "en echelon" into several
segments with relay ramps. These faults are 10 to several tens of kilometers long with vertical
throws of hundreds of meters that fade rapidly toward the fault tips. The sinuous geometry of

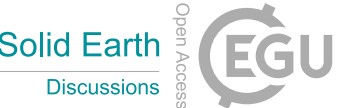

normal undulated faults as well as the rapid lateral variation in fault throw are controlled by
the propagation and the linkage of growing parent and tip synsedimentary normal faults
(Marchal et al., 2003, 1998; Fig. 4A-A').
According to Holbrook and Schumm (1999), river patterns are extremely sensitive to tectonic
structure activity. Here we find that the synsedimentary activity of the extensional structures
is also evidenced by the influence of the fault scarp on the distribution and orientation of
sinuous channelized sandstone body systems (dotted red lines in Fig. 4B-B').

### 4.2 Synsedimentary compressional markers (inversion tectonics)

After the development of the extensional tectonism described previously, evidence of
synsedimentary compressional markers can be identified. These markers are located and
preferentially observable near the Arak-Foum Belrem arch (Fig. 5F-F', F2 in Fig. 6C), the
Azzel Matti arch (2 in Figs 6B), and the Bahar El Hamar area intra-basin arch (2 in Fig. 6D).
The tectonic structures take the form of inverse faulting reactivating former basement faults
(F1' in Fig. 5F-F', F1 in Fig. 6C, F1' in Fig. 6D, F1 in Fig. 6B). The synsedimentary inverse
faulting is demonstrated by the characterization of asymmetric anticlines especially
observable in satellite images and restricted to the fault footwalls (Figs 4A-A' along F1-F2).
Landsat image analysis combined with the line drawing of certain seismic lines reveals
several thickness variations reflecting divergent onlaps (i.e. wedge-shaped units) which are
restricted to the hanging-wall asymmetric anticlines (2 in Figs 5F-F', 6B, 6C and 6D). The
compressional synsedimentary markers clearly post-date extensional divergent onlaps at
hanging-wall syncline-shaped forced folds (1 in Figs 6B, 6C and 6D). This architecture is
very similar to classical positive inversion structures of former inherited normal faults
(Bellahsen and Daniel, 2005; Bonini et al., 2012; Buchanan and McClay, 1991; Ustaszewski
et al., 2005). Tectonic transport from the paleo-graben hanging-wall toward the paleo-horst



footwall (F1, F2-F2', F4-F4' in Fig. 6B; F1-F1' in Fig. 6B) is evidenced. Further positive
tectonic inversion architecture is identified by tectonic transport from the paleo-horst footwall
to the paleo-graben hanging wall (F1-F1' in Fig. 5F-F'; F1, F5, and F6 in Fig. 6C). This
second type of tectonic inversion is very similar to the transported fault models defined by
Butler (1989) and Madritsch et al. (2008). The local positive inversions of inherited normal
faults occurred during Silurian–Devonian (F4' Fig. 6B) and Mid to Late Devonian times (Figs
6B, 6C and 6D). A late significant compression event between the end of the Carboniferous
and the Early Mesozoic was responsible for the exhumation and erosion of the tilted
Paleozoic series. This series is related to the Hercynian angular unconformity surface (Fig.
6B).
**5   Stratigraphy and sedimentology**
The whole sedimentary series described in the literature is composed of fluviatile Cambrian
(Beuf et al., 1968a, 1968b, 1971; Eschard et al., 2005, 2010), glacial Ordovician (Beuf et al.,
1968a, 1968b, 1971; Eschard et al., 2005, 2010), argillaceous deep marine Silurian (Eschard
et al., 2005, 2010; Legrand, 1986, 1996; Lüning et al., 2000) and offshore to embayment
Carboniferous (Wendt et al., 2009) deposits. In this complete sedimentary succession, we
have focused on the Devonian deposits as they are very sensitive to and representative of
basin dynamics. The architecture of the Devonian deposits allows us to approximate the main
forcing factors controlling the sedimentary infilling of the basin and its synsedimentary
deformation. Nine facies associations organized into four depositional environments are
defined to reconstruct the architecture and the lateral and vertical sedimentary evolution of the
basins (Figs 9, 10, supplementary data 2 and 3).
**5.1   Facies association, depositional environments, and erosional unconformities**





Based on the compilation and synthesis of internal studies (Eschard et al., 1999), published
papers on the Algerian platform (Beuf et al., 1971; Eschard et al., 2010, 2005; Henniche,
2002) and on the Ahnet and Mouydir basins (Biju-Duval et al., 1968; Wendt et al., 2006) plus
the present study, eleven main facies associations (AF1 to AF5) and four depositional
environments are proposed for the Devonian succession (table 1). They are associated with
their gamma-ray responses (Figs 7 and 8). They are organized into two continental/fluvial
(AF1 to AF2), four transitional/coastal plain (AF3a to AF3d), three shoreface (AF4a to
AF4c), and two offshore (AF5a to AF5b) sedimentary environments.
**5.1.1   Continental fluvial environments**
This depositional environment features the AF1 (fluvial) and the AF2 (flood plain) facies
association (Table 1). Facies association AF1 is mainly characterized by a thinning-up
sequence with a basal erosional surface and trough cross-bedded intraformational
conglomerates with mud clast lag deposits, quartz pebbles, and imbricated grains (Table 1). It
passes into medium to coarse trough cross-bedded sandstones, planar cross-bedded siltstones,
and laminated shales. These deposits are associated with rare bioturbations (expect at the
surface of the sets), ironstones, phosphorites, corroded quartz grains, and phosphatized
pebbles. Laterally, facies association AF2 is characterized by horizontally laminated and very
poorly sorted silt to argillaceous fine sandstones. They contain frequent root traces, plant
debris, well-developed paleosols, bioturbations, nodules, and ferruginous horizons. Current
ripples and climbing ripples are associated in prograding thin sandy layers.
In AF1, the basal erosional reworking and high energy processes are characteristic of channel-
filling of fluvial systems (Allen, 1983; Owen, 1995). Eschard et al. (1999) identify three
fluvial systems (see A, B, and C in Fig. 8) in the Tassili-N-Ajjers outcrops: braided dominant
(AF1a), meandering dominant (AF1b), and straight dominant (AF1c). They differentiate them



by their different sinuosity, directions of accretion (lateral or frontal), the presence of mud
drapes, bioturbations, and giant epsilon cross-bedding. Gamma-ray signatures of these facies
associations (A, B, and C in Fig. 8) are cylindrical with an average value of 20 gAPI. The
gamma ray shapes are largely representative of fluvial environments (Rider, 1996; Serra,
2009; Wagoner et al., 1990). The bottom is sharp with high value peaks and the tops are
frequently fining-up, which may be associated with high values caused by argillaceous flood
plain deposits and roots (Eschard et al., 1999). AF2 is interpreted as humid floodplain
deposits (Allen, 1983; Owen, 1995) with crevasse splays or preserved levees of fluvial
channels (Eschard et al., 1999). Gamma-ray curves of AF2 (D, Fig. 8) show a rapid
succession of low to very high peak values, ranging from 50 to 200 gAPI. AF1 and AF2 are
typical of the Pragian "Oued Samene" Formation (Wendt et al., 2006). In the Illizi basin,
these facies are mainly recorded in the Cambrian Ajjers Formation and the Lochkovian to
Pragian "Middle Barre" and "Upper Barre" Formations (Beuf et al., 1971; Eschard et al.,

302   2005).

### 5.1.2   Transitional coastal plain environments

This depositional environment comprises facies associations AF3a (delta/estuarine), AF3b
(fluvial/tidal distributary channels), AF3c (tidal sand flat), AF3d (lagoon/mudflat) (table 1).
AF3a is mainly dominated by sigmoidal cross-bedded heterolithic rocks with mud drapes. It is
also characterized by fine to coarse, poorly sorted sandstones and siltstones often structured
by combined flow ripples, flaser bedding, wavy bedding, and some rare planar bedding. Mud
clasts, root traces, desiccation cracks, water escape features, and shale pebbles are common.
The presence of epsilon bedding is attested, which is formed by lateral accretion of a river
point bar (Allen, 1983). The bed surface sets are intensively bioturbated (*Skolithos* and
*Planolites*) indicating a shallow marine subtidal setting (Pemberton and Frey, 1982). Faunas
such as brachiopods, trilobites, tentaculites, and graptolites are present. AF3b exhibits a





fining-up sequence featured by a sharp erosional surface, trough cross-bedded, very coarse-
grained, poorly sorted sandstone at the base and sigmoidal cross-bedding at the top (Figs 7
and 8). AF3c is formed by fine-grained to very coarse-grained sigmoidal cross-bedded
heterolithic sandstones with multidirectional tidal bundles. They are also structured by
lenticular, flaser bedding and occasional current and oscillation ripples with mud cracks. They
reveal intense bioturbation composed of *Skolithos* (Sk), *Thalassinoides* (Th), and *Planolites*
(Pl) ichnofacies indicating a shallow marine subtidal setting (Frey et al., 1990; Pemberton and
Frey, 1982). AF4d is characterized by horizontally laminated mudstones associated with
varicolored shales and fine-grained sandstones. They exhibit mud cracks, occasional wave
ripples, and rare multidirectional current ripples. These sedimentary structures are poorly
preserved because of intense bioturbation composed of *Skolithos* (Sk), *Thalassinoides* (Th),
and *Planolites* (Pl). Fauna includes ammonoids (rare), goniatites, calymenids, pelecypod
molds, and brachiopod coquinas.
In AF3a, both tidal and fluvial systems in the same facies association can be interpreted as an
estuarine system (Dalrymple et al., 1992; Dalrymple and Choi, 2007). The gamma-ray
signature is characterized by a convex bell shape with rapidly alternating low to high values
(30 to 60 gAPI) due to the mud draping of the sets (see E Fig. 8). These forms of gamma ray
are typical of fluvial-tidal influenced environments with upward-fining parasequences (Rider,
1996; Serra, 2009; Wagoner et al., 1990). AF3a is identified at the top of the Pragian "Oued
Samene" Formation and in Famennian "Khenig" Formation (Wendt et al., 2006) in the Ahnet
and Mouydir basins. In the Illizi basin, AF3a is mostly recorded at the top Cambrian of the
Ajjers Formation, in the Lochkovian "Middle bar", and at the top Pragian of the "Upper bar"
Formation (Beuf et al., 1971; Eschard et al., 2005). The AF3b association can be
characterized by a mixed fluvial and tidal dynamic based on criteria such as erosional basal
contacts, fining-upward trends or heterolythic facies (Dalrymple et al., 1992; Dalrymple and



Choi, 2007). They are associated with abundant mud clasts, mud drapes, and bioturbation
indicating tidal influences (Dalrymple et al., 2012, 1992; Dalrymple and Choi, 2007). The
major difference with the estuarine facies association (AF3a) is the slight lateral extent of the
channels which are only visible in outcrops (Eschard et al., 1999). The gamma-ray pattern is
very similar to the estuarine electrofacies (see F Fig. 8). AF3c is interpreted as a tidal sandflat
laterally present near a delta (Lessa and Masselink, 1995) and associated with an estuarine
environment (Leuven et al., 2016). The gamma-ray signature (see G Fig. 8) is distinguishable
by its concave funnel shape with alternating low and high peaks (25 to 60 gAPI) due to the
heterogeneity of the deposits and rapid variations in the sand/shale ratio. These facies are
observed in the "Tigillites Talus" Formation of the Illizi basin (Eschard et al., 2005). In AF4d,
both ichnofacies and facies are indicative of tidal mudflat/lagoonal depositional environments
(Dalrymple et al., 1992; Dalrymple and Choi, 2007; Frey et al., 1990). The gamma-ray
signature has a distinctively high value (80 to 130 gAPI) and an erratic shape (see H Fig. 8).
AF4d is observed in the "Atafaitafa" Formation and in the Emsian prograding shoreface
sequence of the Illizi basin (Eschard et al., 2005). It is also recorded in the Lochkovian "Oued
Samene" Formation and the Famennian "Khenig" Sandstones (Wendt et al., 2006).
**5.1.3   Shoreface environments**
This depositional environment is composed of AF4a (subtidal), AF4b (upper shoreface), and
AF4c (lower shoreface) facies associations (Table 1). AF4a is characterized by the presence
of brachiopods, crinoids, and diversified bioturbations, by the absence of emersion, and by the
greater amplitude of the sets in a dominant mud lithology (Eschard et al., 1999). AF4b is
heterolithic and composed of fine to medium-grained sandstones (brownish) interbedded with
argillaceous siltstones and bioclastic carbonated sandstones. Sedimentary structures include
oscillation ripples, swaley cross-bedding, flaser bedding, cross-bedding, convolute bedding,
wavy bedding, and low-angle planar cross-stratification. Sediments were affected by



moderate to highly diversified bioturbation by *Skolithos* (Sk), *Cruziana*, *Planolites*, (Pl)
*Chondrites* (Ch), *Teichichnus* (Te), *Spirophytons* (Sp) and are composed of ooids, crinoids,
bryozoans, stromatoporoids, tabulate and rugose corals, pelagic styliolinids, neritic
tentaculitids, and brachiopods. AF4c can be distinguished by a low sand/shale ratio, thick
interbeds, abundant HCS, deep groove marks, slumping, and intense bioturbation (Table 1).
AF4a is interpreted as a lagoonal shoreface. The gamma-ray pattern (see I Fig. 8) is
characterized by a concave bell shape influenced by a low sand/shale ratio with values
fluctuating between 100 and 200 gAPI. AF4a is identified in the "Tigillites Talus" Formation
and the Emsian sequence of the Illizi basin (Eschard et al., 2005) and in the Lochkovian
"Oued Samene" Formation (Wendt et al., 2006). AF4b is interpreted as a shoreface
environment. The presence of swaley cross-bedding produced by the amalgamation of storm
beds (Dumas and Arnott, 2006) and other cross-stratified beds is indicative of upper shoreface
environments (Loi et al., 2010). The gamma-ray pattern (see J and K Fig. 8) displays concave
erratic egg shapes with a very regularly decreasing-upward trend and ranging from offshore
shale with high values (80 to 60 gAPI) to clean sandstone with low values at the top (40 to 60
gAPI). AF4b is observed in the "Atafaitafa" Formation corresponding to the "Passage zone"
Formation of the Illizi basin (Eschard et al., 2005). AF4c is interpreted as a lower shoreface
environment (Dumas and Arnott, 2006; Suter, 2006). The gamma-ray pattern displays the
same features as the upper shoreface deposits with lower values (i.e. muddier facies) ranging
from 100 to 80 gAPI (see J and K Fig. 8).
**5.1.4   Offshore marine environments**
This depositional environment is composed of AF5a and AF5b facies associations (table 1).
AF5a is mainly defined by wavy to planar-bedded heterolithic silty-shales interlayered with
fine-grained sandstones. It also contains bundles of skeletal wackestones and calcareous



mudstones. The main sedimentary structures are lenticular sandstones, rare hummocky cross-bedding, mud mounds, low-angle cross-bedding, tempestite bedding, slumping, and deep groove marks. Sediments can present rare horizontal bioturbation such as *Zoophycos* (Z), *Teichichnus* (Te), and *Planolites* (Pl). AF5b is characterized by an association of black silty shales with occasional bituminous wackestones and packstones. It is composed of graptolites, gonitaties, orthoconic nautiloids, pelagic pelecypods, limestone nodules, tentaculitids, ostracods, and rare fish remains. Rare bioturbation such as *Zoophycos* (Z) is visible.

In AF5a, the occurrence of HCS, the decrease in sand thickness and grain size together with the fossil traces indicate a deep marine environment under the influence of storms (Aigner, 1985; Reading, 2002). The gamma-ray pattern is serrated and erratic with values well grouped around high values from 120 to 140 gAPI (see L Fig. 8). Positive peaks may indicate siltstone to sandstone ripple beds. AF5b is interpreted as lower offshore deposits (Aigner, 1985; Stow and Piper, 1984; Stow et al., 2001). Here again the gamma-ray signature is serrated and erratic with values well grouped around 140 gAPI (see L Fig. 8). Hot shales with anoxic conditions are characterized by gamma-ray peaks (>140 gAPI). These gamma-ray patterns are typical of offshore environments dominated by shales (Rider, 1996; Serra, 2009; Wagoner et al., 1990). AF5a and AF5b are observed in the Silurian "Graptolites shales" Formation and the Emsian "Orsine" Formation of the Illizi basin (Beuf et al., 1971; Eschard et al., 2005; Legrand, 1996, 1986). The "Meden Yahia" and "Termatasset" Shales have the same facies (Wendt et al., 2006).

## 5.2 Sequential framework and unconformities

The high-resolution facies analysis, depositional environments, stacking patterns, and surface geometries observed in the Paleozoic succession reveal at least two different orders of depositional sequences (large and medium scale, Fig. 7) considered as



transgressive/regressive T/R (e.g. Catuneanu et al., 2009). The sequential framework
proposed in Fig. 7B result from the integration of the vertical evolution the main surfaces
(Fig. 7A) and the gamma-ray pattern (Fig. 8). The Devonian series under focus exhibits nine
medium-scale sequences (D1 to D9, Fig. 7; Figs 9, 10, supplementary data 2 and 3) bounded
by 10 major sequence boundaries (HD0 to HD9), and nine major flooding surfaces (MFS1 to
MFS9). The correlation of the different sequences at the scale of the different basins and
arches is used to build two E–W (Figs 9, 10, supplementary data 2) and one N–S
(supplementary data 3) cross-sections.
The result of the analysis of the general pattern displayed by the successive sequences reveal
two major patterns (Figs 9 and 10) limited by a major flooding surface MFS5. The first
pattern extends from the Oued Samene to Adrar Morrat Formations and is dated from the
Lochkovian to Givetian. D1 to D5 medium-scale sequences indicate a general proximal
clastic depositional environment (dominated by fluvial to transitional and shoreface facies)
with intensive lateral facies evolution. This first pattern is thin (from 500 m in the basin
depocenter to 200 m around the basin rim) and with successive amalgamated surfaces on the
edge of the arches between the "Passage zone" and "Oued Samene" Formations (Figs 5C-C',
6A, 6C, 6D, and 9). It is delimited at the bottom by the HD0 surface corresponding to the
Silurian/Devonian boundary. D1 to D3 are composed of T-R sequences with a first deepening
transgressive trend indicative of a transition from continental to marine deposits bounded by a
major MFS and evolving into a second shallowing trend from deep marine to shallow marine
depositional environments. D1 to D3 thin progressively toward the edge and the continental
deposits, in the central part of the basin, pass laterally into a major unconformity. The
amalgamation of the surfaces and rapid lateral variations of facies between the Ahnet basin
and Azzel Matti and Arak-Foum Belrem arches demonstrate a tectonic control related to the
presence of subsiding basins and paleo-highs (i.e. arches).

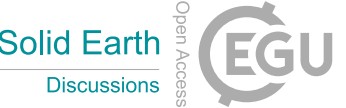

D4 and D5 display the same T-R pattern with a reduced continental influence and upward
decrease in lateral facies variations and thicknesses. The D5 sequence is mainly composed of
shoreface carbonates. Evidence of mud mounds preferentially located along faults are well-
documented in the area for that time (Wendt et al., 2006, 1997, 1993; Wendt and Kaufmann,
1998). This change in the general pattern indicates reduced tectonic influence.
MFS5, at the transition between the two main patterns, represents a major flooding surface on
the platform and is featured worldwide by deposition of "hot shales" during the early Frasnian
(Lüning et al., 2004, 2003; Wendt et al., 2006).
The second pattern extends from the "Meden Yahia", "Temertasset" to "Khenig" Formations
dated Frasnian to Lower Tournaisian. This pattern is composed of part of D5 to D9 medium-
scale sequences. It corresponds to homogenous offshore depositional environments with no
lateral facies variations. However, local deltaic (fluvio-marine) conditions are observed
during the Frasnian at the Arak Foum Belrem arch (Fig. 10). A successive alternation of
shoreface and offshore deposits is organized into five medium-scale sequences (part of D5,
and D6 to D9; Figs 9 and 10). This pattern corresponds to the general maximum flooding
(Lüning et al., 2003, 2004; Wendt et al., 2006) under eustatic control with no tectonic
influences.
**6    An association of low rate extensional subsidence and positive inversion pulses**
The backstripping approach (Fig. 11) was applied to five wells (W1, W5, W7, W17, and
W21). The morphology of the backstripped curve and subsidence rates can provide clues as to
the nature of the sedimentary basin (Xie and Heller, 2006). In intracratonic basins,
reconstructed tectonic subsidence curves are almost linear to gently exponential in shape,
similar to those of passive margins and rifts (Xie and Heller, 2006). The compilation of
tectonic backstripped curves from several wells in peri-Hoggar basins (Fig. 11A, see Fig. 1





for location) and from wells in the study area (Fig. 11B) display low rates of subsidence (from
5 to 50 m/Myr) organized in subsidence patterns of: Inversion of the Low Rate Subsidence
(ILRS type c, red line, Fig. 11C), Deceleration of the Low Rate Subsidence (DLRS type b,
black line), and Acceleration of the Low Rate Subsidence (ALRS type a, blue line).
Each period of ILRS, DLRS, and ALRS may be synchronous among the different wells
studied (see B1 to J, Fig. 11B) and some wells of published data (see D to J Fig. 11A).
The Saharan Platform is marked by a rejuvenation of basement structures, around arches (Figs
2, 3, and 4), linked to regional geodynamic pulses during Neoproterozoic to Paleozoic times
(Fig. 11). A compilation of the literature shows that the main geodynamic events are
associated with discriminant association of subsidence patterns:
(A) Late Pan-African compression and collapse (patterns a, b, and c, A Fig. 11A). The Infra-
Cambrian (i.e. top Neoproterozoic) is characterized by horst and graben architecture
associated with wedge-shaped unit DO0 in the basement (Fig. 9 and 10). This structuring
probably related to Pan-African post-orogenic collapse is illustrated by intracratonic basins
infilled with volcano-sedimentary molasses series (Ahmed and Moussine-Pouchkine, 1987;
Coward and Ries, 2003; Fabre et al., 1988; Oudra et al., 2005).
(B) Cambrian-Ordovician geodynamic pulse (Fig. 11A-B). Highlighted by the wedge-shaped
units DO1 (Figs 5A-A' and 6), the horst-graben system is correlated with deceleration (DLRS
pattern a, B1) and with local acceleration of the subsidence (ALRS pattern b, B2). The
Cambrian–Ordovician extension is documented on arches (Arak-Foum Belrem, Azzel Matti,
Amguid El Biod, Tihemboka, Gargaf, Murizidié, Dor El Gussa, etc.) of the Saharan Platform
by synsedimentary normal faults, reduced sedimentary successions (Bennacef et al., 1971;
Beuf et al., 1971, 1968a, 1968b; Beuf and Montadert, 1962; Borocco and Nyssen, 1959;
Claracq et al., 1958; Echikh, 1998; Eschard et al., 2010; Fabre, 1988; Ghienne et al., 2013,




2003; Zazoun and Mahdjoub, 2011) and by stratigraphic hiatuses (Mélou et al., 1999;
Oulebsir and Paris, 1995; Paris et al., 2000; Vecoli et al., 1999, 1995).
(C) Late Ordovician geodynamic pulse (i.e. Hirnantian glacial and isostatic rebound; Fig.
11A-B). Late Ordovician incisions mainly situated at the hanging walls of normal faults (Fig.
6C and 6D) are interpreted as Hirnantian glacial valleys (Le Heron, 2010; Smart, 2000) and
followed by local inversion of low rate subsidence (ILRS of type c, C in Fig. 11A).
(D) Silurian extensional geodynamic pulse (D, Figs 11A-B). The Silurian post-glaciation
period is featured by the reactivation and sealing of the inherited horst and graben fault
system (i.e. wedge-shaped unit DO2; Figs 5B-B', 5C-C', 6A and 6B). It is linked to an
acceleration of the subsidence (ALRS of pattern b in Fig. 11A-B).
(E) Late Silurian geodynamic pulse (Caledonian compression; E Fig. 11A-B). Late Silurian
times are marked by reactivation and local positive inversion of the former structures (Figs
5C-C' and 6B); by truncations located at fold hinges (Figs 5C-C' and 6); and by a major shift
from marine to fluvial/transitional environments (Figs 9, 10 supplementary data 2 and 3).
Backstripped curves register an inversion of the subsidence (ILRS of pattern c, in Fig. 11A-
B). The Caledonian event is mentioned as related to large-scale folding or uplifted arches (e.g.
the Gargaff, Tihemboka, Ahara, and Amguid El Biod arches) and it is associated with breaks
in the series and with angular unconformities (Beuf et al., 1971; Biju-Duval et al., 1968;
Boote et al., 1998; Boudjema, 1987; Boumendjel et al., 1988; Carruba et al., 2014; Chavand
and Claracq, 1960; Coward and Ries, 2003; Dubois and Mazelet, 1964; Echikh, 1998;
Eschard et al., 2010; Fekirine and Abdallah, 1998; Follot, 1950; Frizon de Lamotte et al.,
2013; Ghienne et al., 2013; Gindre et al., 2012; Legrand, 1967b, 1967a; Magloire, 1967).
(F) Early Devonian tectonic quiescence (F Figs 11A-B). This is characterized by a
deceleration of the low rate subsidence (DLRS of pattern a, F in Figs 11A-B).



(G) Middle to late Devonian geodynamic pulse (extension and local inversions, G Fig. 11A-
B). The Mid to Late Devonian period is characterized by large wedge hiatuses and truncations
associated with the reactivation of horst and graben structures and local positive inversion
(OD3 in Figs 5D-D', 6, 9, 10 supplementary data 2 and 3). This period is characterized by
inversion and acceleration of low rate subsidence (patterns c and b: ILRS - ALRS, Fig. 11A-
B). Some of the Middle to Late Devonian hiatuses (Early Eifelian) are noticed in the Ahnet
basin (Hassan Kermandji et al., 2008, 2003; Kermandji, 2007; Kermandji et al., 2009; Wendt
et al., 2006), in the Reggane (Jäger et al., 2009), on the Amguid Ridge (Wendt et al., 2006),
and in the Illizi basin (Boudjema, 1987; Chaumeau et al., 1961).
(H to K) Pre-Hercynian to Hercynian geodynamic pulses (Fig. 11A-B). This period is
organized in Early Carboniferous pre-Hercynian (H, Fig. 11A-B) to Late Carboniferous–Early
Permian Hercynian (K, Fig. 11A-B) compressions limited by Mid Carboniferous tectonic
quiescence (J, Fig. 11A-B). The Carboniferous period is characterized by a normal
reactivation and local positive inversion of the previous structural patterns involving reverse
faults, overturned folds, transpressional flower structures along strike-slip fault zones (Figs 3,
5F F', 6B, 6C and 6D). The major Carboniferous tectonic event on the Saharan Platform
impacted all arches and it is mainly controlled by near-vertical basement faults with a strike-
slip component (Boote et al., 1998; Caby, 2003; Liégeois et al., 2003; Haddoum et al., 2001,
2013; Zazoun 2008; J. Wendt et al., 2009 Carruba et al., 2014). Two major hiatuses (i.e. Mid
Tournaisian to Mid Visean– Serpukhovian) are recognized (Wendt et al., 2009b).
The geodynamic pulses attest to the reactivation of the terranes and associated lithospheric
fault zones. This observation questions the nature of the Precambrian basement and associated
structural heritage.



## 7 Precambrian structural heritage: accreted lithospheric terranes limited by vertical strike-slip mega shear zones

The 800 km² outcrop of basement rocks of the Hoggar shield provides an exceptional case of an exhumed mobile belt composed of accreted terranes of different ages. The Hoggar shield is composed of several accreted, sutured, and amalgamated terranes of various ages and compositions resulting from multiple phases of geodynamic events (Bertrand and Caby, 1978; Black et al., 1994; Caby, 2003; Liégeois et al., 2003).

To reconstruct the nature of the basement, a terrane map (Fig. 12) was put together by integrating geophysical data (aeromagnetic anomaly map: https://www.geomag.us/, Bouguer gravity anomaly map: http://bgi.omp.obs-mip.fr/), satellite images (7ETM+ from Landsat USGS: https://earthexplorer.usgs.gov/) data, geological maps (Berger et al., 2014; Bertrand and Caby, 1978; Black et al., 1994; Caby, 2003; Fezaa et al., 2010; Liégeois et al., 1994, 2003, 2005, 2013), and geochronological data (e.g. U-Pb radiochronology, see supplementary data 5). Geochronological data from published studies were compiled and georeferenced (Fig. 1). Thermo-tectonic ages were grouped into eight main thermo-orogenic events (Archean, Eburnean (i.e. Paleproterozoic), Kibarian (i.e. Mesosproterozoic), Neoproterozoic oceanization-rifting, Neoproterozoic Pan-African orogeny, Caledonian orogeny, and Hercynian orogeny). Geochronological data show that the different terranes were reworked during several main thermo-orogenic events. Twenty-three well preserved terranes in the Hoggar were identified and grouped into Archean, Paleoproterozoic, and Mesoproterozoic– Neoproterozoic juvenile Pan-African terranes (see legend in Fig. 1). In the West African Craton, the Reguibat shield is composed of Archean terrains in the west and of Paleoproterozoic terranes in the east (Peucat et al., 2005, 2003). The two main events deduced from geochronological data are the Neoproterozoic (i.e. Pan-African) and Paleoproterozoic (i.e. Eburnean) episodes (Bertrand and Caby, 1978). Aeromagnetic anomaly surveys are

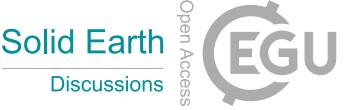

commonly used to analyze geological features such as rock types and fault zones (Gibson and
Millegan, 1998; Schubert, 2007; Vacquier et al., 1951). In this study, these data highlight the
geometries and the extension of the different terranes under the sedimentary cover. Four main
domains can be identified from the aeromagnetic anomaly map, delimited by contrasted
magnetic signatures and interpreted as suture zones (thick black lines, Fig. 12A). The study
area is bounded to the south by the Tuareg Shield (TS), to the north, by the south Atlasic
Range, to the west by the West African Craton (WAC) and at the east by the East Saharan
Craton (ESC) or Saharan Metacraton (Abdelsalam et al., 2002).
The magnetic disturbance features (Fig. 12A) show three main magnetic trends. A major NS
sinuous fabric and two minor sinuous 130–140°E and N45°E trends. The major NS
lineaments coincide with terrane boundaries and mega-shear zones (e.g. 4°50', 4°10', WOSZ,
EOSZ, 8°30', RSZ shear zones; Fig. 1). Sigmoidal-shaped terranes 200 to 500 km long and
100 km wide are characterized (red lines in Fig. 12A). The whole assemblage forms a typical
SC-shaped shear fabric (cf. Choukroune et al., 1987) associated with vertical mega-shear
zones and suture zones (e.g. WOSZ, EOSZ, 4°10', 4°50' or 8°30' Hoggar shear zones in Fig.
1). The SC fabrics combined with subvertical lithospheric shear zones are typical features of
the Paleoproterozoic accretionary orogens (Cagnard et al., 2011; Chardon et al., 2009). This
architecture is concordant with the Neoproterozoic collage of the Tuareg Shield (i.e. mobile
belt) between the West African Craton and the East Saharan Craton (i.e. cratonic blocks)
described by Coward and Ries, (2003) and Craig et al. (2006).
The gravimetric anomaly map (Fig. 12B) shows a correlation between gravimetric anomalies
and tectonic architecture (intracratonic syncline-shaped basin and neighboring arches).
Positive anomalies (> 66 mGal) are mainly associated with arches whereas negative
anomalies are related to intracratonic basins (< 66 mGal). Nevertheless, negative anomaly
disturbance is found in the Hoggar massif probably due to Cenozoic volcanism and the





Hoggar swell (Liégeois et al., 2005) or to Eocene Alpine intraplate lithospheric buckling
(Rougier et al., 2013). Arches are linked to Archean to Paleoproterozoic continental terranes
in contrast to syncline-shaped basins which are associated with Meso-Neoproterozoic terranes
(Figs 1 and 12A-B-C).

**8 Low subsidence rate intracratonic Paleozoic basins of the Central Sahara provide a**
**basis for an integrated modeling study**

Paleozoic intracratonic basins with similar characteristics (architecture, subsidence rate,
stratigraphic partitioning, alternating episodes of intraplate extension and short duration
compressions with periods of tectonic quiescence, etc.) have been documented in North
America (e.g. Allen and Armitage, 2011; Beaumont et al., 1988; Burgess, 2008; Burgess et
al., 1997; Eaton and Darbyshire, 2010; Pinet et al., 2013; Potter, 2006; Sloss, 1963; Xie and
Heller, 2006), South America (e.g. Allen and Armitage, 2011; de Brito Neves et al., 1984; de
Oliveira and Mohriak, 2003; Milani and Zalan, 1999; Soares et al., 1978; Zalan et al., 1990),
Russia (e.g. Allen and Armitage, 2011; Nikishin et al., 1996) and Australia (e.g. Harris, 1994;
Lindsay and Leven, 1996; Mory et al., 2017). However, the nature of the potential driving
processes (lithospheric folding, far-field stresses, local increase in the geotherm, mechanical
anisotropy from lithospheric rheological heterogeneity, etc.) associated with the formation of
intracratonic Paleozoic basins remains highly speculative (e.g. Allen and Armitage, 2011;
Armitage and Allen, 2010; Braun et al., 2014; Burgess and Gurnis, 1995; Burov and
Cloetingh, 2009; Cacace and Scheck-Wenderoth, 2016; Célérier et al., 2005; Gac et al., 2013;
Heine et al., 2008; Leeder, 1991; Vauchez et al., 1998).
The multiscale and multidisciplinary analysis performed in this study enable us to document a
model of Paleozoic intracratonic Central Saharan basins coupling basin architecture and
basement structures (Fig. 13). While we do not provide any quantitative explanations for the
dynamics of these basins, our synthesis highlights that their subsidence is not the result of a



single process and we attempt here to make a check-list of the properties that a generic model
of formation of such basins must capture:
(A) The association of syncline-shaped wide basins and neighboring arches (i.e. paleo-highs).
The structural framework shows a close association of syncline-shaped basins, inter-basin
principal to secondary arches, and intra-basin secondary arches (see Fig. 3).
(B) By local horst and graben architecture linked to steep-dipping planar normal faults and
associated with normal fault-related fold structures (i.e. forced folds; a, Fig. 13A). Locally,
the extensional structures are disrupted by positive inversion structures (b, Fig. 13A) or
transported normal faults (c, Fig. 13A).
(C) A low rate of subsidence ranging between 5 to 50 m/Myr (Fig. 11).
(D) Long periods of extension and tectonic quiescence are interrupted by brief periods of
compression or glaciation/deglaciation events (Beuf et al., 1971; Denis et al., 2007; Le Heron
et al., 2006). These periods of compression are possibly related to intraplate compression
linked to distal orogenies (i.e. Late Silurian Caledonian event, Late Carboniferous Hercynian,
Frizon de Lamotte et al., 2013, Ziegler et al., 1995) or to intraplate arch uplift related to
magmatism (Derder et al., 2016; Fabre, 2005; Frizon de Lamotte et al., 2013; Moreau et al.,

623     1994).

(E) Synsedimentary divergent onlaps and local unconformities are identified from integrated
seismic data, satellite images, and borehole data (Figs 4, 5, 6, 9 and 10). The periods of
tectonic activity are characterized by normal to reverse reactivation of border faults,
emplacement of wedge-shaped units, and erosional unconformities neighboring the arches
(Figs 3, 4, 5, 6, 9, 10 and 13).
(F) The stratigraphic architecture displays a lateral facies variation and partitioning between
distal marine facies infilling the intracratonic basins (i.e. offshore deposits) and proximal



amalgamated facies (i.e. fluvio-marine, shoreface) associated with prominent stratigraphic
hiatus and erosional unconformities in the vicinity of the arches.
(G) A close connection is evidenced between the period of tectonic deformation and the
presence of erosional unconformities (i.e. 2, 3, 6, 8, 10 geodynamic events in Fig. 13B). By
contrast, the periods of tectonic quiescence and extension are characterized by low lateral
facies variations, thin deposits, and the absence of erosional surfaces.
(H) The Precambrian heritage corresponds to Archean to Paleoproterozoic terranes identified
in the Hoggar massif and reactivated during the Meso-Neoproterozoic Pan-African cycle (Fig.
1). The Precambrian lithospheric heterogeneity illustrated by the different characteristics of
Precambrian terranes (wavelength, age, nature, fault zones) spatially control the emplacement
of the syncline-shaped intracratonic basins underlain by Meso–Neoproterozoic oceanic
terranes and the arches underlain by Archean to Paleoproterozoic continental terranes (Figs 1,
3 and 13). Many authors suggest control of the basement fabrics is inherited from the Pan-
African orogeny in the Saharan basins (Beuf et al., 1968a, 1971; Boote et al., 1998; Carruba
et al., 2014; Coward and Ries, 2003; Eschard et al., 2010; Guiraud et al., 2005; Sharata et al.,

2015).

**9    Conclusion**
Our integrated approach using both geophysical (seismic, gravity, aeromagnetic, etc.) and
geological (well, seismic, satellite images, etc.) data has enabled us to decrypt the
characteristics of the intracratonic Paleozoic Saharan basins and the control of the
heterogeneous lithospheric heritage of the horst and graben architecture, low rate subsidence,
association of long-lived broad synclines and anticlines (i.e. arches swells, domes, highs or
ridges) with very different wavelengths ($\lambda$) (tens to hundreds of kilometers). A coupled basin
architecture and basement structures model is proposed.





This study highlights a tight control of the heterogeneous lithosphere over the structuring of the intracratonic Central Saharan basin. This particular type of basin is characterized by a low rate of subsidence and fault activation controlling the homogeneity of sedimentary facies and the distribution of the main unconformities. The low rate activation of vertical mega-shear zones bounding the intracratonic basin during Paleozoic times contrasts markedly with classic rift kinematics and architecture. Three different periods of tectonic compressional pulses, extension and quiescence are identified and controlled the sedimentary distribution. An understanding of tectono-sedimentary interaction is key to understanding the distribution of the Paleozoic petroleum reservoirs of this first-order oil province.

**Acknowledgements**

We are most grateful to ENGIE/NEPTUNE who provided the database used in this paper and who funded the work. Special thanks to the data management service of ENGIE/NEPTUNE (especially Aurelie Galvani) for their help with the database.

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

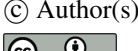



**List of figures**






Figure 1: Geological map of the Paleozoic North Saharan Platform (North Gondwana)
georeferenced, compiled and modified from (1) Paleozoic subcrop distribution below the
Hercynian unconformity geology of the Saharan Platform (Boote et al., 1998; Galeazzi et al.,
2010); (2) Geological map (1/500000) of the Djado basin (Jacquemont et al., 1959); (3)
Geological map (1/20000) of Algeria (Bennacef et al., 1974; Bensalah et al., 1971), (4)
Geological map (1/50000) of Aïr (Joulia, 1963), (5) Geological map (1/2000000) of Niger
(Greigertt and Pougnet, 1965), (6) Geological map (1/5000000 ) of the Lower Paleozoic of
the Central Sahara (Beuf et al., 1971), (7) Geological map (1/1000000) of Morocco (Hollard
et al., 1985), (8) Geological map of the Djebel Fezzan (Massa, 1988); Basement
characterization of the different terranes from geochronological data compilation (see
supplementary data) and geological maps (Berger et al., 2014; Bertrand and Caby, 1978;
Black et al., 1994; Caby, 2003; Fezaa et al., 2010; Liégeois et al., 1994, 2003, 2005, 2013);
Terrane names: Tassendjanet (Tas), Tassendjanet nappe (Tas n.), Ahnet (Ah), In Ouzzal
Granulitic Unit (IOGU), Iforas Granulitic Unit (UGI), Kidal (Ki), Timétrine (Tim), Tilemsi
(Til), Tirek (Tir), In Zaouatene (Za), In Teidini (It), Iskel (Isk), Tefedest (Te), Laouni (La),
Azrou-n-Fad (Az), Egéré-Aleskod (Eg-Al), Serouenout (Se), Tazat (Ta), Issalane (Is), Assodé
(As), Barghot (Ba), Tchilit (Tch), Aouzegueur (Ao), Edembo (Ed), Djanet (Dj); Shear zone
and lineament names: Suture Zone East Saharan Craton (SZ ESC), West Ouzzal Shear Zone
(WOSZ), East Ouzzal Shear Zone (EOSZ), Raghane Shear Zone (RSZ), Tin Amali Shear
Zone (TASZ), 4°10' Shear Zone, 4°50' Shear Zone, 8°30' Shear Zone.

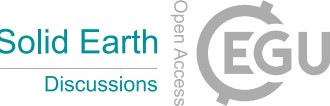


| AGE (Ma) | CHRONO-STRATIGRAPHY (1) | REGGANE (2) | AHNET (3) | MOUYRDIR (4) | ILLIZI (5) | TASSILI (6) | 2nd ORD. (7) | EUSTATIC & CLIMATIC CHART (8) | TECTONIC EVENTS (9) |
|---|---|---|---|---|---|---|---|---|---|
| 300–315 | CARBONIFEROUS / PENNSYLVANIAN — GZHELIAN, KASIMOVIAN, MOSCOVIAN | Hassi Bachir | | | Tiguentourine Series; Adeb Larache; Unit F | | E | Icehouse | Hercynian Compression |
| 315–330 | BASHKIRIAN / SERPUKHOVIAN | Djebel Berga; Gart Dehb | Red Fm.; Hassi Taibine Gypses; Djebel Berga Fm. | | Oued Oubarakat; Unit E; Unit D | | | | Pre-Hercynian Compression |
| 330–355 | MISSISSIPPIAN / VISEAN / TOURNAISIAN | Upper Tirechoumine; Kreb ed Douro; Low. Tirechoumine; Tibaradine; Teguentour | Garet Dehb Sandstones; Tirechoumine Shales (hV1); Iridets; Tibaradine Fm. (hTn2); Teguentour Shales | Garet Dehb Sandstones; Tirechoumine Shales; Kreb Ed Douro; Iridets; Tibaradine Fm.; Teguentour Shales | Assekaifaf; Issendjel Fm.; Unit C; Unit B; Unit A | | | | |
| 360–385 | DEVONIAN / UPPER / FAMENNIAN / FRASNIAN / GIVETIAN | Khenig Fm.; Temertasset Shales; Meden Yahia S.; Meden Yahia Shales; Hot shales; Azzel Matti Limestones; Adrar Morrat Shales | Khenig Fm. (dh); Temertasset Shales (d3b); Meden Yahia S.; Meden Yahia Shales (d3a); Hot shales; Takoula Limstones (d2b); Adrar Morrat Shales | Khenig Fm.; Temertasset Shales; Meden Yahia S.; Meden Yahia Shales; Hot shales; Adrar Morrat Shales | Gazelle fm.; F2; Shale Series; F3 Shale Series | Illerene; Tin Meras; Illizi Fm. | | Greenhouse | Middle/Late Devonian transtension & transpression |
| 390–405 | EIFELIAN / EMSIAN | Adrar Morrat Shales | Adrar Morrat Shales | Adrar Morrat Shales | Arrar Shaly Sandstones Fm.; F4 Shale Series; F5 Shale Series | Orsine | | | |
| 405–420 | LOWER / PRAGIAN / LOCHKOVIAN / PRIDOLI | Oued Samene Fm. (Dkhissa); Saheb El Djir; Zemlet Fm. | Oued Samene Fm. (diag, dSa); Idjorane Sandstones (dAs2); Passage Zone | Oued Samene Fm.; Passage Zone (sdAs1); Asedjrad Fm. | Hassi Taban. Sand.; C3 C2 C1; F6 Reservoir; Oued Titist | Upper Bar; Sidewalks; Middle Bar; Mederba | D | | "Caledonian" Compression |
| 420–435 | SILURIAN / LUDLOW / WENLOCK / LLANDOVERY | Oued Ali Shales; "Hot Shales" | Passage Zone; Tiounkeline; "Hot Shales" | Graptolites Shales; "Hot Shales" | Tifernine fm. (dTi); Graptolites Shales; "Hot Shales" | B2 B1; A M; Atafaïtafa (sdAt); Oued Imirhou formation (sIm); "Hot Shales" | | | Silurian Subsidence; Isostatic Rebound |
| 440–450 | HIRNANTIAN / ORDOVICIAN / UPPER | Tamadjert Fm. (U. IV) | Felar Felar | M'Kratta El Golea Microc. Shales | Gara-Louki; M'Kratta El Golea | Tamadjert (U. IV); OTj | B | Icehouse | Glacial Tectonism; "Taconic" event |
| 450–470 | UPPER / MIDDLE / KATIAN / SANDBIAN / DARRIWILIAN | In Tahouite Fm. (Unit III); Sandstones Fm. | Oued Saret; Unit III-3; Azzel Tiferouine Shales; Unit III-2; Ouargla; Hamra Q. | Oued Saret; Unit III-3; Azzel Tiferouine Shales; Unit III-2; Ouargla; Hamra Q. | Hassi Touareg Fm.; O. Saret Sand. Mb.; Q.S. Shaly Mb.; Azzel Shales Mb.; OuarglaMb.; Hamra Fm. | O. Saret Sand. Mb.; In Tahouite Fm. (Unit III-2) (OTh); Edjeleh Fm.; Banquette (Unit III-2) | | | Cambro-Ordovician Extension |
| 475–485 | LOWER / FLOIAN / TREMADOCIAN | | El Atchane; El Gassi; Alternation zone; Unit III-1 | El Atchane; El Gassi; Alternation zone; Unit III-1 | El Atchane Fm.; El Gassi Fm. | Vire du Mouflon (Unit III-1); In Kraf Fm. (Ajjers); Ham. | | | |
| 485–495 | CAMBRIAN / FURONGIAN / STAGE 10 / JIANGSHANIAN / PAIBIAN | Ajjers Sandstones (Unit II) | Ajjers/Azzel Matti Sandstones (Unit II) | Ajjers/Amguid Sandstones (Unit II) (coAj) | Ajjers/Hassi Leila Sandstones (Unit II) | Tin Taradjelli Sandstones | | Greenhouse | |
| 505–530 | SERIES 3 / SERIES 2 / STAGE 5,4,3,2 | | | | | | | | Infracambrian collapse extension |
| 535–540 | TERRENEUVIAN / FORTUNIAN | | Ouallen in Semmen & Bled El Mass Series | "Pourprée" Series (SSi) | El Moungar Conglomerat | El Moungar Conglomerat | A | | Panafrican Collision |
| | PRECAMBRIAN | | | | | | | | |



Figure 2: Paleozoic litho-stratigraphic, sequence stratigraphy and tectonic framework of the
North Peri-Hoggar basins (North African Saharan Platform) compiled from (1)
Chronostratigraphic chart (Ogg et al., 2016), (2) The Cambrian–Silurian (Askri et al., 1995)
and the Devonian–Carboniferous stratigraphy of the Reggane basin (Cózar et al., 2016;
Lubeseder, 2005; Lubeseder et al., 2010; Magloire, 1967; Wendt et al., 2006), (3) The
Cambrian–Silurian (Paris, 1990; Wendt et al., 2006) and the Devonian–Carboniferous
stratigraphy of the Ahnet basin (Beuf et al., 1971; Conrad, 1984, 1973; Legrand-Blain, 1985;
Wendt et al., 2009b, 2006), (4) The Cambrian–Silurian (Askri et al., 1995; Paris, 1990; Videt
et al., 2010) and the Devonian–Carboniferous stratigraphy of the Mouydir basin (Askri et al.,
1995; Beuf et al., 1971; Conrad, 1984, 1973; Wendt et al., 2009b, 2006), (5) The Cambrian–
Silurian (Eschard et al., 2005; Fekirine and Abdallah, 1998; Jardiné and Yapaudjian, 1968;
Videt et al., 2010) and the Devonian–Carboniferous stratigraphy of the Illizi basin (Eschard et
al., 2005; Fekirine and Abdallah, 1998; Jardiné and Yapaudjian, 1968), (6) The Cambrian–
Silurian (Dubois, 1961; Dubois and Mazelet, 1964; Eschard et al., 2005; Henniche, 2002;
Videt et al., 2010) and the Devonian–Carboniferous stratigraphy of the Tassili-N-Ajjers
(Dubois et al., 1967; Eschard et al., 2005; Henniche, 2002; Wendt et al., 2009b), (7) Sequence
stratigraphy of the Saharan Platform (Carr, 2002; Eschard et al., 2005; Fekirine and Abdallah,
1998), (8) Eustatic and climatic chart (Haq and Schutter, 2008; Scotese et al., 1999), (9)
Tectonic events (Boudjema, 1987; Coward and Ries, 2003; Craig et al., 2006; Guiraud et al.,
2005; Lüning, 2005); (A) Infra-Tassilian (Pan-African) unconformity, (B) Taconic and glacial
unconformity, (C) Isostatic rebound unconformity, (D) Caledonian unconformity, (E)
Hercynian unconformity.





Figure 3: (A) Geological map of the Paleozoic of the Reggane, Ahnet, and Mouydir basins.

Legend and references see Fig. 1. A: Inter-basin principal arch, B: Inter-basin boundary



secondary arch, C: Intra-basin secondary arch, D: Syncline-shaped basin. (B) E–W cross-
section of the Reggane, Ahnet, and Mouydir basins associated with the different terranes and
highlighting the classification of the different structural units (A: Inter-basin principal arch, B:
Inter-basin boundary secondary arch, C: Intra-basin arch, D: Syncline-shaped basin).
Localization of the interpreted sections (seismic profiles and satellite images). See figure 1 for
location of the geological map A and cross section B.

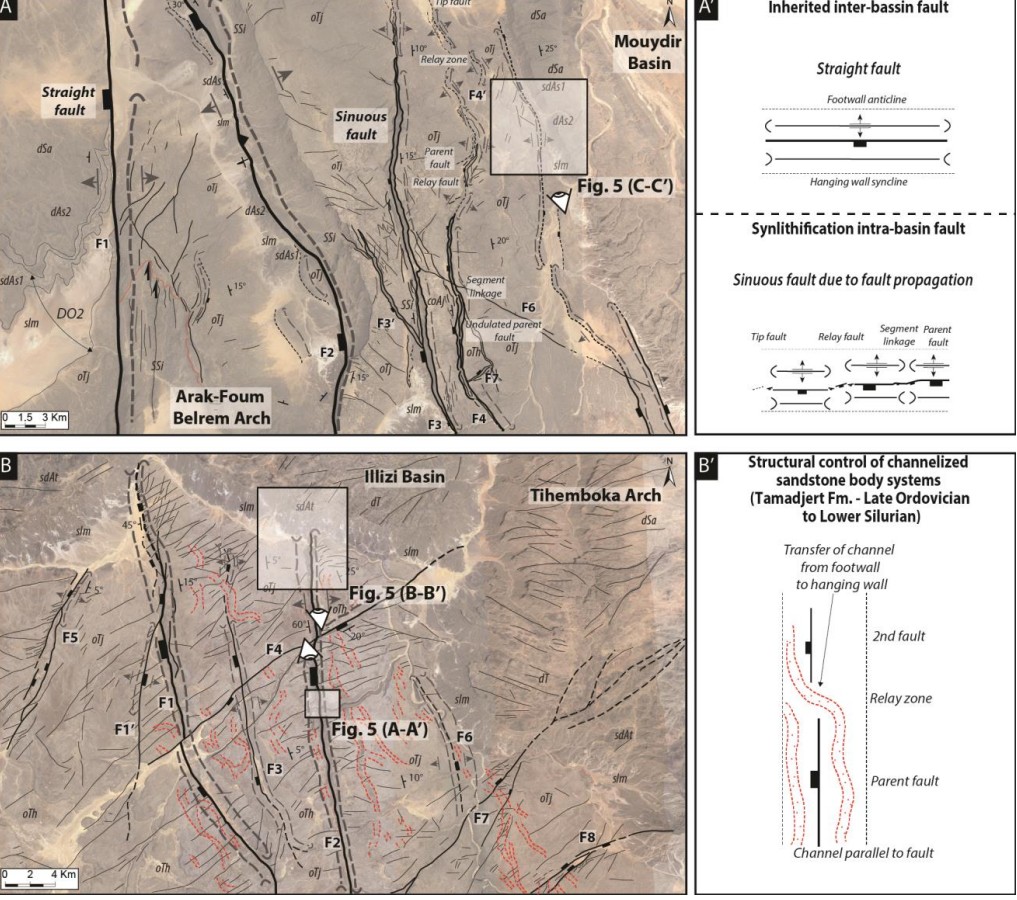


Figure 4: (A) Google Earth satellite images structural interpretation of the Djebel Settaf
(Arak-Foum Belrem arch; inter-basin boundary secondary arch between the Ahnet and





Mouydir basins) of the Cambrian–Ordovician series showing straight and sinuous normal
faults; (A') Typology of different types of faults (inherited straight faults vs sinuous short
synlithification propagation faults). (B) Google Earth satellite images structural interpretation
normal fault propagation in Cambrian–Ordovician series of South Adrar Assaouatene, Tassili-
N-Ajjers (Tihemboka inter-basin boundary secondary arch between the Illizi and Murzuq
basins); (B') Schematic model of structural control of channelized sandstone bodies. Dotted
red line: Tamadjert Fm. channelized sandstone bodies. *OTh*: In Tahouite Fm (Early to Late
Ordovician, Floian to Katian), *OTj*: Tamadjert Fm (Late Ordovician, Hirnantian), *sIm*:
Imirhou Fm (Early Silurian), *sdAs1*: Asedjrad Fm 1 (Late Silurian to Early Devonian), *dAs2*:
Asedjrad Fm 2 (Early Devonian, Lochkovian), *dSa*: Oued Samene Fm (Lower Devonian,
Pragian). See Fig. 3 for map and cross-section location.

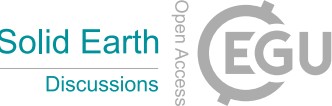







Figure 5: (A) Google Earth satellite images structural interpretation of South Adrar
Assaouatene, Tassili-N-Ajjers (Tihemboka inter-basin boundary secondary arch between the
Illizi and Murzuq basins) showing a normal fault (F2) associated with a footwall anticline and
a hanging wall syncline with divergent onlaps (i.e. wedge-shaped unit DO1) in the In
Tahouite series; (A') Cross-section between XY. 1: Cambrian–Ordovician extension during
the deposition of In Tahouite series (Early to Late Ordovician). See fig. 4B for location. (B)
Google Earth satellite images structural interpretation of North Adrar Assaouatene, Tassili-N-
Ajjers (Tihemboka inter-basin boundary secondary arch between the Illizi and Murzuq
basins) showing an ancient normal fault F2 escarpment reactivated and sealed during Silurian
deposition (poly-historic paleo-reliefs) linked to thickness variation, divergent onlaps (DO2)
in the hanging wall synclines, and onlaps on the fold hinge anticline; (B') Cross-section
between X'Y'. 1: Early to Late Ordovician extension, 2: Late Ordovician to Early Silurian
extension, 3: Middle to Late Silurian sealing (horizontal drape). See fig. 4B for location. (C)
Google Earth satellite images structural interpretation of Dejbel Settaf (Arak-Foum Belrem
arch; inter-basin boundary secondary arch between the Mouydir and Ahnet basins) showing a
normal fault (F5) associated with forced fold with divergent strata (syncline-shaped hanging
wall syncline and associated wedge-shaped unit DO2) and truncation in Silurian–Devonian
series; (C') Cross-section between XY. 1: Cambrian–Ordovician extension, 2: Silurian–
Devonian extensional reactivation (Caledonian extension). Red line: Unconformity. See fig.
4A for location. (D) Google Earth satellite images structural interpretation in the Ahnet basin
(Arak-Foum Belrem arch; inter-basin boundary secondary arch between the Mouydir and
Ahnet basins) showing blind basement normal fault (F1) associated with forced fold with in
the hanging wall syncline divergent onlaps of Lower to Upper Devonian series (wedge-
shaped unit DO3) and intra-Emsian truncation; (D') Cross-section between X'Y'. (E) Google
Earth satellite images structural interpretation in the Mouydir basin (near Arak-Foum Belrem



arch, eastward inter-basin boundary secondary arch) showing N170° normal blind faults F1
and F2 forming a horst-graben system associated with forced fold with Lower to Upper
Devonian series divergent onlaps (wedge-shaped unit DO3) and intra-Emsian truncation in
the hanging-wall syncline; (E') Cross-section between XY. (F) Google Earth satellite images
structural interpretation of Djebel Idjerane in the Mouydir basin (Arak-Foum Belrem arch,
eastwards inter-basin boundary secondary arch) showing an inherited normal fault F1
transported from footwall to hanging wall associated with inverse fault F1' and
accommodated by a detachment layer in Silurian shales series (Thickness variation of Imirhou
Fm between footwall and hanging wall) and spilled dip strata markers of overturned folding;
(F') Cross-section between X'Y'. 1: Cambrian–Ordovician extension, 2: Middle to Late
Devonian compression. *OTh*: In Tahouite Fm (Early to Late Ordovician, Floian to Katian),
*OTj*: Tamadjert Fm (Late Ordovician, Hirnantian), *sIm*: Imirhou Fm (Early to Mid Silurian),
*sdAt*: Atafaïtafa Fm (Middle Silurian), *dTi*: Tifernine Fm (Middle Silurian), *sdAs1*: Asedjrad
Fm 1 (Late Silurian to Early Devonian), *dAs2*: Asedjrad Fm 2 (Early Devonian, Lochkovian),
*dSa*: Oued Samene Fm (Early Devonian, Pragian), *diag*: Oued Samene shaly-sandstones Fm
(Early Devonian, Emsian?), *d2b*: Givetian, *d3a*: Meden Yahia Fm (Late Devonian, Frasnian),
*d3b*: Meden Yahia Fm (Late Devonian, Famennian), *dh*: Khenig sandstones (late Famennian
to early Tournaisian), *hTn2*: late Tournaisian, *hV1*: early Visean. See Figs. 1 and 3 for map
and cross-section location.





Figure 6: (A) N–S interpreted seismic profile in the Ahnet basin near Erg Tegunentour (near
Arak-Foum Belrem arch, westward inter-basin boundary secondary arch) showing steeply-
dipping northward basement normal blind faults associated with forced folding. Strata lapout
geometries shows Lower Silurian onlaps on the top Ordovician, Upper Silurian and Lower
Devonian truncations, onlaps and dowlaps of Frasnian series on top Givetian unit and onlap
near top Famennian. (B) NW–SE interpreted seismic profile of near Azzel Matti arch (inter-
basin principal arch) showing steeply-dipping south-eastwards basement normal blind faults
associated with forced folds. The westernmost structures are featured by reverse fault related
propagation fold. Strata lapout geometries show Silurian onlaps on top Ordovician, Frasnian
onlaps, thinning of Frasnian and Silurian series near the arch; truncation of Paleozoic series
by Mesozoic unit on Hercynian unconformity. (C) W–E interpreted profile of the Ahnet basin
(Arak-Foum Belrem arch, westward inter-basin boundary secondary arch) showing horst and
graben structures influencing Paleozoic tectonics associated with forced folds. Strata lapout
geometries show Precambrian basement tilted structure overlain by Cambrian–Ordovician
angular unconformity, incised valley in the Ordovician series, Silurian onlaps on top
Ordovician, Silurian onlaps on top Ordovician, Silurian–Devonian truncation, Frasnian onlaps
on top Givetian and near top Famennian onlaps. (D) W–E interpreted seismic profile of Bahar
el Hammar in the Ahnet basin (Ahnet intra-basin secondary arch) showing steeply-dipping
normal faults F1 and F2 forming a horst positively inverted associated with folding. Strata
lapout geometries show glacial valley in the Ordovician series, Silurian onlaps on top
Ordovician, Silurian onlaps on top Ordovician; Silurian–Devonian truncation re-folded,
Frasnian onlaps on top Givetian. Multiple activation and inversion of normal faults are
correlated to divergent onlaps (wedge-shaped units): DO0, Infra-Cambrian extension, DO1
Cambrian–Ordovician extension, DO2 Silurian extension with local Silurian–Devonian
positive inversion, and DO3 Frasnian–Famennian extension-local compression (transported

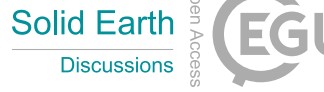

fault with a tectonic transport from footwall to hanging wall). See figure 3 for map and cross-
section location.







Figure 7: Core description, palynological calibration and gamma-ray signatures of well W7
modified from internal core description report (Dokka, 1999) and internal palynological
report (Azzoune, 1999). For location of well W7 see figure 3A. Lithological and
sedimentological studies were synthetized from internal Sonatrach (Dokka, 1999), IFP reports
(Eschard et al., 1999), and published articles (Beuf et al., 1971; Biju-Duval et al., 1968;
Wendt et al., 2006). Biozonations from Magloire (1967) and Boumendjel et al. (1988) are
based on palynological data from internal unpublished data (wells W1, W7, W12 W13, W19,
and W20 in Figs. 9, 10 supplementary data 3; Abdesselam-Rouighi, 1991; Azzoune, 1999;
Khiar, 1974). Well W18 is supported by palynological data and biozonations from Hassan
Kermandji et al. (2008).



| Facies associations | \[Criteria & characteristics\] Textures/Lithology | Sedimentary structures | Biotic/non biotic grains | Ichnofacies | Depositional environments | |
|---|---|---|---|---|---|---|
| AF1 | Conglomerates, mid to coarse sandstones, siltstones, shales | Trough cross-bedding, mud clasts, lag deposits, fluidal and overturn structures, imbricated grains, lenticular laminations, oblique stratification | Rare oolitic intercalations, imbricated pebbles, sandstones, ironstones, phosphorites, corroded quartz grains, calcareous matrix, brachiopod coquinas, phosphatized pebbles, hematite, azurite, quartz | Rare bioturbation | Fluvial | Continental (Fluvial) |
| AF2 | Silt to argillaceous fine sandstone | Current ripples, climbing ripples, crevasse splay, root traces, paleosols, plant debris | Nodules, ferruginous horizon | | Flood plain | |
| AF3a | Fine to coarse sandstones, argillaceous siltstones, shales (heterolithic) | Trough cross-bedding, some planar bedding, flaser bedding, mud clasts, mud drapes, root trace, desiccation cracks, water escape, wavy bedding, shale pebble, sigmoidal cross-bedding | Brachiopods, trilobites, tentaculites graptolites | Bioturbations, *Skolithos* (Sk), *Planolites*, (Pl) | Delta/Estuarine channels | Coastal Plain (Transitional Marine/Continental) |
| AF3b | Very coarse-grained poorly sorted sandstone | Trough cross-bedding, sigmoidal cross-bedding, abundant mud clasts and mud drapes | | Increasing upward bioturbation *Skolithos* (Sk) | Fluvial/Tidal distributary channels | |
| AF3c | Fine-grained to very coarse-grained heterolithic sandstone | Sigmoidal cross-bedding with multidirectional tidal bundles, wavy, lenticular, flaser bedding, occasional current and oscillation ripples, occasional mud cracks | | Intense bioturbation, *Skolithos* (Sk), *Planolites*, (Pl) *Thalassinoides* (Th) | Tidal sand flat | |
| AF3d | Mudstones, varicolored shales, thin sandstone layers | Occasional wave ripples, mud cracks, horizontal lamination, rare multi-directional ripples | Absence of ammonoids, goniatites, calymenids, pelecypod molds, brachiopods coquinas | Intense bioturbation, *Skolithos* (Sk), *Planolites*, (Pl) *Thalassinoides* (Th) | Lagoon/Mudflat | |
| AF4a | Silty mudstone associated with coarse to very coarse argillaceous sandstone, poorly sorted, heterolithic silty mudstone | Sigmoidal cross-bedding, abundant mud clasts, wavy, lenticular cross-bedding and flaser bedding, abundant current and oscillation ripples, mud drapes | Shell debris (crinoids, brachiopods) | Strongly bioturbated *Skolithos* (Sk), *Planolites*, (Pl) | Subtidal | Shoreface |
| AF4b | Fine to mid grained sandstones interbedded with argillaceous siltstone and mudstone, bioclastic carbonates sandstones, brownish sandstones and clays, silts | Oscillation ripples, swaley cross-bedding, bidirectional bedding, flaser bedding, rare hummocky cross-bedding, mud cracks (syneresis), convolute bedding, wavy bedding, combined flow ripples, planar cross low angle stratification, cross-bedding, ripple marks, centimetric bedding, shale pebbles | Ooids, crinoids, bryozoans, coral clasts, fossil debris, stromatoporoids, tabulates, colonial rugose corals, myriad pelagic styliolinids, neritic tentaculitids, brachiopods, iron ooliths, abundant micas | *Skolithos* (Sk), *Cruziana*, *Planolites*, (Pl) *Chondrites* (Ch), *Teichichnus* (Te), *Spirophytons* (Sp) | Open marine-upper shoreface | |
| AF4c | Silty shales to fine sandstones (heterolithic) | Hummocky cross-bedding, planar bedding, combined flow ripples, convolute bedding, dish structures, mud drapes, remnant ripples, flat lenses, slumping | Intense bioturbation, *Cruziana* | *Thalassinoides* (Th), *Planolites* (Pl), *Skolithos* (Sk), *Diplocraterion* (Dipl), *Teichichnus* (Te), *Chondrites* (Ch), *Rogerella* (Ro), *Climactichnites* (Cl) | Lower shoreface | |
| AF5a | Grey silty-shales, bundles of skeletal wackestones, silty greenish shale interlayers fine grained sandstones, calcareous mudstones, black shales, polychrome clays (black, brown, grey, green, red, pink), grey and reddish shales | Lenticular sandstones, rare hummocky cross-bedding, mud mounds, mud buildups, low-angle cross-bedding, tempestite bedding, slumping, deep groove marks | Intensive burrowing, bivalve debris, horizontal burrows, skeletal remains (goniatites, orthoconic, nautiloids, styliolinids, trilobites, crinoids, solitary rugose, corals, limestones nodules, ironstone nodules and layers | *Zoophycos* (Z), *Teichichnus* (Te), *Planolites* (Pl) | Upper offshore | Offshore |
| AF5b | Black silty-shales (mudstones), bituminous mudstones-wackestones, packstones | Rare structures | parallel-aligned styliolinids, gonitaties, orthoconic nautiloids, pelagic pelecypod *Buchiola* anoxic conditions, limestone nodules, goniatites, *Buchiola*, *tentaculitids*, ostracods and rare fish remains, *Tornoceras*, *Aulatornoceras*, *Lobotornoceras*, *Manticoceras*, *Costamanticoceras* and *Virginoceras*, graptolites | *Zoophycos* (Z) | Lower offshore | |



Table 1: Synthesis of facies associations (AF1 to AF5), depositional environments, and
electrofacies in the Devonian series compiled from internal (Eschard et al., 1999) and
published studies (Beuf et al., 1971; Biju-Duval et al., 1968; Wendt et al., 2006).



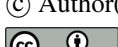



Figure 8: The main depositional environments (A to L) and their associated electrofacies (i.e.
gamma-ray patterns) modified and compiled from Eschard et al., (1999).





Figure 9: SE–W cross-section between the Reggane basin, Azzel Matti arch, Ahnet basin,
Arak-Foum Belrem arch, Mouydir basin, and Amguid El Biod arch (well locations in fig. 3).
Well W1 biozone calibration from Hassan, (1984) internal report is based on Magloire's
(1967) classification: biozone G3-H (Wenlock–Ludlow, Upper Silurian), biozone I-K
(Lochkovian–Emsian, Lower Devonian), biozone L1-3 (Eifelian–Givetian, Middle
Devonian), biozone L4 (Frasnian, Upper Devonian), biozone L5-7 (Famennian, Upper
Devonian), biozone M2 (Tournaisian–Lower Carboniferous). Well W7 biozone calibration
from Azzoune's (1999) internal report is based on Boumendjel's (1987) classification:
biozone 7-12 (Lochkovian, Lower Devonian), biozone 15 (Emsian, Lower Devonian).
Interpretation of the basement is based on Figs. 1, 3 and supplementary data 4. Outcrop
location is in Fig. 3.






Figure 10: NE–W cross-section between the Reggane basin, Azzel Matti arch, Ahnet basin,
Arak-Foum Belrem arch, Mouydir basin, and Amguid El Biod arch (well locations in fig. 3).
Well W18 biozone calibration is based on Kermandji et al. (2009): biozone (Tm) *tidikeltense*
*microbaculatus* (Lochkovian, Lower Devonian), biozone (Es) *emsiensis spinaeformis*
(Lochkovian-Pragian, Lower Devonian), biozone (Ac) *arenorugosa caperatus* (Pragian,
Lower Devonian), biozone (Ps) *poligonalis subgranifer* (Pragian–Emsian, Lower Devonian),
biozone (As) *annulatus svalbardiae* (Emsian, Lower Devonian), biozone (Mp) *microancyreus*
*protea* (Emsian–Eifelian, Lower to Middle Devonian), biozone (Vl) *velatus langii* (Eifelian,
Middle Devonian). Well W19 and W20 biozones calibration from internal reports
(Abdesselam-Rouighi, 1991; Khiar, 1974) is based on Magloire's (1967) classification:
biozone H (Pridoli, Upper Silurian), biozone I (Lochkovian, Lower Devonian), biozone J
(Pragian, Lower Devonian), biozone K (Emsian, Lower Devonian), biozone L1-5 (Middle
Devonian to Upper Devonian). Interpretation of the basement is based on Figs. 1, 3 and
supplementary data 4. Outcrop location is in Fig. 3.

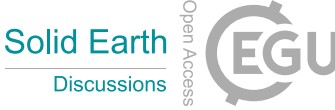

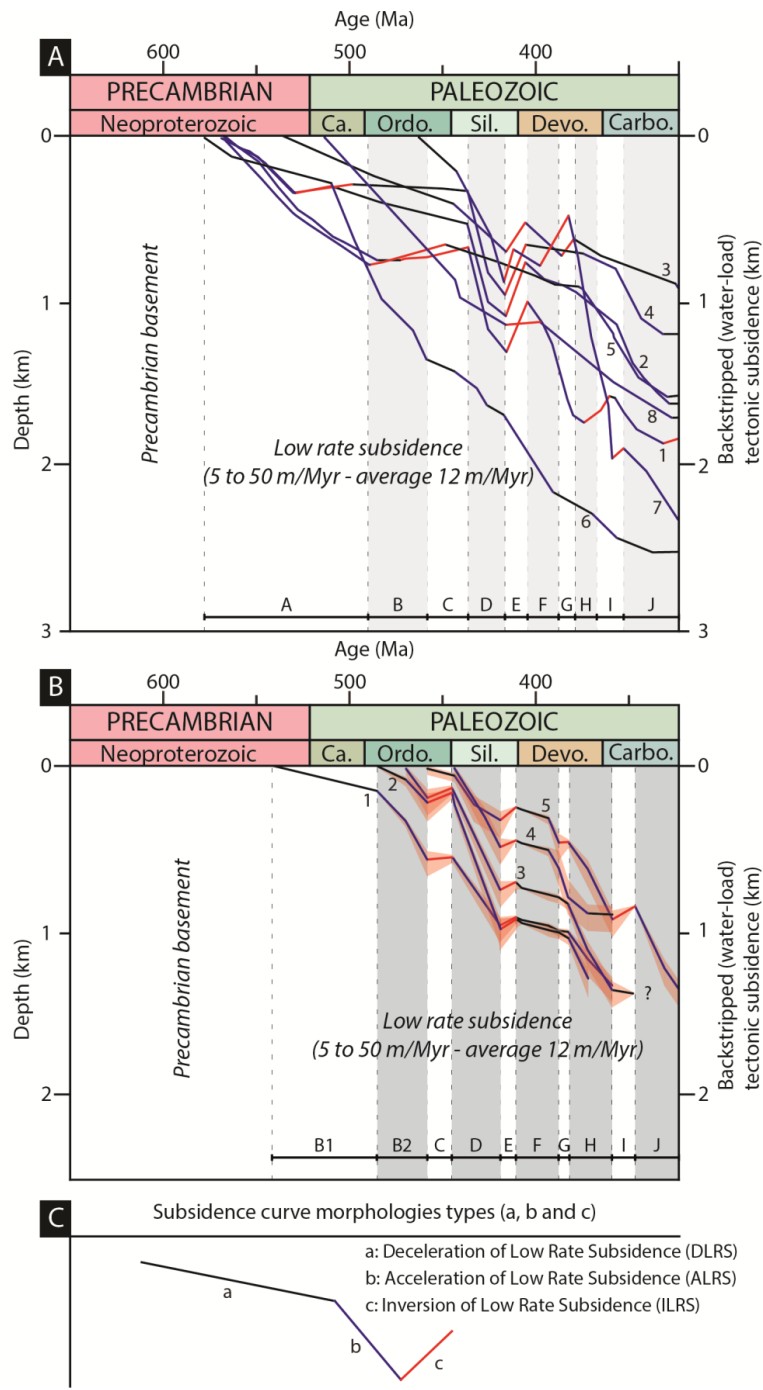



Figure 11: (A) Tectonic backstripped curves of the Paleozoic North Saharan Platform (peri-Hoggar basins) compiled from literature (1: HAD-1 well in Ghadamès basin (Makhous and Galushkin, 2003a); 2: Well RPL-101 in Reggane basin (Makhous and Galushkin, 2003a); 3: L1-1 well in Murzuq basin (Galushkin and Eloghbi, 2014); 4: TGE-1 in Illizi basin (Makhous and Galushkin, 2003b); 5: REG-1 in Timimoun basin (Makhous and Galushkin, 2003a); 6: Ghadamès-Berkine basin (Allen and Armitage, 2011; Yahi, 1999); 7: well in Sbâa basin (Tournier, 2010); 8: well B1NC43 in Al Kufrah basin (Holt et al., 2010). (B) Tectonic backstripped curves of wells in the study area (1: well W17 in Ahnet basin; 2: well W5 in Ahnet basin; 3: well W7 in Ahnet basin; 4: well W21 in Mouydir basin; 5: well W1 in Reggane basin); (C) The data show low rate subsidence with periods of deceleration (Deceleration of Low Rate Subsidence: DLRS), acceleration (Acceleration of Low Rate Subsidence: ALRS), or inversion (Inversion of Low Rate Subsidence: ILRS) synchronous and correlated with regional tectonic pulses (i.e. major geodynamic events). A: Late Pan-African compression and collapse (type a, b, and c subsidence), B: Undifferentiated Cambrian–Ordovician (type a, b, and c subsidence), B1: Cambrian–Ordovician tectonic quiescence (type a subsidence), B2: Cambrian–Ordovician extension (type b subsidence), C: Late Ordovician glacial and isostatic rebound (type c subsidence), D: Silurian extension (type b subsidence), E: Late Silurian Caledonian compression (type c subsidence), F: Early Devonian tectonic quiescence (type a subsidence), G-H: Middle to late Devonian extension with local compression (i.e. inversion structures, type b and c subsidence), I: Early Carboniferous extension with local tectonic pre-Hercynian compression (type c and b subsidence), J: Middle Carboniferous tectonic extension (type b subsidence), K: Late Carboniferous–Early Permian Hercynian compression (type c subsidence).







Figure 12: (A) Interpreted aeromagnetic anomaly map (https://www.geomag.us/) of the
Paleozoic North Saharan Platform (peri-Hoggar basins) showing the different terranes
delimited by NS, NW–SE and NE–SW lineaments and mega-sigmoid structures (SC shear
fabrics); (B) Bouguer anomaly map (from International Gravimetric Bureau:
http://bgi.omp.obs-mip.fr/) of North Saharan Platform (peri-Hoggar basins) presenting
evidence of positive anomalies under arches and negative anomalies under basins; (C)
Interpreted map of basement terranes according to their age (compilation of data sets in Fig. 1
and supplementary data 4); (D) Satellite images (7ETM+ from USGS:
https://earthexplorer.usgs.gov/) of Paleoproterozoic Issalane-Tarat terrane, Central Hoggar
(see fig. 12C for location); (E) Interpreted satellite images of Paleoproterozoic Issalane-Tarat
terrane showing sinistral sigmoid mega-structures associated with transcurrent lithospheric
shear fabrics SC.





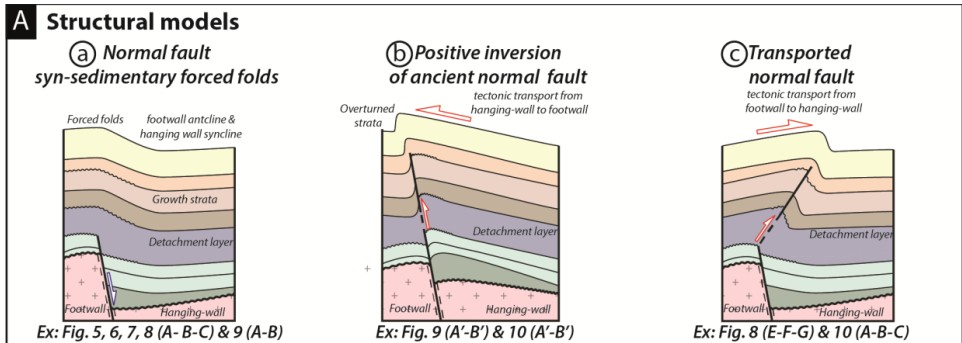

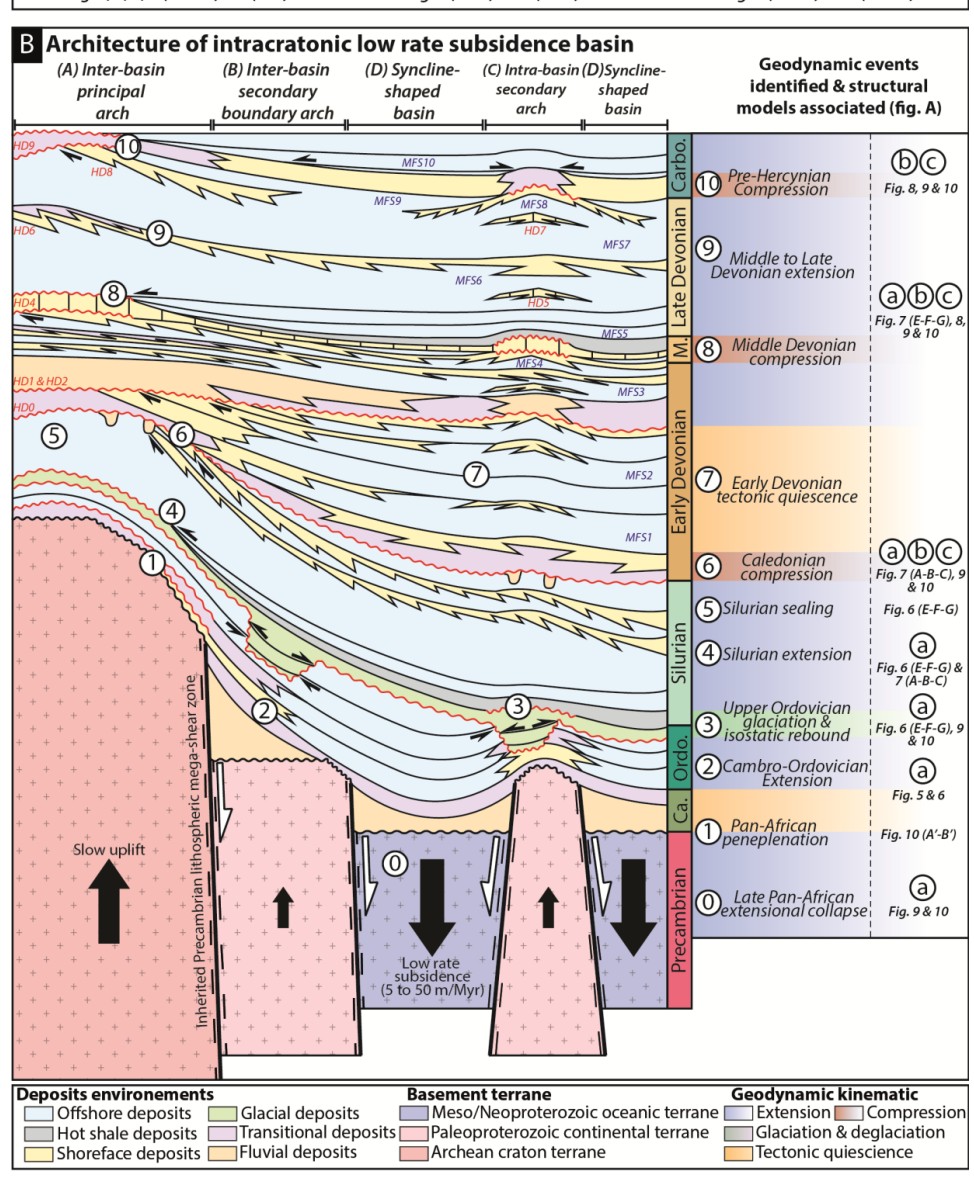



Figure 13: (A) Different structural model styles identified from the analysis of seismic
profiles and from interpretation of the satellite images; (B) Conceptual model of the
architecture of intracratonic low rate subsidence basin and synthesis of the tectonic kinematics
during the Paleozoic. Note that the differential subsidence between arches and basins is
controlled by terrane heterogeneity (i.e. thermo-chronologic age, rheology, etc.).