# Peer review of "Manuscript under review for journal Solid Earth"

_Solid Earth, 2018_

## Referee Comment (RC1) · R. S. Zazoun (Referee) · 20 Jul 2018

**Referee comments (RCs) I-Text 1- Lines 100-101; 106 to 110; 113; 115 to 116; 118 to 120. . . . . .: Put references from oldest to most recent. 2- Line 253: is composed of fluviatile Cambrian. The whole sedimentary series described in the literature is composed of fluviatile to Braid-deltaic plain Cambrian, not only fluviatile (eg. Brahmaputra River analogue), with a transitional facies from continental to shallow marine (Sabaou et al., 2009, p160). 3- Line 254:. . . . glacial Ordovician. Upper Ordovician glaciogenic deposits. . . . 4- Line 255:. . . . . argillaceous deep marine Silurian. . .argillaceous deep**

[Figure]

marine Silurian deposits. . .. 5- Line 255: Missing the last reference for the Silurian (Djouder et al., 2018). Djouder, H., Lüning,S ., Da Siva, A-C., Abdellah, H., Boulvain, F. (2018), Silurian deltaic progradation, Tassili n'Ajjer plateau, south-eastern Algeria: Sedimentology, ichnology and sequence stratigraphy. Journal of African Earth Sciences, Volume 142, June 2018, Pages 170-192 6- Line 266: the term Algerian platform don't exit, you speak probably about the Algerian part of Saharan platform

7- Line 300 and Figure 2: In the Illizi basin, these facies are mainly recorded in the Cambrian Ajjers Formation

In the Illizi Basin, and Saharan platform, the Ajjers Formation and/or the equivalent formation are dated: Upper Cambrian? to Ordovician (Tremadoc to Caradoc)(Fabre, 2005 ; Vecoli, 2000 ; Vecoli & Playford, 1997; Vecoli et al., 1995 ;Vecoli et al .,1999).

References for the datation:

Vecoli, M., 2000. Palaeoenvironmental interpretation of microphytoplancton diversity trends in the Cambrian–Ordovician of the northern Saharan Platform. Palaeogeography, Palaeoclimatology, Palaeoecology 160, 329–346.

Vecoli, M., Playford, G., 1997. Stratigraphically significant acritarchs in uppermost Cambrian to basal Ordovician strata of Northwest Algeria. Grana 36, 17–28.

Vecoli, M., Albani, R., Ghomari, A., Massa, D., Tongiorgi, M., 1995. Précisions sur la limite Cambrien–Ordovicien au Sahara Algérien (secteur de Hassi-R'mel). Comptes Rendus de l'Académie des Sciences, Paris 320 (IIa), 515–522.

Vecoli, M., Tongiorgi, M., Abdesselam-Rouighi, F.F., Benzarti, R., Massa, D., 1999. Palynostratigraphy of Upper Cambrian–Upper Ordovician intracratonic clastic sequences, North Africa. Bollettino della Società Paleontologica Italiana 38 (2–3), 331–341.

Vecoli, M., Videt, B., Paris, F., 2008. First biostratigraphic (palynological) dating of Middle and Late Cambrian strata in the subsurface of northwestern Algeria, North Africa: implications for regional stratigraphy. Review of Palaeobotany and Palynology 149 (1–

2), 57–62.

8- There was confusion about the subdivision used. Often, it is extracted from English papers. It is best to keep the original nomenclature in French, as it was established by geologists (BRPA, 1964; in Beuf et al., 1971), in fact: (Beuf et al., 1971, Tab-1, p.158: Bennacef et al., 1971; Fabre, 2005). From de base to the top, we observe: A- The Ajjers Formation It's divided from the bottom to the top: In the Tassilis-N-Ajjers (Outcrops) - Les Grès/Conglomérats d' El Moungar (Unit I) - Les Grès de Tin Taradjelli (Unit II) - La Vire du Mouflon (Unit III-1) - La Banquette (Unit III-2) In the Illizi basin (Subsurface equivalents) - (Unit II) : R3+R2+Ra +Ri - (Unit III-1) : La Zone des Alternances + Les Argiles d'El Gassi + les Grès d'El Atchane - (Unit III-2) : Les Quartzites de Hamra

B- The In Tahouite Formation (Unité III-3) (outcrops) (Unit III-3) : Les Grès de Ouargla + Argiles d'Azzel + Grès de Oued Saret (subsurface)

C- The Tamadjert Formation (Unit IV) (outcrops)

Ajjers Formation + In Tahouite Formation represent the Sequence 1 (Preglacial-deposits) Tamadjert Formation (Outcrops) and the equivalent in the subsurface (Unit-IV) is the sequence 2 (syn-glacial deposits)

Also for The Siluro-Devonian, keep the original nomenclature in French (Text and Fig.2): Tigillites Talus: Talus à Tigilites (Lines 348 ; 371) Passage Zone: Zone de passage (Lines 379, 427) Middle Bar: Barre Moyenne (Lines 301, 335) Sidewalks: Trottoirs Upper Bar: Barre Supérieure (Lines 301, 335). . ...etc.

References for The Ajjers Formation:

Bennacef, A., Beuf, S., Biju-Duval, B., De Charpal, O., Gariel, O., Rognon, P., 1971. Example of cratonic sedimentation: Lower Palaezoic of Algerian Sahara. The American Association of Petroleum Geologists Bulletin 55 (12), 2225–2245.

Beuf, S., Biju-Duval, B:, de Charpal, D., Rognon, R., Bennacef,A., 1971. Les gres du Paleozoïique inferieur au Sahara.Sedimentation et discontinuite: evolution structurale

d'un craton. Institut Francais Petrole, Collection Sciences Techniques P&role 18, 464∼.

Sabaou, N., Aït Salem, H., Zazoun, R.S., 2009. Chemostratigraphy, tectonic setting and provenance of the Cambro-Ordovician clastic deposits of the subsurface Algerian Sahara. Journal of African Earth Sciences 55 (3–4), 158–174.

Zazoun, R.S., Mahdjoub, Y., 2011. Strain analysis of Late Ordovician tectonic events in the In-Tahouite and Tamadjert Formations (Tassili-n-Ajjers area, Algeria). J. Afr. Earth 1266 Sci. 60, 63–78.

9- Line 489: as Hirnantian glacial valleys ….as Hirnantian glacial-Palæovalleys 10- Line 534: Hoggar shield ? sometimes, you speak about Tuareg shield (TS)…Hoggar is the massif and the Tuareg is the shield. 11- Line 526 : Missing reference: Hercynian Tectonic Event According to Zazoun (2001)……………………………………………………………………………………The basement fabric features exerted a very strong control on the structural evolution during the Hercynian deformation (See also Haddoum et al., 2001).

References for The Hercynian Tectonic Event

Haddoum, H., Guiraud, R., Moussine-Pouchkine, A., 2001. Hercynian compressional deformations of the Ahnet-Mouydir Basin, Algerian Saharan platform: far-field stress effects of the Late Paleozoic orogeny. Terra Nova 13, 220–226.

Zazoun, R, S. 2001. La tectogenèse hercynienne dans la partie occidentale du bassin de l'Ahnet et la région de Bled El-Mass, Sahara Algérien: un continuum de déformation Journal of African Earth Sciences Vo.32, N°4, 869-887.

12- Lines 671: Abdesselam-Roughi, F ; Abdesselam-Rouighi, F.F 13- Line 902: Fabre, J. 2005. Géologie du sahara occidental et central. Musée Royal de l'Afrique Centrale-Belgique.Tervuren African Geoscience Collection Vol.108. ISBN 90-75894-66-x ; ISSN : 1780-8551, 610 p. 14- Line 980: Hassan Kermandji, A.M or Kermandji, A.M.H., ?

15- For the terranes notion please see the last publication about this topic :

Sonia Brahimi, Jean-Paul Liégeois, Jean-François Ghienne, Marc Munschy, Amar Bourmatte. 2018. The Tuareg shield terranes revisited and extended towards the northern Gondwana margin: Magnetic and gravimetric constraints. Earth-Science Reviews, doi:10.1016/j.earscirev.2018.07.002 (Accepted manuscript)

16- Lines 64 and 1313 Missing : Intra-Arenig unconformity cited by (Fabre 2005, pp 169) in the North Africa, Eschard et al., (2010) and Kracha thesis, (2011) in the Ahnet Basin and Beuf et al., (1968-1971) in the Tassilli-N-Ajjers and Eschard et al. (2006) in the Berkine Basin. "Erosion occurred in many places during an intra-Arenig unconformity" (Eschard et al., 2010)

" Une discordance de faible amplitude, mais dont l'extension intéressera tout le Maghreb depuis l'Anti-Atlas occidental jusqu'à la Libye marque la fin. . .de ce premier cycle Ordovicien Âż c'est-à-dire Tremadoc et Arenig inférieur. . ..(Fabre, 2005) Âń De même, les Âń Argiles de Tiferouine Âż transgressent progressivement sur la série de Bled El Mass, et sont limitées à la base par une discordance angulaire conglomératique, qui pourrait bien correspondre à la discordance Intra-Arénig, mise en évidence dans de nombreuses régions sahariennes (Beuf et al., 1968; 1971; Eschard et al., 2006). . . .. . .... Âż in Kracha., 2011, pages 73-86-100

Âń Le passage des Âń Quartzites de Hamra Âż vers les Âń Grès de Ouargla Âż se fait de manière progressive, et l'on assiste par la suite à la troncature des termes sommitaux de cette formation sous la discordance Intra-Arénig qui annonce l'arrivée brutale des Âń Argiles d'Azzel-Tiferouine Âż. Les Âń Grès d'Oued Saret Âż sont peu préservés sous les ravinements glaciaires de l'Unité IV (et/ou la discordance Taconique) (Kracha, 2011). . . . .. .. Pages 113 et 121.

References for the Intra-Arenig unconformity Beuf. S., Biju-Duval. B., Mauvier, A., Legrand Ph. (1968). Nouvelles observations sur le "Cambro-Ordovicien" du Bled El Mass (Sahara central), Publ. Serv. Géol. Algérie, Bulletin n° 38, p.39-51.

Beuf. S., Biju-Duval. B., De Charpal. O., Rognon. P., Gariel. O., et Bennacef, A.

(1971). Les grès du Palaeozoique inferieur du Sahara. Publication Institute Français du Petroleum, Collection Science et Technologie du Petrole, Paris, No. 18. 484 p.

Eschard.R, Hellat.C, Malla.M, Bénamane.K, Betioui.H, Callot.J.P, Carpentier.B, Chelcheb.S, Couprie.E, Dahi.M, Delmarre.S, Desaubliaux.G, Deschamps.G, Euzen.T, Hachemi.L, Hannoun.R, Jacolin.J.E, Lassal.A, Leblond.C, Levêque.I, Lorant.F, Mokhtari.N, Lorin.T, Rabary.G, Rudkiewicz.J.L, Wattine.A. (2006). Berkine Gas project. Evaluation of the Gas potential in the Berkine basin (Algeria). Rap. Conf. Ifp.

Eschard. R, Braik.F, Bekkouche.D, Ben Rahuma.M, Desaubliaux.G Deschamps.R et Proust. J. N. (2010). Paleohighs: their influence on the North African Paleozoic petroleum systems. Geological Magazine, 147 (1), 28-41.

Kracha, N. 2011. Relations entre sédimentologie, fracturation naturelle et diagenèse d'un réservoir à faible perméabilité : application aux réservoirs de l'Ordovicien, Bassin de l'Ahnet, Sahara Central, Algérie. Thèse de Doctorat, Université des sciences et Technologies de Lille. 458p, Thèse non publiée.

17- It would be desirable to speak also about the work of Boote et al (1998). These authors define the notion of ''The Gondwana Super-cycle" between the infra-Cambrian (Panafrican) Unconformity and the Hercynian Unconformity. This Super-cycle has been divided in 02 Super-cycles: a Lower Gondwana Super-cycle and an Upper Gondwana Super-cycle. This nomenclature which is defined at the scale of north Africa also applies in South America (Souza Cruz et al., 2000) and Saudi Arabia (Sharland et al., 2000, Davidson et al. 2001), even if a slight diachrony may exist between these areas.

1- Lower Gondwana Super-cycle (Infra-Cambrian Unconformity to Caledonian Unconformity): is a major second order transgressive-regressive megacycle, in the sens of Vail et al. ( 1977) 2- Upper Gondwana Super-cycle (from the Caledonian Unconformity to the Hercynian Unconformity) (in: The Lower Paleozoic sedimentation of the Algerian Saharan craton, Tassili N'Ajjer oucrops. Field Guide Book, February 20-25, 2003. AAPG symposium, Algiers 18-20 February 2003, 103p). References for "The

[Figure]

Gondwana Super-cycle"

Davidson, L., Bestwetherick, S., Craig, J., Eales, M., Fisher,A., Himmmali, A., Jho, J., Mejrab, B. and Smart, J. 2000. The structure, stratigraphy and petroleum geology of the Murzuk basin, southwest Libya. In Geological exploration in Murzuq basin, (in: Eds M.A. Sola and D.Worsley), Elsevier, 295-320.

Sharland, P.R., Archer, R., Casey, D.M., Davies, R.B., Hall, S.H., Heward, A.P.,Horbury, A.D. and Simmons, M.D., 2001. Arabian Plate Sequence Stratigraphy. Geoarabia,Special Publications, 2, 371p, 2 plates.

Souza Cruz, C.E., Miranda, A. P., and Oller, J. 2000. Facies analysis and depositional systems of Late Silurian-Devonian subandean basin, southern Bolivia and Northern Argentina. In: Memorias del Congreso Geologico Boliviano, La Paz, 14-18 Noviembre 2000, 85-90.

18- Thermal History of Ahnet and Sbaa Basins (Histoire Thermique) (See Akkouche Thesis) : you will find below the conclusions of Mr. Akkouche about the Fission track study of Ahnet and Sbaâ

Akkouche, M. 2007, Application de la datation par traces de fission à l'analyse de la thermicité de bassins à potentialités pétrolières. Exemple de la cuvette de Sbaâ et du bassin de l'Ahnet-Nord (plate-forme saharienne occidentale, Algérie). Thèse de Doctorat, Université de Bordeaux 1, 297p.

You can download the thesis at the link below : www.iaea.org/inis/collection/NCLCollectionStore/_Public/41/021/41021844.pdf

Histoire Thermique du bassin de l'Ahnet Les traces de fission témoignent d'épisodes thermiques et d'érosion différents suivant les domaines du bassin avec une tendance générale à l'effacement des traces du Nord vers le Sud. Au Nord du bassin de l'Ahnet (MRS-1 et MSL-1), les âges obtenus dans les formations de l'Ordovicien et de la base du Silurien sont de 50 Ma environ. Ils témoignent d'une phase thermique post-varisque

qui a affecté la colonne paléozoïque. On peut estimer que les températures atteintes au cours de cet épisode thermique, probablement >100°C, sont également à mettre en relation avec l'épisode thermique d'extension régionale d'âge triasico-jurassique. Cet événement thermique pourrait également être à l'origine de l'effacement total des traces pré-existantes du niveau dévonien du forage MSR-1 qui présente, à 505 m sous la discordance varisque, un âge TF de 100 Ma. Plus au Sud, les âges respectifs de 37 Ma et 26 Ma obtenus à des profondeurs de 1030 m (forage MKRN-1) et 1532 m (forage BH-5), suggèrent une altération thermique cénozoïque des traces de fission plus intense que celle mise en évidence dans le Nord du bassin. Cette caractéristique pourrait refléter une érosion récente plus importante, mais également l'action éventuelle de gradients géothermiques plus élevés

Histoire Thermique de la cuvette de Sbaâ basin (See Akkouche Thesis) La plupart des résultats obtenus sur les échantillons prélevés dans les forages de la cuvette de Sbaâ est compatible avec ceux obtenus par les chercheurs de la Société GEOTRACK sur le forage OTRA-1, situé à 50 kilomètres du forage ODZ 1-bis. Dans le forage ODZ-1bis, les résultats suggèrent une histoire thermique marquée par trois événements de refroidissement. Le plus ancien se situe entre 200 et 300 Ma. Le second aurait un âge proche de 170-210 Ma et le plus récent entre 30 et 50 Ma. Ces âges témoigneraient successivement des événements de refroidissement carbonifère (fin de l'orogenèse varisque), jurassique (détumescence thermique post-rifting atlantique) et éocène (bombement du Hoggar). Un seul échantillon présente un résultat surprenant. Il s'agit de l'échantillon n°644 (forage ODZ-1bis) qui donne un âge de 472±11 Ma, prélevé dans un horizon détritique du Tournaisien. Onze des 12 apatites de cet échantillon présentent des traces de fission fossiles pré-datant nettement le Carbonifère. En d'autres termes, l'épisode varisque n'a jamais atteint un seuil thermique suffisant pour effacer ces traces au niveau de la bordure septentrionale de la cuvette de Sbaâ. Ce constat suggère que la cuvette de Sbaâ est un des domaines de la plate forme saharienne occidentale les moins affectées par les événements thermiques post-varisques. De toute évidence, à partir de la fin du Carbonifère, voire du Permien, la cuvette de

Sbaâ est structurée et demeurera dans un climat structural relativement superficiel.

II-Figures 1- Figure 2 : - Column 4 : Mouydir not Mouyrdir - Column 6 (Tassili) : In Tahouite Formation is the Unit III-3, not The Unit III-2 Tin Taradjelli Sandstones is the Unit II - Column 6: Orsine Formation not Orsine Tin Meras Formation not Tin Meras Illrene Formation not Illrene El Moungar Conglomerat is the Unit I - The intra-Arenig unconformity is missing

III- Conclusions : (See Referee comments above) - Several references have been omitted (Ajjers Formation, Silurian, Hercynian tectonic Event, Terranes. . .): see bibliography attached for each remark. - Very minor corrections are required. - The intra-Arenig unconformity was not mentioned. - Figure 2 is to be corrected. - Put references from oldest to most recent. - Some typographic mistakes about the names of the authors. - Ovoid the translation of the nomenclature in English language.

Réda Samy ZAZOUN Senior Géologue Structuraliste Sonatrach Division Petroleum Engineering & Développement 8, Chemin du Réservoir, Hydra, Alger, Algérie E-mail :reda-samy.zazoun@sonatrach.dz Alternatif E-mail: redasamy@hotmail.com

Please also note the supplement to this comment:
https://www.solid-earth-discuss.net/se-2018-50/se-2018-50-RC1-supplement.pdf

---

## Short Comment (SC1) · 3 Aug 2018

Perron et al. describe a study into the role of inherited structures in intracratonic basins in the Central Sarara using a combination of seismic interpretation, various GIS analyses, stratigraphy and geochronology. In my opinion the background information is adequate (just requiring a few minor amendments), the analyses appear to have been appropriately conducted and the discussion and conclusions appear to suitably draw upon the results. In addition, the figures are generally of high quality and should be commended. However, at times some sections seem longer than required and there

is potentially some repetition of information. The study is of current relevance with a number of recent papers also addressing similar topics, including a paper in Solid Earth (Phillips et al., 2018). Thus, Solid Earth seems like a suitable place for publication. I would therefore like to recommend publication if the relatively minor points suggested here and in the other comments are appropriately addressed. These minor points should not be too arduous, but I think that they will improve the manuscript. I hope you find these suggestions useful and I look forward to seeing a final version of the paper. If you would like clarification of any of my suggestions feel free to contact me.

First, I felt that it was not clear from the abstract what the purpose, aims and main findings of the study were. The reason for this appears to be that much of the abstract (and introduction to some extent) is devolved to regional background information. Although such information is obviously important I suggest more clearly outlining the main findings and study aims in the abstract. In addition, I think the introduction would benefit from a short but general overview of structural inheritance, including information from geological settings from beyond the present study to demonstrate the significance of such processes. Moreover, particularly in the abstract but also in the introduction, it is not always clear what is a finding of this study and what information is from previous work. I suggest Perron et al. clarify these sections with this in mind.

Related to the previous point is that the introduction contains a lot of material that would likely be better placed in the subsequent dedicated 'Geological setting' section. An example of this is the material in lines 59-75. I think that moving such information into the geological setting and using the introduction to better set up the aims and rationale would be better.

In addition to the previous points I think that the 'Data and methods' section requires additional information to be of more use to readers. For example 'Geographic Information System analysis (GIS)' (Line 124) is very ambiguous as this could mean any one of a number of approaches. Also minimal details are provided regarding the seismic

data or the methods deploying in its interpretation. I suggest that Perron et al. consider adding additional technical information regarding their data sets and the analyses used. Some of these details are provided later in various sections but I think they would be better placed here in the dedicated section.

The descriptions of results in sections 4-6 are generally very good. In particular, good use of the figures is made in the text. However, I found that in section 7 there is an abundance of material that might be better placed in the 'Geological Setting' or the 'Data and Methods' sections. Some examples of this are noted below but I suggest Perron et al. reconsider the location of some of the material in this section.

Other minor points include:

-Line 32 – This line is quite awkward. Suggest rewording.

-Line 35 – Replace 'activated' with 'reactivated'?

-Lines 59-60 – I don't quite understand this sentence.

-Line 89 – Figure 3 appears to be called before figure 2.

-Lines 89-93 – This opening paragraph of the geological setting feels like it needs references.

-Lines 130-132 – This reads more like results. Consider moving it.

-Line 145 – Suggest providing more details of the seismic data.

-Line 171 – 'oval' has been mentioned quite a few times before this. Is it really necessary to mention it this often?

-Line 258 – Suggest clarifying why the Devonian deposits are sensitive to such processes.

-Lines 534-538 – This paragraph reads more like geological setting.

-Lines 539-549 – This description of the analyses would be better placed in the 'Data

and methods' section.

-Lines 551-554 – The geological setting section might be more appropriate for this information.

---

## Referee Comment (RC2) · J. Wendt (Referee) · 27 Aug 2018

Preliminary remarks:

Though I have been working on the Paleozoic of the Algerian Sahara for many years (1987-2006) I am only familiar with the Devonian and Carboniferous, but not with the older formations and the crystalline basement. Therefore, I can only judge these aspects of the above manuscript. Likewise, I feel not competent enough to consider some tectonic reconstructions. I hope that the other reviewer(s) are able to review these as-

pects of the manuscript with a better competence.

The manuscript is an overview of the bio- and lithostratigraphic, sedimentologic, paleo-geographic and paleotectonic evolution of the Ahnet-Mouydir area in southern Algeria based on field data from previous authors, well log analysis, satellite images and geo-physical data. As such it is a good summary of the evolution of a marginal basin-and-ridge system which farther north in central Algeria has yielded enormous oil and gas reservoirs.

Detailed critical remarks

Title: The research areas covers a much larger area (including also the Reggane, Basin, Illizi Basin, Hoggar Shield) than expressed in the title. This should be made clear in the title.

Line 20: Pan-African orogeny. Strictly spoken this was around 600 MA, but including earlier phases it was 900-520 MA. What do you mean exactly?

Line 35: "Devonian compression". I consider this as a mere speculation. According to all previously gathered data the Devonian was a period of tectonic quiescence accom-panied by slight extension.

Line 52: 7 m/MA. Give reference.

Line 61: 16 million km2. Impossible! The entire Sahara occupies about 9 million km2.

Line 121 ff. and 133: It is not clear if the authors have ever been in the field; equivalent data seem to be based on previous published sources only. This should be made clear unequivocally.

Line 141: Please separate both calibration of well-logs by palynomorphs (which are poorly reliable biostratigraphic markers) and field sections by conodonts (which give by far the best time resolution), goniatites and brachiopods. Both biostratigraphic subdivi-sions can be only roughly be correlated.

Line 144 (and later): "Synsedimentary extensional and compressional markers": This means during the Devonian and Carboniferous. On which evidence these important tectonic events are based? Apparently not on field data. During about 9 months of personal field work I followed typical marker levels (e.g. the upper Eifelian/Givetian limestone ridge) for tens of kilometers (walking from ridge into basin deposits), but I have never seen something like that. The observation of doubtless Hercynian faults does not automatically allow the conclusion that they are rejuvenated earlier structures.

Line 146: Outcrop sections O1- O12 cannot be detected in Figs 9 and 10. Are they personal field data? Position of well logs W1-W21 can only very roughly be located from Fig. 3A. Given the importance of these data (which apparently have never been published previously) it is absolutely necessary to indicate individual coordinates (best as an appendix) for both.

Line 152: add: major "depositional" unconformities, in order to avoid confusion with angular unconformities.

Lines 153-154: The top Pragian unconformity is diachronous (comprises also the lowermost Emsian in the Reggane Basin and on the Azel Matti ridge). Top Givetian and top mid-Frasnian are no unconformities over the entire study area. Top Quaternary is an unconformity worldwide, therefore omit. Or do you mean base Quaternary? But this would be trivial. In this list you have omitted the most important depositional unconformity, the transgression of the lower Eifelian (costatus-Zone).

Line 156: geological map is 1: 200.000, not 1:20.000.

Line 171: circular of oval shape of basins. This is pure imagination. Basins and ridges are capped by erosion in the south and by overlying Jurassic or Cretaceous in the north. Thus the second dimension of the paleogeographic units is unknown.

Line 174: major faults are all Hercynian. Eventual pre-Hercynian faults are inferred, but have never been documented in the field, thus are mere speculation.

Line 178: "long" instead of "length".

After line 178: Generally, at this point there is a paragraph entitled "Previous work", but this is missing here.

Line 179: this chapter should be re-written avoiding speculations, even if they would fit well into a hypothetical and inferred depositional image. Regarding eventual "synsedimentary extensional markers" see above.

Line 191: Hercynian folding is restricted to the Reggane, Ahnet and western Mouydir Basins, but decreases markedly towards the east (eastern Mouydir and Illizi Basins) where Paleozoic strata are completely flat-lying.

Lines 205-207: synsedimentary horst and graben structures – see above (lines 174 and below). What is a "synsedimentary forced fold"? A slump?

Line 247: From Google Earth images it is possible to recognize faults, but it is impossible to determine their age. Please explain why the faults figures in Figs 4 and 6 are Silurian-Devonian and Middle to Late Devonian age.

Line 261: "Nine facies associations" cannot be detected in Figs 9 and 10. Do you mean the depositional environments? (these are 5). I also could not find the "supplementary data".

Line 291: There is no clear horizontal (gAPI) scale in Fig. 8. Thus it is impossible to check the numbers.

Line 298: values range to 120, not 200 in Fig. 8D.

Lines 329-330: 30-60 gAPI are low, not high.

Line 346: 25-60 gAPI are low, not high.

Line 366: stromatoporoids, tabulate and rugose corals are not mentioned on Tab. 1.

Line 378: same as above.

[Figure]

Line 382-83: same as above.

Line 395: HCS probably stands for hummocky cross stratification. If this should be the case, these structures indicate a shallow marine environment, not deep marine. The same interpretation refers to "influence of storms", i. e. shallow, not deep.

Line 396: The ichnofauna of AF5a does not necessarily indicate a deep marine environment, but could also be much more shallow, as indicated by the "influence of storms".

Line 406: The Grès de Mehden (not "Meden) Yahia and the Temertasset (not "Termatasset) shales were deposited during a regressive phase and should be discussed in one of the preceding paragraphs.

Line 410: not "Paleozoic" but "Devonian". Fig. 7 shows almost exclusively Devonian.

Line 421: The major flooding surface is not MFS5 but MFS4 (Eifelian transgression).

Line 423: same error. Moreover, you have omitted the gap in the Emsian.

Lines 433-436: This is highly exaggerated. The facies variations between the Ahnet Basin and the adjacent ridges are very weak.

Line 442: MFS5 is not a major flooding surface. The corresponding black shales are diachronous (earliest ones in the Givetian, latest ones in the upper Frasnian), and their occurrence depends mainly on paleogeographic factors. It is true that there is an evident gap between the Givetian and the Frasnian, but this occurs only on the ridges, not in the basins, and it is caused by non-deposition, not by transgression.

Line 451: not a maximum flooding but regression (see above). Line 514: an "early Eifelian" hiatus does not exist. Or do you mean the partitus Zone which in fact has not been documented? But I did not check the other references which appear to depend on palynomorph stratigraphy which, compared to conodont stratigraphy, is much less reliable.

Line 660: Which are the "Three different periods of tectonic compressional pulses"? I am aware only of one, the Hercynian.

Lines 668-1266: References:

The reference list occupies almost the same space as the preceding text and should be drastically reduced, at least to one half. In order to avoid the impression that the article is nothing but a general review paper. Only articles referring to the study area should be included in the reference list. Unfortunately, the latter in its present length shows many incomplete citations (missing volume, missing pages, missing dots in abbreviations, missing editor, missing town (for books), missing capitalizing, wrong spelling), such as in lines 673, 676, 681, 685, 690, 696, 699, 740, 743, 745, 755, 762, 764, 765, 777, 814, 828, 830, 844, 863, 873, 893, 900, 902, 938, 957, 963, 979, 982, 1001, 1003, 1013, 1018, 1033, 1037, 1041, 1075, 1081, 1082, 1095, 1099, 1112, 1124, 1129, 1158, 1160, 1162, 1169, 1176, 1181, 1185, 1186, 1195, 1221, 1222, 1226, 1244, 1253, 1255, 1257, 1260. This list, however, is not complete.

I did not check, if every reference in the text does also appear in the reference list and vice versa. This can be done much more accurately by a simple computer program (which I do not have). On the other hand, important local works are not cited.

Remarks to figures:

Fig. 1: line 1275: 1: 200.000.

Line 1281: where are the supplementary data?

Map and reference of Monod (1931-1932) are missing.

Fig. 2: Illizi Fm. Is missing in the Illizi column.

Fig. 3: give exact coordinates for wells (W1- W21) and for outcrops (O1 – O9). What are the latter? Own data or previously published ones? Why there is no cross section along the O1-O9 line?

Fig. 7: larger lettering is required. (I had to use a 3x magnifier to read it). What are the tiny arrows in the left gamma-ray-column?

Tab. 1: Please add a column with the equivalent individual formation names. In the present form this table is rather theoretical and shows no relation to the Devonian depositional areas.

Fig. 8: Because of its tiny lettering this figure is almost unreadable. Stages and formation names should be added for each sub-figure. The accompanying sections are unreadable. I could not check the source because the equivalent reference is incomplete. In the present form this figure appears rather useless. Gamma-ray-curves often do not correspond to their interpretation in the text (see above). It would make a certain sense, if there were a comparison with equivalent well logs in each sub-figure, but it would better to omit this figure completely.

Fig. 9: Needs larger lettering! In Fig. 2 the Emsian is a gap (which is correct), but in

Fig. 9 this stage is represented by strata, which is an obvious contradiction.

Fig. 10: same as Fig. 9.

Fig. 11: "K" is missing on A and B. (line 1486).

Fig. 12: larger lettering, the smallest ones are illegible.

Conclusion

As a whole the paper is well written, rather concise and accompanied by good illustrations (apart from the above remarks). It is an example of a modern interpretation of a basin and ridge paleogeography using all available techniques. An important contribution is the representation of well data which are difficult to obtain by non-oil geologists. Nevertheless, it cannot be overlooked that as a whole the paper appears to be based almost exclusively on pre-existing data. The personal contribution to the subject is difficult to distinguish Thus, in several aspects and conclusions the interpretations of

the data are not or only poorly compatible with well-established field data. Some of them are highly speculative. It should also be made clear that the depositional units (basins and ridges) are nothing else than the southern prolongation of the same (but more accentuated) ones farther north. It should also be clearly expressed that the basin-and-ridge paleotopography in the Ahnet and Mouydir is of relatively short duration (early Eifelian to early Famennian). The depositional pattern of the late Famennian and the Carboniferous Is totally different from the Devonian one. A Devonian sea-level curve would be highly desirable. Absolutely necessary are several block diagrams to show the basin-and-ridge configuration at various stages.

I recommend publication of the manuscript after major revision, but I would be glad to receive the revised manuscript once more before its final acceptance.

Jobst Wendt Department of Earth Sciences University of Tübingen, Germany

---

## Referee Comment (RC3) · F. Lottaroli (Referee) · 6 Sep 2018

Beside the technical and scientific value of the paper I have collected some observations on the structure of the article. How data are presented and how much is clear which is the original work performed with respect of what has been re-digested from previous published work. On this respect there is to me some room of improvement. The reader needs a bit of help in being directed towards the key messages. As it is structured now I found it a little difficult, too much relevant oservations in "brackets", too frequently reader is requested to look at more than one figures with reference to the

same concept, jumping ahed and back-word. I would suggest some degree of simplification. At the end, even if is important to mark how big has been the effort of to give spatial relevance to info olready published (e.g geochronology, etc...) the important think is to convey the new and original message of the work. I Have never seen in my life Figures Captions as these. Captions represents basically another article. The relevance of illustrations must be stated in the article text. The Figure needs clear legends, but captions , if possible must be concise.

Abstract Row 17 – 40: It is lacking reference to your original work and the results of it. What make this paper one of original scientific content? Which is the new aspects of your approach with respect to what ( a lot) have been already published on North Africa Paleozoic? Shorten the introductive remarks on Paleozoic Basins and expand the above.

Introduction R47_ No need to cite specific Figure of Heine for a general remarks like this, work citation is enough R48_ Non conventional exploration has revived interest...... explain why and where R50,51,60,61_ incorporate in text all the (i.e. in brackets), too much. ...... Example: Share several common features, being generally from circular to ...., being filled by continental to shallow marine........... etc. ...etc R64_Depositional environment....This is just the same you just stated in Row 51. Try to be more short in introductive remarks, these concepts are enough when explained once, you do it too frequently through the text. R82-86. In abstract too

Geol Setting..... R99_ sandwiching ! R105_Individualizing?

Data & meth R122 – 129. The integrated work you describe is a standard one. You should underline more which are the new data you are presenting in this paper for the first time. Making use of GIS or looking at log and seismic doesn't represent itself something that is worth of note. R138 – 157. Describing data you analyze and methods you make reference to one table and Figs 3, 4, 5,6, 9,10. The reader would be supposed to jump ahed and backward to text to try and figure where these data

are. . .. . .It would be possible to limitate this to wells, Outcrops, seismic, geol profiles and eventually have all of these on one or two Figures only? R147_ supplementary data? What are they and where I found it in the paper?

Structural framework 170_ Same is stated in the introduction. . .no need to repeat 4.1 & 4.2 Syn-sedimentary Same comment for these two and is relevant to Figures citation. e.g. R187, R204, 231,242 To figure out one simple geological concept that comes from an original observation made by you the reader is supposed to jump omong 3 or more Figures, looking for, with a lens because is almost always un-readable, a fault F2. . .. . .a label DO1,DO2. . ...a part of Fig 5 that is 5AA' (R189) or 5A-A' (182). The reader is lost with all these citation. In fig 5 you have one map and one section, which is the need to label it differently if they are univocally designed by a 5A? Try if possible to simplify these two paragraphs because they are currently impossible to follow. Minimize the reference to the figures that display a concept, you do not need to cite all, just the more significative.

5.1 Facies association. . .. . . R268. Explain how the present study add knowledge to what stated above in definining the facies associations. Which are the new data? Which is the news with respect to the works cited? R454 6. An association. . ...refer the title to an observation not to a conclusion. . ..eg: subsidence and tectonic history R532. Same as above R546. . .list of thermos orogenic events is complex, brackets inside brackets, cite name or age

Figures Fig.1 . Too full. Very difficult to read. I would simplify the Paleozoic series legend it is impossible to identify on Map different grades of colours within Cambrian or Carboniferous. Too much writings in the AOI (fig3A) too small figure and too dense posting of the Geochronology data to appreciate their relationships with terrains Fig.3. W=well and O=Outcrops not in the legend. Reference to location of sections Fig 5 & 6 difficult to read in Map. Capital letters of arches and basins confusing with letters that make reference to following figures. Fig.4. I do not understand which is the extent of this area with respect to the previous Map (Fig.3A). The small writings (eg: Otj. . .. . .)

are completely unreadable, remove it or enlarge. Fig.5. Same comment made above for the small writings in Figures. I do not understand the need to differentiate map and section with A, A' Fig.7. symbols on core description section are too small to be understand. Fig11. K cited in legend but not in Figure Fig.12. A,B & C are too small, it should be enlarged. D &E are necessary?

---

## Author Comment (AC1) · 26 Sep 2018

Dear M. Zazoun,

Thank you for your work and patience, we have tried to respond to all your comments. Your remarks have contributed to a better constrained of the Palaeozoic stratigraphy of the Saharan platform and to added new recent publication. Please find our answers marked in red. The description of necessary changes in the revised manuscript are attached to this reply.

Sincerely,

Paul Perron, Michel Guiraud, Emmanuelle Vennin, Eric Portier, Isabelle Moretti, Moussa Konaté

Color legend in attached manuscript:

 + Green = moved sentences

 = deleted sentences

Red = added sentences

**#Referee comments (RCs)**
**I-Text**
**1- Lines 100-101; 106 to 110; 113; 115 to 116; 118 to 120…..:** Put references from oldest to most recent.
-"We are using the publication style of Solid Earth where references are classified alphabetically."

**2- Line 253:** is composed of fluviatile Cambrian. The whole sedimentary series described in the literature is composed of fluviatile to Braid-deltaic plain Cambrian, not only fluviatile (eg. Brahmaputra River analogue), with a transitional facies from continental to shallow marine (Sabaou et al., 2009, p160).
-"We have modified and added reference (line 308-311)."

**3- Line 254:**….. glacial Ordovician. Upper Ordovician glaciogenic deposits….
-"We have modified (line 311)."

**4- Line 255:**….. argillaceous deep marine Silurian…argillaceous deep marine Silurian deposits….
-"We have modified (line 313)."

**5- Line 255:** Missing the last reference for the Silurian (Djouder et al., 2018).
Djouder, H., Lüning,S ., Da Siva, A-C., Abdellah, H., Boulvain, F. (2018), Silurian deltaic progradation, Tassili n'Ajjer plateau, south-eastern Algeria: Sedimentology, ichnology and sequence stratigraphy. Journal of African Earth Sciences, Volume 142, June 2018, Pages 170-192
-"We have added reference (line 313-314)."

**6- Line 266:** the term Algerian platform don't exit, you speak probably about the Algerian part of Saharan platform
-"We have modified (line 324)."

**7- Line 300 and Figure 2**: In the Illizi basin, these facies are mainly recorded in the Cambrian Ajjers Formation

In the Illizi Basin, and Saharan platform, the Ajjers Formation and/or the equivalent formation are dated: Upper Cambrian? to Ordovician (Tremadoc to Caradoc)(Fabre, 2005 ; Vecoli, 2000 ; Vecoli & Playford, 1997; Vecoli et al., 1995 ;Vecoli et al .,1999).

-"We have modified and added references (line 358-360)."

**References for the datation:**

Vecoli, M., 2000. Palaeoenvironmental interpretation of microphytoplancton diversity trends in the Cambrian–Ordovician of the northern Saharan Platform. Palaeogeography, Palaeoclimatology, Palaeoecology 160, 329–346.

Vecoli, M., Playford, G., 1997. Stratigraphically significant acritarchs in uppermost Cambrian to basal Ordovician strata of Northwest Algeria. Grana 36, 17–28.

Vecoli, M., Albani, R., Ghomari, A., Massa, D., Tongiorgi, M., 1995. Précisions sur la limite Cambrien–Ordovicien au Sahara Algérien (secteur de Hassi-R'mel). Comptes Rendus de l'Académie des Sciences, Paris 320 (IIa), 515–522.

Vecoli, M., Tongiorgi, M., Abdesselam-Rouighi, F.F., Benzarti, R., Massa, D., 1999. Palynostratigraphy of Upper Cambrian–Upper Ordovician intracratonic clastic sequences, North Africa. Bollettino della Società Paleontologica Italiana 38 (2–3), 331–341.

Vecoli, M., Videt, B., Paris, F., 2008. First biostratigraphic (palynological) dating of Middle and Late Cambrian strata in the subsurface of northwestern Algeria, North Africa: implications for regional stratigraphy. Review of Palaeobotany and Palynology 149 (1–2), 57–62.

**8-** There was confusion about the subdivision used. Often, it is extracted from English papers. It is best to keep the original nomenclature in French, as it was established by geologists (BRPA, 1964; in Beuf et al., 1971), in fact: (Beuf et al., 1971, Tab-1, p.158: Bennacef et al., 1971; Fabre, 2005).

-"We have modified in the text and in Fig. 2 keeping the original French nomenclature (see Fig. Fig. 3 and line 360-361, 395-396, 409, 433-434, 442, 470-473, 493, 518)."

From de base to the top, we observe:
A- The Ajjers Formation

It's divided from the bottom to the top:

In the Tassilis-N-Ajjers (Outcrops)
- Les Grès/Conglomérats d' El Moungar (Unit I)
- Les Grès de Tin Taradjelli (Unit II)
- La Vire du Mouflon (Unit III-1)
- La Banquette (Unit III-2)

In the Illizi basin (Subsurface equivalents)

- (Unit II) : R3+R2+Ra +Ri

- (Unit III-1) : La Zone des Alternances + Les Argiles d'El Gassi + les Grès d'El Atchane

- (Unit III-2) : Les Quartzites de Hamra

B- The In Tahouite Formation (Unité III-3) (outcrops)
(Unit III-3) : Les Grès de Ouargla + Argiles d'Azzel + Grès de Oued Saret (subsurface)

C- The Tamadjert Formation (Unit IV) (outcrops)

Ajjers Formation + In Tahouite Formation represent the Sequence 1 (Preglacial-deposits)

Tamadjert Formation (Outcrops) and the equivalent in the subsurface (Unit-IV) is the sequence 2 (syn-glacial deposits)

Also for The Siluro-Devonian, keep the original nomenclature in French (Text and Fig.2):
Tigillites Talus: Talus à Tigilites (Lines 348 ; 371)
-"We have modified in the text and in Fig. 2 (see above)."

Passage Zone: Zone de passage (Lines 379, 427)
-"We have modified in the text and in Fig. 2 (see above)."

Middle Bar: Barre Moyenne (Lines 301, 335)
-"We have modified in the text and in Fig. 2 (see above)."

Sidewalks: Trottoirs
-"We have modified in the text and in Fig. 2 (see above)."

Upper Bar: Barre Supérieure (Lines 301, 335)…..etc.
-"We have modified in the text and in Fig. 2 (see above)."

**References for The Ajjers Formation:**

Bennacef, A., Beuf, S., Biju-Duval, B., De Charpal, O., Gariel, O., Rognon, P., 1971. Example of cratonic sedimentation: Lower Palaezoic of Algerian Sahara. The American Association of Petroleum Geologists Bulletin 55 (12), 2225–2245.

Beuf, S., Biju-Duval, B:, de Charpal, D., Rognon, R., Bennacef,A., 1971. Les gres du Paleozoïique inferieur au Sahara.Sedimentation et discontinuite: evolution structurale d'un craton. Institut Francais Petrole, Collection Sciences Techniques P&role 18, 464~.

Sabaou, N., Aït Salem, H., Zazoun, R.S., 2009. Chemostratigraphy, tectonic setting and provenance of the Cambro-Ordovician clastic deposits of the subsurface Algerian Sahara. Journal of African Earth Sciences 55 (3–4), 158–174.

Zazoun, R.S., Mahdjoub, Y., 2011. Strain analysis of Late Ordovician tectonic events in the In-Tahouite and Tamadjert Formations (Tassili-n-Ajjers area, Algeria). J. Afr. Earth 1266 Sci. 60, 63–78.

**9- Line 489:** as Hirnantian glacial valleys ….as Hirnantian glacial-Palæovalleys

-"We have modified (line 562-563)."

**10- Line 534:** Hoggar shield ? sometimes, you speak about Tuareg shield (TS)…Hoggar is the massif and the Tuareg is the shield.

-"We have modified (line 200)."

**11- Line 526 :** Missing reference: Hercynian Tectonic Event

According                                                        to                                                        Zazoun (2001)……………………………………………………………………………………………Th e basement fabric features exerted a very strong control on the structural evolution during the Hercynian deformation (See also Haddoum et al., 2001).

-"We have modified and added references (line 613-615)."

**References for The Hercynian Tectonic Event**

Haddoum, H., Guiraud, R., Moussine-Pouchkine, A., 2001. Hercynian compressional deformations of the Ahnet-Mouydir Basin, Algerian Saharan platform: far-field stress effects of the Late Paleozoic orogeny. Terra Nova 13, 220–226.

Zazoun, R, S. 2001. La tectogenèse hercynienne dans la partie occidentale du bassin de I'Ahnet et la région de Bled El-Mass, Sahara Algérien: un continuum de déformation Journal of African Earth Sciences Vo.32, N°4, 869-887.

**12- Lines 671:** Abdesselam-Roughi, F ; Abdesselam-Rouighi, F.F

-"We have modified."

**13- Line 902:** Fabre, J. 2005. Géologie du sahara occidental et central. Musée Royal de l'Afrique Centrale-Belgique.Tervuren African Geoscience Collection Vol.108. ISBN 90-75894-66-x ; ISSN : 1780-8551, 610 p.

-"We have modified."

**14- Line 980:** Hassan Kermandji, A.M or Kermandji, A.M.H., ?

-"We have modified."

**15- For the terranes notion please see the last publication about this topic :**
Sonia Brahimi, Jean-Paul Liégeois, Jean-François Ghienne, Marc Munschy, Amar Bourmatte. 2018. The Tuareg shield terranes revisited and extended towards the northern Gondwana margin: Magnetic and gravimetric constraints. Earth-Science Reviews, doi:10.1016/j.earscirev.2018.07.002 (Accepted manuscript)

-"It is a very interesting publication sent at the same time, globally coherent with our observations. We have added to our citations (line 649-650)."

**16- Lines 64 and 1313**
Missing : Intra-Arenig unconformity cited by (Fabre 2005, pp 169) in the North Africa, Eschard et al., (2010) and Kracha thesis, (2011) in the Ahnet Basin and Beuf et al., (1968-1971) in the Tassilli-N-Ajjers and Eschard et al. (2006) in the Berkine Basin.

*"Erosion occurred in many places during an intra-Arenig unconformity" (Eschard et al., 2010)*
*" Une discordance de faible amplitude, mais dont l'extension intéressera tout le Maghreb depuis l'Anti-Atlas occidental jusqu'à la Libye marque la fin...de ce premier cycle Ordovicien » c'est-à-dire Tremadoc et Arenig inférieur....(Fabre, 2005)*

*« De même, les « Argiles de Tiferouine » transgressent progressivement sur la série de Bled El Mass, et sont limitées à la base par une discordance angulaire conglomératique, qui pourrait bien correspondre à la discordance Intra-Arénig, mise en évidence dans de nombreuses régions sahariennes (Beuf et al., 1968; 1971; Eschard et al., 2006)........... » in Kracha., 2011, pages 73-86-100*
*« Le passage des « Quartzites de Hamra » vers les « Grès de Ouargla » se fait de manière progressive, et l'on assiste par la suite à la troncature des termes sommitaux de cette formation sous la discordance Intra-Arénig qui annonce l'arrivée brutale des « Argiles d'Azzel-Tiferouine ». Les « Grès d'Oued Saret » sont peu préservés sous les ravinements glaciaires de l'Unité IV (et/ou la discordance Taconique) (Kracha, 2011)........ Pages 113 et 121.*

-"We have added these references and modified Fig. 2 integrating this unconformity (see Fig. 3 and line 76)"

**References for the Intra-Arenig unconformity**

Beuf. S., Biju-Duval. B., Mauvier, A., Legrand Ph. (1968). Nouvelles observations sur le "Cambro-Ordovicien" du Bled El Mass (Sahara central), Publ. Serv. Géol. Algérie, Bulletin n° 38, p.39-51.

Beuf. S., Biju-Duval. B., De Charpal. O., Rognon. P., Gariel. O., et Bennacef, A. (1971). Les grès du Palaeozoique inferieur du Sahara. Publication Institute Français du Petroleum, Collection Science et Technologie du Petrole, Paris, No. 18. 484 p.

Eschard.R, Hellat.C, Malla.M, Bénamane.K, Betioui.H, Callot.J.P, Carpentier.B, Chelcheb.S, Couprie.E, Dahi.M, Delmarre.S, Desaubliaux.G, Deschamps.G, Euzen.T, Hachemi.L, Hannoun.R, Jacolin.J.E, Lassal.A, Leblond.C, Levêque.I, Lorant.F, Mokhtari.N, Lorin.T, Rabary.G, Rudkiewicz.J.L, Wattine.A. (2006). Berkine Gas project. Evaluation of the Gas potential in the Berkine basin (Algeria). Rap. Conf. Ifp.

Eschard. R, Braik.F, Bekkouche.D, Ben Rahuma.M, Desaubliaux.G Deschamps.R et Proust. J. N. (2010). Paleohighs: their influence on the North African Paleozoic petroleum systems. Geological Magazine, 147 (1), 28-41.

Kracha, N. 2011. Relations entre sédimentologie, fracturation naturelle et diagenèse d'un réservoir à faible perméabilité : application aux réservoirs de l'Ordovicien, Bassin de l'Ahnet, Sahara Central, Algérie. Thèse de Doctorat, Université des sciences et Technologies de Lille. 458p, Thèse non publiée.

**17-** It would be desirable to speak also about the work of Boote et al (1998). These authors define the notion of ''The Gondwana Super-cycle" between the infra-Cambrian (Panafrican) Unconformity and the Hercynian Unconformity. This Super-cycle has been divided in 02 Super-cycles: a Lower Gondwana Super-cycle and an Upper Gondwana Super-cycle. This nomenclature which is defined at the scale of north Africa also applies in South America (Souza Cruz et al., 2000) and Saudi Arabia (Sharland et al., 2000, Davidson et al. 2001), even if a slight diachrony may exist between these areas.

1- Lower Gondwana Super-cycle (Infra-Cambrian Unconformity to Caledonian Unconformity): is a major second order transgressive-regressive megacycle, in the sens of Vail et al. ( 1977)

2- Upper Gondwana Super-cycle (from the Caledonian Unconformity to the Hercynian Unconformity)

(in: The Lower Paleozoic sedimentation of the Algerian Saharan craton, Tassili N'Ajjer oucrops. Field Guide Book, February 20-25, 2003. AAPG symposium, Algiers 18-20 February 2003, 103p).

-"We have chosen to present in Fig. 2 column (7) the 2nd order transgressive-regressive cycle highlighted by Carr, (2002), and Eschard et al., (2005) (see Fig. 3)."

**References for "The Gondwana Super-cycle"**

Davidson, L., Bestwetherick, S., Craig, J., Eales, M., Fisher,A., Himmmali, A., Jho, J., Mejrab, B. and Smart, J. 2000. The structure, stratigraphy and petroleum geology of the Murzuk basin, southwest Libya. In Geological exploration in Murzuq basin, (in: Eds M.A. Sola and D.Worsley), Elsevier, 295-320.

Sharland, P.R., Archer, R., Casey, D.M., Davies, R.B., Hall, S.H., Heward, A.P.,Horbury, A.D. and Simmons, M.D., 2001. Arabian Plate Sequence Stratigraphy. Geoarabia,Special Publications, 2, 371p, 2 plates.

Souza Cruz, C.E., Miranda, A. P., and Oller, J. 2000. Facies analysis and depositional systems of Late Silurian-Devonian subandean basin, southern Bolivia and Northern Argentina. In: Memorias del Congreso Geologico Boliviano, La Paz, 14-18 Noviembre 2000, 85-90.

**18- Thermal History of Ahnet and Sbaa Basins (Histoire Thermique)**
**(See Akkouche Thesis) : you will find below the conclusions of Mr. Akkouche about the Fission track study of Ahnet and Sbaâ**

Akkouche, M. 2007, Application de la datation par traces de fission à l'analyse de la thermicité de bassins à potentialités pétrolières. Exemple de la cuvette de Sbaâ et du bassin de l'Ahnet-Nord (plate-forme saharienne occidentale, Algérie). Thèse de Doctorat, Université de Bordeaux 1, 297p.

You can download the thesis at the link below : www.iaea.org/inis/collection/NCLCollectionStore/_Public/41/021/41021844.pdf

**Histoire Thermique du bassin de l'Ahnet**
Les traces de fission témoignent d'épisodes thermiques et d'érosion différents suivant les domaines du bassin avec une tendance générale à l'effacement des traces du Nord vers le Sud. Au Nord du bassin de l'Ahnet (MRS-1 et MSL-1), les âges obtenus dans les formations de l'Ordovicien et de la base du Silurien sont de 50 Ma environ. Ils témoignent d'une phase thermique post-varisque qui a affecté la colonne paléozoïque. On peut estimer que les températures atteintes au cours de cet épisode thermique, probablement >100°C, sont également à mettre en relation avec l'épisode thermique d'extension régionale d'âge triasico-jurassique. Cet événement thermique pourrait également être à l'origine de l'effacement total des traces pré-existantes du niveau dévonien du forage MSR-1 qui présente, à 505 m sous la discordance varisque, un âge TF de 100 Ma. Plus au Sud, les âges respectifs de 37 Ma et 26 Ma obtenus à

des profondeurs de 1030 m (forage MKRN-1) et 1532 m (forage BH-5), suggèrent une altération thermique cénozoïque des traces de fission plus intense que celle mise en évidence dans le Nord du bassin. Cette caractéristique pourrait refléter une érosion récente plus importante, mais également l'action éventuelle de gradients géothermiques plus élevés

**Histoire Thermique de la cuvette de Sbaâ basin (See Akkouche Thesis)**

La plupart des résultats obtenus sur les échantillons prélevés dans les forages de la cuvette de Sbaâ est compatible avec ceux obtenus par les chercheurs de la Société GEOTRACK sur le forage OTRA-1, situé à 50 kilomètres du forage ODZ 1-bis. Dans le forage ODZ-1bis, les résultats suggèrent une histoire thermique marquée par trois événements de refroidissement. Le plus ancien se situe entre 200 et 300 Ma. Le second aurait un âge proche de 170-210 Ma et le plus récent entre 30 et 50 Ma. Ces âges témoigneraient successivement des événements de refroidissement carbonifère (fin de l'orogenèse varisque), jurassique (détumescence thermique post-rifting atlantique) et éocène (bombement du Hoggar). Un seul échantillon présente un résultat surprenant. Il s'agit de l'échantillon n°644 (forage ODZ-1bis) qui donne un âge de 472±11 Ma, prélevé dans un horizon détritique du Tournaisien. Onze des 12 apatites de cet échantillon présentent des traces de fission fossiles pré-datant nettement le Carbonifère. En d'autres termes, l'épisode varisque n'a jamais atteint un seuil thermique suffisant pour effacer ces traces au niveau de la bordure septentrionale de la cuvette de Sbaâ. Ce constat suggère que la cuvette de Sbaâ est un des domaines de la plate forme saharienne occidentale les moins affectées par les événements thermiques post-varisques. De toute évidence, à partir de la fin du Carbonifère, voire du Permien, la cuvette de Sbaâ est structurée et demeurera dans un climat structural relativement superficiel.

-"We are aware of post-Hercynian tectono-thermal events, however it wasn't the object of our study. We have focused the paper on the Palaeozoic."

**II-Figures**

**1- Figure 2 :**

- Column 4 : Mouydir not Mouyrdir -"corrected"
- Column 6 (Tassili) : In Tahouite Formation is the Unit III-3, not The Unit III-2 -"corrected"
Tin Taradjelli Sandstones is the Unit II -"added"
- Column 6: Orsine Formation not Orsine -"corrected"
Tin Meras Formation not Tin Meras -"corrected"
Illrene Formation not Illrene -"corrected"
El Moungar Conglomerat is the Unit I -"added"
- The intra-Arenig unconformity is missing
-"see Fig. 3."

**III- Conclusions : (See Referee comments above)**

- Several references have been omitted (Ajjers Formation, Silurian, Hercynian tectonic Event, Terranes…): see bibliography attached for each remark.
- Very minor corrections are required.
- The intra-Arenig unconformity was not mentioned.
- Figure 2 is to be corrected.
- Put references from oldest to most recent.
- Some typographic mistakes about the names of the authors.
- Ovoid the translation of the nomenclature in English language.

---

## Author Comment (AC3) · 26 Sep 2018

Dear M. Peace,

Thank you for your work and patience, we have tried to respond to all your comments. Your remarks have contributed to a better organisation of the paper. Please find our answers marked in red. The description of necessary changes in the revised manuscript are attached to this reply.

Sincerely,

Paul Perron, Michel Guiraud, Emmanuelle Vennin, Eric Portier, Isabelle Moretti, Moussa Konaté

Color legend in attached manuscript:

 + Green = moved sentences

 = deleted sentences

Red = added sentences

Perron et al. describe a study into the role of inherited structures in intracratonic basins in the Central Sarara using a combination of seismic interpretation, various GIS analyses, stratigraphy and geochronology. In my opinion the background information is adequate (just requiring a few minor amendments), the analyses appear to have been appropriately conducted and the discussion and conclusions appear to suitably draw upon the results. In addition, the figures are generally of high quality and should be commended. However, at times some sections seem longer than required and there is potentially some repetition of information. The study is of current relevance with a number of recent papers also addressing similar topics, including a paper in Solid Earth (Phillips et al., 2018). Thus, Solid Earth seems like a suitable place for publication.

I would therefore like to recommend publication if the relatively minor points suggested here and in the other comments are appropriately addressed. These minor points should not be too arduous, but I think that they will improve the manuscript. I hope you find these suggestions useful and I look forward to seeing a final version of the paper. If you would like clarification of any of my suggestions feel free to contact me.

First, I felt that it was not clear from the abstract what the purpose, aims and main findings of the study were. The reason for this appears to be that much of the abstract (and introduction to some extent) is devolved to regional background information. Although such information is obviously important I suggest more clearly outlining the main findings and study aims in the abstract. In addition, I think the introduction would benefit from a short but general overview of structural inheritance, including information from geological settings from beyond the present study to demonstrate the significance of such processes. Moreover, particularly in the abstract but also in the introduction, it is not always clear what is a finding of this study and what information is from previous work. I suggest Perron et al. clarify these sections with this in mind. Related to the previous point is that the introduction contains a lot of material that would likely be better placed in the subsequent dedicated 'Geological setting' section. An example of this is the material in lines 59-75. I think that moving such information into the geological setting and using the introduction to better set up the aims and rationale would be better.

In addition to the previous points I think that the 'Data and methods' section requires additional information to be of more use to readers. For example 'Geographic Information System analysis (GIS)' (Line 124) is very ambiguous as this could mean any one of a number of approaches.

Also minimal details are provided regarding the seismic data or the methods deploying in its interpretation. I suggest that Perron et al. consider adding additional technical information regarding their data sets and the analyses used. Some of these details are provided later in various sections but I think they would be better placed here in the dedicated section.

The descriptions of results in sections 4-6 are generally very good. In particular, good use of the figures is made in the text. However, I found that in section 7 there is an abundance of material that might be better placed in the 'Geological Setting' or the 'Data and Methods' sections. Some examples of this are noted below but I suggest Perron et al. reconsider the location of some of the material in this section.

Other minor points include:
-Line 32 – This line is quite awkward. Suggest rewording.
-"We have reformulated the abstract."

-Line 35 – Replace 'activated' with 'reactivated'?
-"We have modified (line 39)."

-Lines 59-60 – I don't quite understand this sentence.
-"We wanted to explain that features of worldwide intracratonic basin are identified in the Sharan Platform."

-Line 89 – Figure 3 appears to be called before figure 2.
-"We have changed the order of figures (see Figs. 2 and 3)."

-Lines 89-93 – This opening paragraph of the geological setting feels like it needs references.
-"This is basic geographic localisation."

-Lines 130-132 – This reads more like results. Consider moving it.
-"Well-exposed area is the reason why we choose to use satellite images for tectonic interpretation."

-Line 145 – Suggest providing more details of the seismic data.
-"What kind?"

-Line 171 – 'oval' has been mentioned quite a few times before this. Is it really necessary to mention it this often?
-"We have deleted repetition (line 106)."

-Line 258 – Suggest clarifying why the Devonian deposits are sensitive to such processes.

-Lines 534-538 – This paragraph reads more like geological setting.
-"We have placed it in geological settings paragraph (line 121-127)."

-Lines 539-549 – This description of the analyses would be better placed in the 'Data and methods' section.
-"We have placed it in data and methods paragraph (line 200-215)."

-Lines 551-554 – The geological setting section might be more appropriate for this information.

-"We have placed it in data and methods paragraph (line 121-127)."

---

## Author Comment (AC4) · 26 Sep 2018

Please find attached a tracked changes document of the manuscript, detailing the changes made in response to the reviewers comments. Individual replies to comments and reviewers are made in response to each comment.

Please also note the supplement to this comment:
https://www.solid-earth-discuss.net/se-2018-50/se-2018-50-AC4-supplement.pdf

---

## Author Comment (AC5) · 26 Sep 2018

Dear M. Wendt,

Thank you for your work and patience, we have tried to respond to all your comments. They contribute to improve the paper. Please find our answers marked in red. We have tried to explain the better possible the interrogation on the tectonic part and added comments on the sedimentologic part. The description of necessary changes in the revised manuscript are attached to this reply.

Sincerely,

Paul Perron, Michel Guiraud, Emmanuelle Vennin, Eric Portier, Isabelle Moretti, Moussa Konaté

Color legend in attached manuscript:

 + Green = moved sentences

 = deleted sentences

Red = added sentences

**Preliminary remarks:**

Though I have been working on the Paleozoic of the Algerian Sahara for many years (1987-2006) I am only familiar with the Devonian and Carboniferous, but not with the older formations and the crystalline basement. Therefore, I can only judge these aspects of the above manuscript. Likewise, I feel not competent enough to consider some tectonic reconstructions. I hope that the other reviewer(s) are able to review these as-pects of the manuscript with a better competence.

The manuscript is an overview of the bio- and lithostratigraphic, sedimentologic, paleogeographic and paleotectonic evolution of the Ahnet-Mouydir area in southern Algeria based on field data from previous authors, well log analysis, satellite images and geophysical data. As such it is a good summary of the evolution of a marginal basin-and ridge system which farther north in central Algeria has yielded enormous oil and gas reservoirs.

**Detailed critical remarks:**

Title: The research areas covers a much larger area (including also the Reggane, Basin, Illizi Basin, Hoggar Shield) than expressed in the title. This should be made clear in the title.

-"We have added this remarks to the title even if the study is essentially focusing the Ahnet-Mouydir basins (line 2)."

Line 20: Pan-African orogeny. Strictly spoken this was around 600 MA, but including earlier phases it was 900-520 MA. What do you mean exactly?

- "We have corrected (line 46). Indeed, the Pan-African orogeny result from the accretion, then collision of different terranes during different phases. This polyphased event has constrained the structural framework of the Saharan Platform."

Line 35: "Devonian compression". I consider this as a mere speculation. According to all previously gathered data the Devonian was a period of tectonic quiescence accompanied by slight extension.

-"We refer to the (a) Siluro-Devonian (also called Caledonian) and the (b) Mid to Late Devonian events. We bring in our paper new evidences in favour of these tectonic events through seismic lines (Fig. 7) and satellite images (Fig. 6C, D, E, F). Besides, they are already mentioned in the literature:

(a) In the Saharan platform, the Caledonian tectonic event, is mainly mentioned as uplifting of some trends, large-scale folding or blocktilting (e.g. Gargaff arch, Tihemboka arch, Ahara high, Amguid El Biod), associated with breaks in the series and frequent angular unconformities below Early Devonian formations (Beuf et al., 1971; Boote et al., 1998; Boudjema, 1987; Carruba et al., 2014; Coward and Ries, 2003; Echikh, 1998; Eschard et al., 2010; Frizon de Lamotte et al., 2013; Ghienne et al., 2013; Gindre et al., 2012; Legrand, 1967b, 1967a). During this compressive event, large wavelength folds and paleohighs were accentuated, affecting sedimentation and facies distribution in the sedimentary basins (Eschard et al., 2010; Galeazzi et al., 2010). Locally, paleohighs may have provided detrital material (Eschard et al., 2010; Galeazzi et al., 2010). Evidence of the Caledonian event is documented, in the southwestern and southern flank of the Ghadames Basin, the Lower Devonian Tadrart formation is seem to directly overly the Upper Silurian basal Acacus series with a progressive truncation of the Acacus (Upper Silurian) units from NE to SW on this unconformity (Echikh, 1998). In the Illizi basin, only the lowermost part of Acacus Formation is preserved (Echikh, 1998). Besides, seismic data may show folding of the Silurian section below flat-lying Devonian deposits (Echikh, 1998). Well described indication of Caledonian unconformity are also highlighted in the Murzuq basin (Ghienne et al., 2013) and Al Kufrah basin (Gindre et al., 2012). Massive sand injection associated with igneous intrusion triggered by basin-scale uplift are also described in the Murzuq basin (Moreau et al., 2012). These structural features imply NW-SE shortening, probably of moderate intensity, though much weaker than the Hercynian one (Guiraud et al., 2005). Elsewhere, in the Drâa basin, in the NW Libya and over the Al Kabir trend, there is also no sign of this event in Lower Devonian series (Echikh, 1998; Ouanaimi and Lazreq, 2008).

Moreover, a widespread near top Emsian unconformity probably triggered by regional tecontic activity has been identified in the Illizi basin (Abdesselam-Rouighi, 2003; Boudjema, 1987; Boumendjel et al., 1988; Brice and Latrèche, 1998; Moreau-Benoit et al., 1993), in the Ahnet-Mouydir basin (Wendt et al., 2006), in the Libyan Ghadames and Al Kufra basins (Bellini and Massa, 1980). It is associated to basaltic volcanism and intrusive activity in the Ahnet basin (?) and Anti-Atlas (Belka, 1998; Wendt et al., 1997)

Many authors have correlated the Late Silurian to Early Devonian tectonism as the maximum collisional deformation of the Caledonian Orogeny (see references below). However, this event clearly relates to collisions involving far away continents and terranes where Gondwana was located thousands of kilometres to the south and separated from the collisional zone by a major ocean during this time (Craig et al., 2006; Mckerrow et al., 2000; Stampfli and Borel, 2002). Tectonic events in North Africa during post-Infracambrian-pre-Hercynian times were therefore independent of the Caledonian Orogeny. Time-descriptive terms may be preferred instead (Craig et al., 2006). This denomination is thus controversial. The origin of this intra-plate stress could be linked to far field stresses, knowing that, in continental craton compression stresses can be transmitted through distances of up to 1600 km from a collision front (Ziegler et al., 1995). The origin of Late Silurian to Early Devonian intra-plate stress in North Africa is currently unclear but is possibly associated either with a phase of rifting along the Gondwana margin (Boote et al., 1998) or with initial closure of the Iapetus Ocean (Fekirine and Abdallah,

1998). Frizon de Lamotte et al., 2013 didn't interpreted it as a far effect of the Variscan orogeny, contrary to Fabre, 2005 who associated to the beginning of it.

(b) The Middle to Late Devonian is the time for two contrasting large-scale tectonic processes: the onset of the Variscan Orogeny along the Gondwana-Laurussia margin on the one hand and the development of magmatism, rifting and domal basement uplift within these continents on the other hand (Frizon de Lamotte et al., 2013). The collision between Gondwana and Laurasia that ultimately produced the Hercynian Orogeny possibly first affected North Africa during the mid-Devonian, creating extension/transtension pull-appart basins (Craig et al., 2006). This Devonian deformation has reactivated megashear zone systems coeval with semi-regional uplift of the Ghadames and Illizi basins and of the adjacent Tihemboka, Ahara, Gargaf and Brak-Bin Ghanimah arches in the mid-Eifelian and at the end of the mid-Devonian (Late Givetian) and with the related development of the Frasnian Unconformity (Craig et al., 2006). Evidence of extensional structures and/or tectonic activity during the Late Devonian, as proved by the major thickness variations of these series are documented in the Anti-Atlas (Baidder et al., 2008; Michard et al., 2008; Wendt, 1985), in the northern Africa and Arabia platform (Frizon de Lamotte et al., 2013) and in the Ahnet basin (Wendt et al., 2006). This event corresponds to a major collapse and even "disintegration" of the north-western Gondwana margin prior to the Variscan Orogeny (Wendt, 1985). While, the activity of the palaeohighs (e.g. Ahara, Gargaff and Tihemboka High) almost ceased during the Frasnian times, with marine shales onlapping different elements of the Palaeozoic succession below and sealing most of the palaeohighs (Eschard et al., 2010)."

Line 52: 7 m/MA. Give reference.
-"We have calculated it: 200m/300Ma=~7m/Ma (cf. figure below). It is not very precise but it is in the order of magnitude. Holt et al., (2010) indicate 22.2m/Myr (Ghadames) and 10.1 m/Myr (Al Kufrah) for the highest rates. Sloss, (1988) show 20-30 m/Myr to 3-4 m/Myr values."

[Figure]

1 HAD-1 Ghadames Basin (Makhous and Galushkin, 2003)
2 RPL-101 Reggane Basin (Makhous and Galushkin, 2003)
3 L1-1 Murzuq Basin (Galushkin and Eloghbi, 2014)
4 TGE-1 Ilizi Basin (Makhous and Galushkin, 2003)
5 REG-1 Timimoun Basin (Makhous and Galushkin, 2003)
6 Ghadamès/Berkine (Yahi et al., ?)

Line 61: 16 million km2. Impossible! The entire Sahara occupies about 9 million km2.
-"We have modified this value (line 73)."

Line 121 ff. and 133: It is not clear if the authors have ever been in the field; equivalent data seem to be based on previous published sources only. This should be made clear unequivocally.

-"This study is written in the frame of a phD and there wasn't fieldtrip during this time. However, some of the authors have been on the field and have many years of experience of the area throughout the oil industry (NEPTUNE former ENGIE/GDFsuez) or throughout academic. We have better specified the new data (especially satellite images, seismic lines and well-logs) which have been used in this study. We have integrated a figure presenting the method and original work (cf. fig. 4 in manuscript in supply)."

Line 141: Please separate both calibration of well-logs by palynomorphs (which are poorly reliable biostratigraphic markers) and field sections by conodonts (which give by far the best time resolution), goniatites and brachiopods. Both biostratigraphic subdivisions can be only roughly be correlated.

-"We have modified and we are more careful with the data set (line 168-174). Indeed, we aware about the poor resolution of palynomorphs calibration. Unfortunately, it is the only data available in wells."

Line 144 (and later): "Synsedimentary extensional and compressional markers": This means during the Devonian and Carboniferous. On which evidence these important tectonic events are based? Apparently not on field data. During about 9 months of personal field work I followed typical marker levels (e.g. the upper Eifelian/Givetian limestone ridge) for tens of kilometers (walking from ridge into basin deposits), but I have never seen something like that. The observation of doubtless Hercynian faults does not automatically allow the conclusion that they are rejuvenated earlier structures.

-"In our area, evidences of tectono-sedimentary structures (i.e. thickness variations, lateral facies variations, current directions variations) showing activation and reactivations of arches were already highlighted in the literature by field studies:

-In the Arak-Foum Belrem arch see (a), (b) and (c) below from (Beuf et al., 1971, 1968b) during the Ajjers deposition (i.e. Upper Cambrian Lower Ordovician).
-In the Bled El mass area of the Azzel Matti arch see e below from (Beuf et al., 1968a; Eschard et al., 2010) during the Ajjers deposition (i.e. Upper Cambrian Lower Ordovician).
-In the Tanezrouft area of the Azzel Matti arch see d below from (Beuf et al., 1971) during the Ajjers deposition (i.e. Upper Cambrian Lower Ordovician).

Then, during the Caledonian (i.e. Siluro Devonian), these structures are also known in the field:

-In the Assedjrad area on the Azzel Matti arch see (h) below from (Beuf et al., 1971).
-In the Arak-Foum Belrem arch see (g) below from (Legrand, 1967a), see (h) below from (Beuf et al., 1971) and see also in (Biju-Duval et al., 1968).
-In the Idjerane axis see (f) below from (Legrand, 1967a).

These latter are evidences of the early activity of the arches leading to the individualization of the different basins since the Cambro-Ordovician time. Then, the arches were reactivated during the Caledonian (i.e. Siluro-Devonian). Here, we don't cite other arches (i.e. Tihemboka, Ahara, Amguid El Biod, Gargaf, Dor El Gussa-Murizidié…) of the Saharan Platform where these syn-sedimentary structures are also described (Borocco and Nyssen, 1959; Carruba et al., 2014; Chaumeau et al., 1961; Chavand and Claracq, 1960; Collomb, 1962; Dubois and Mazelet, 1964; Eschard et al., 2010; Fabre, 2005; Frizon de Lamotte et al., 2013; Ghienne et al., 2013; Massa, 1988)".

From this literature, we bring new evidences of syn-sedimentary tectonic reactivating successively arches structures during Paleozoic by using new unpublished data (from 3D Google Earth satellite images and seismic lines).

[Figure]

(a) Variation of Ajjers series on the Arak-Foum Belrem Arch (Beuf et al., 1968b).

[Figure]

(b) Syn-tetconic conglomerates in Ajjers series (i.e. Cambro-Ordovician) on the Arak-Foum Belrem Arch (Beuf et al., 1968b).

[Figure]

FIG. 301. — Réductions de série et variations sédimentaires dans la formation des Ajjers
(partie nord-orientale du môle de Foum Belrem).

(c) Syn-tectonic conglomerates in Ajjers series (i.e. Cambro-Ordovician) on the Arak-Foum Belrem Arch (see fig. 301, p. 366) (Beuf et al., 1971).

[Figure]

FIG. 331. — Influences de la tectonique synsédimentaire sur le dépôt de la formation des Ajjers (môle d'Amguid et bordure du Tanezrouft).

(d) Influence of syn-sedimentary tectonic in Ajjers series on the Azzel Matti Arch and Amguid El Biod Arch (Beuf et al., 1971).

[Figure]

(e) NNW-SSE cross section on the Azzel Matti Arch (Ahnet Basin) showing variation of thickness, wedges strata in the Cambro-Ordovician series (Fig. 15) (Eschard et al., 2010).

[Figure]

(f) Thickness and facies variations of Siluro-Devonian formations on the Idjerane axis (i.e. Arak-Foum belrem arch) (Legrand, 1967a).

[Figure]

PI. 4. Schéma hypothétique des variations de faciès dans la "Zone de Passage" et le Dévonien inférieur gréseux depuis les Chaines d'Ougarta jusqu'au Tassili externe occidental (Région de Ouallene).

(g) Thickness and facies variations of Siluro-Devonian formations in the Bled El Mass area (i.e. Azzel Matti arch) (Legrand, 1967a).

[Figure]

(h) Example of local influence of tectonic in Lower Devonian series (Beuf et al., 1971).

Line 146: Outcrop sections O1- O12 cannot be detected in Figs 9 and 10. Are they personal field data? Position of well logs W1-W21 can only very roughly be located from Fig. 3A. Given the importance of these data (which apparently have never been published previously) it is absolutely necessary to indicate individual coordinates (best as an appendix) for both.
-"We have integrated supplementary data to the paper (cf. fig. 4, 11 and 13), even if there weren't indispensable for the comprehension of the paper (showing some redundancy). They

allow a better understanding. This has changed the order of the figures. O1-O12 are presented in fig. 11, they are based and modified from Wendt et al., (2006), (2009)."

Line 152: add: major "depositional" unconformities, in order to avoid confusion with angular unconformities.
-"We have modified (line 184)."

Lines 153-154: The top Pragian unconformity is diachronous (comprises also the lowermost Emsian in the Reggane Basin and on the Azel Matti ridge). Top Givetian and top mid-Frasnian are no unconformities over the entire study area. Top Quaternary is an unconformity worldwide, therefore omit. Or do you mean base Quaternary? But this would be trivial. In this list you have omitted the most important depositional unconformity, the transgression of the lower Eifelian (costatus-Zone).
-"The calibration of the seismic lines to wells is not really precise (due to differential resolution between the two) that's why we added "Near top" for each horizons. Due to faulting or calibration issues, choose seismic horizons that are extendable is often not easy. Some of the horizon are not unconformities sensu stricto but just well-identifiable and extendable.

-The horizon named near top Pragian doesn't represent the top Pragian but a mid lower Devonian reflector horizon that is easily extendable (i.e. near top Pragian horizon).

-Top Quaternary was corrected to near base Quaternary.
-The transgression of the Lower Eifelian was difficulty extendable to the entire area due to lack of well-calibration and faulting between hanging-wall/footwall (furthermore, the seismic reflectors aren't "bright"), so was dismissed."

Line 156: geological map is 1: 200.000, not 1:20.000.
-"We have modified (line 189)."

Line 171: circular of oval shape of basins. This is pure imagination. Basins and ridges are capped by erosion in the south and by overlying Jurassic or Cretaceous in the north. Thus the second dimension of the paleogeographic units is unknown.
-"The observations of Fig. 1 resulting from Pre-Mesozoic subcrop geological maps of the Saharan Platform (Boote et al., 1998; Galeazzi et al., 2010) show this circular and oval shaped feature. They are bordered by the different arches (cf. Fig. 1 from Eschard et al., 2010 below). For example, it is well-represented by the Reggane or the Mouydir basins (cf. below Plate 1 from Galeazzi et al., 2010)."

[Figure]

**Plate 1.** (A) Enhanced satellite image of the Maghreb region (NW Africa), showing the main tectonic domains of the area and the location of the study area (NW Africa). The Berkine–Ghadames and Illizi basins are Palaeozoic intracratonic depressions developed within the Saharan Platform. The Pan-African Tilemsi Suture separates the Eglab and Hoggar massifs. N–S oriented basement faults formed during the Pan-African Orogeny and strongly influenced the structural grain of the Berkine–Ghadames and Illizi basins. The Amguid fault and its continuation into the Amguid–El Biod fault trend bounds both basins to the west. (B) Pre-Mesozoic subcrop map of the Saharan Platform, showing the main Late Palaeozoic (mostly 'Hercynian')–Early Mesozoic tectonic elements.

[Figure]

**Fig. 1.** Structural map showing the distribution of the main highs and basins across the Saharan Platform (partially redrawn from Boote *et al.* 1998). The hatched areas correspond to the main highs described in the text.

Line 174: major faults are all Hercynian. Eventual pre-Hercynian faults are inferred, but have never been documented in the field, thus are mere speculation.
-"Previous work from Beuf et al. monograph essentially based on field studies has documented the formation and maintain of arches-basins shape through the Paleozoic (see previously). It is documented elsewhere on the Saharan Platform."

Line 178: "long" instead of "length".
-"We have modified (line 226)."

After line 178: Generally, at this point there is a paragraph entitled "Previous work", but this is missing here.
-"Previous works are already well summarized in Wendt et al., (2006) and since there weren't major studies on the area. However, we can add a little summary if it is needed."

Line 179: this chapter should be re-written avoiding speculations, even if they would fit well into a hypothetical and inferred depositional image. Regarding eventual "synsedimentary extensional markers" see above.
-"see explanation Line 174 above."

Line 191: Hercynian folding is restricted to the Reggane, Ahnet and western Mouydir Basins, but decreases markedly towards the east (eastern Mouydir and Illizi Basins) where Paleozoic strata are completely flat-lying.
-"Indeed, the strain deformation of the Hercynian is decreasing eastwards (see fig. a below). Nevertheless, there still exist folding as far as Murzuq basin (even farther) visible in seismic (see b below) or satellite images (see c below). These structures are often difficult to observe eastwards because of sand dunes or Mesozoic series."

[Figure]

(a) Intensity of Hercynian deformation on the Saharan Platform modified from (Craig et al., 2006)

[Figure]

(b) Seismic sections through selected structures in the Murzuq Basin, southwest Libya showing evidence of abrupt thickening of Cambro-Ordovician and Late Devonian to Carboniferous sequences across steeply-dipping faults (Craig et al., 2006).

[Figure]

(c) Fold on the Tihemboka arch (right), Atchan arch (left) visible on Google Earth images (Murzuq basin).

Lines 205-207: synsedimentary horst and graben structures – see above (lines 174 and below). What is a "synsedimentary forced fold"? A slump?
-"Forced folds were defined by Stearns, (1978) as 'folds in which the final overall shape and trend are dominated by the shape of some forcing member below'. There are also referred to as extensional fault-propagation folds, form in response to the upward propagation of normal faults (i.e. developed above tips of propagating faults) (e.g. Withjack et al., 1990). Growth strata, onlaps and thickness variation upon have permitted to date the deformation (see a below). See also abundant literature on the subject (e.g. Gawthorpe and Hardy, 2002; Hardy and McClay, 1999; Kane et al., 2010; Lewis et al., 2015). The kinematic is extensional in these cases. In our study, this structural style is coherent with the accommodation of deformation by

basement block movement. The Silurian shales have the role of decoupling between Cambro-Ordovician sandstones and Devonian series."

[Figure]

Fig. 13. Synoptic model illustrating the influence of extensional forced-fold development on syn-rift stratigraphic architecture (a) Geometry of basement, Zechstein Supergroup and pre-rift strata prior to fault propagation – units are assumed to be isochronous. (b) Early Callovian to Late Oxfordian (SU1). Upward fault propagation is inhibited by the presence of the salt, resulting in forced folding (*sensu* Withjack et al., 2002) of pre-rift cover strata and thinning of SU1 sediments towards the developing fold. (c) Late Oxfordian to Volgian (SU2). Early SU2 sediments onlap the upper surface of SU1 and continue to thin towards the fault until after the fault has breached the salt layer and overlying cover strata. This results in displacement at-surface and the migration of the depocentre towards the fault. Note that stage (c) is only applicable to the central and southern parts of the Sleipner Basin.

Figure 17. A schematic model illustrating how forced folding and eustasy combine to control the stratal architecture of early synrift deposits on the flanks of extensional forced folds. Two end-member scenarios, which both depict shallow marine shoreface sandstone deposition during falling base level (forced regression) only, are illustrated: (1) fault propagation and forced fold amplification during rising base level only, fold amplification and hanging-wall deepening during rising base level results in basinward thickening of mudstone-dominated bodies. Shoreface sand bodies are deposited during times of tectonic quiescence thus are uniform in thickness, and truncate underlying mudstones and amalgamate toward the fold crest; and (2) fault growth during falling base level only, fold amplification and hanging-wall subsidence during falling base level results in basinward thickening of sandstone-dominated bodies. Mudstone-dominated units are deposited during times of tectonic quiescence thus are uniform in thickness, being truncated toward and thinning onto the fold crest. The regressive surfaces of marine erosion that bound the bases of individual sand bodies may be sharp near the fold crest and pass basinward into a correlative conformity.

(a) Left from (Kane et al., 2010), and right from (Lewis et al., 2015).

Line 247: From Google Earth images it is possible to recognize faults, but it is impossible to determine their age. Please explain why the faults figures in Figs 4 and 6 are Silurian-Devonian and Middle to Late Devonian age.

-"The age is based on stratigraphic markers identified from georeferenced geological maps (Bennacef et al., 1974; Bensalah et al., 1971) which were veneering on 3D Google Earth images (i.e. associated with a digital elevation model DEM) (see Figs. below).

The high quality of the new satellite images permits also the differentiation between shale and sandstones levels (when knowing the stratigraphic succession of the area it is a help too).

It is sure that it does not replace a field mission but it allows to have an overview on very large objects. Kmz format (i.e. Google Earth format) of geological maps can be added to supplementary data. Or these figures can be added to the paper?!

The timing of faults activity (in seismic or in satellite images) is done by identifying sedimentary structures such as divergent onlaps (growth strata), thickness variation and truncatures in the hanging wall synclines of forced folds (cf. above).

For example:

-In figure 5A, the sinuous morphologies of the faults indicate synsedimentary fault propagation. So, the age of faults is given by stratigraphic layer impacted (i.e. *oTj* here Tamadjert fm. i.e. Unit IV).

-In figure 5B, age of faults are given by their control on channelized sandstone body systems which are dated late Hirnantian (Girard et al., 2012). Besides, in figure 6A, divergent onlaps (DO1) in In Tahouite series (*oTh* i.e. Unit III) located in the hanging-wall syncline of fault F2 permit to date the (re)activation of fault during the Ordovician. Then, F2 is reactivated during the Silurian (fig. 6B i.e. DO2 in *sIm* series).

-In figure 6C, divergent onlaps (DO2) in Asedjrad series (*sdAs1, dAs2*) located in the hanging-wall syncline of fault F5 permit to date the (re)activation of fault during the Siluro-Devonian.

-In figure 6E, divergent onlaps (DO3) in Givetian to Mehden Yahia series (*d2b, d3a, d3b*) located in the hanging-wall syncline of fault F2 permit to date the (re)activation of fault during the Middle to Late Devonian.

-In figure 7, divergent onlaps (DO0 to DO3), thickness variations in hanging walls or and footwalls are evidence of faults reactivations."

[Figure]

Fig. 5A (previous Fig. 4A)

[Figure]

Fig. 5B (previous Fig. 4B)

[Figure]

Fig. 6A and B (previous Fig. 5A and B)

[Figure]

Fig. 6E and F (previous  Fig. 5E and F)

[Figure]

Fig. 6C and D (previous Fig. 5C and D)

Line 261: "Nine facies associations" cannot be detected in Figs 9 and 10. Do you mean the depositional environments? (these are 5). I also could not find the "supplementary data".
-"We have modified (line 319). We have integrated supplementary data to the paper (cf. fig. 4, 11 and 13)."

Line 291: There is no clear horizontal (gAPI) scale in Fig. 8. Thus it is impossible to check the numbers.
-"We have done a bigger lettering (see Fig. 9)."

Line 298: values range to 120, not 200 in Fig. 8D.
-"modified (line 298)."

Lines 329-330: 30-60 gAPI are low, not high.
-"modified (line 389)."

Line 346: 25-60 gAPI are low, not high.
-"modified (line 407)."

Line 366: stromatoporoids, tabulate and rugose corals are not mentioned on Tab. 1.
-"modified (see Table 1)."

Line 378: same as above.*
-"modified (line 440)."

Line 382-83: same as above.
-"modified (line 445)."

Line 395: HCS probably stands for hummocky cross stratification. If this should be the case, these structures indicate a shallow marine environment, not deep marine. The same interpretation refers to "influence of storms", i.e. shallow, not deep.
-"We have modified (line 459-463) and proposed that it corresponds to deeper than shoreface deposits. AF5a is interpreted as upper offshore (i.e. a kind of offshore transition). The occurrence of HCS and wavy-bedded structures, as well as the fossil traces indicate that this facies association recorded deposition in a marine environment between the fair-weather (MFWB) and the storm-wave base (MSWB) under the influence of storm wave's oscillatory currents (Dott and Bourgeois, 1982; Reading, 2002). No influence of waves has been recorded and the storm-induced deposits are embedded in fine grain sediment (mud dominated)."

[Figure]

**Figure 6.6** Generalized shoreline profile showing subenvironments, processes and facies.

From (Reading and Collinson, 2009) p. 160.

[Figure]

**Figure 22. Diagram showing inferred storm origin of hummocky stratification and graded sand laminae on shelves. Storm surge erodes sand at shore; hummocky stratification is deposited and preserved in stormy seas between fair-weather and storm-wave bases; graded laminae may be deposited and preserved at greater depths by simple settling from suspension and/or from turbidity currents. (Modified from Walker, 1979, Fig. 15, and Dott and Bourgeois, 1980, Fig. 3.)**

From (Dott and Bourgeois, 1982).

Line 396: The ichnofauna of AF5a does not necessarily indicate a deep marine environment, but could also be much more shallow, as indicated by the "influence of storms".
-"We agree that *Zoophycus* by itself can be interpreted as formed in shallow environments but also in deeper offshore setting (MacEachern et al., 2007; Vinn and Toom, 2015). In the literature, even if *Zoophycus* can be found in broad environmental systems, it occurs preferentially in deeper environments especially in slope area (Seilacher, 1967).

[Figure]

**Figure 7.41** Typical trace fossil assemblages within offshore storm (hummocky cross-stratified) sand layers and their bounding mud units. 1, *Chondrites*; 2, *Cochlichnus*; 3, *Cylindrichnus*; 4, *Diplocraterion*; 5, *Gyrochorte*; 6, *Muensteria*; 7, *Ophiomorpha*; 8, *Palaeophycus*; 9, *Phoebichnus*; 10, *Planolites*; 11, *Rhizocorallium*; 12, *Rosselia*; 13, *Skolithus*; 14, *Thalassinoides*; 15, *Zoophycus* (from Ekdale, Bromley & Pemberton, 1984).

From (Reading, 2002) p. 264.

Line 406: The Grès de Mehden (not "Meden) Yahia and the Temertasset (not "Termatasset) shales were deposited during a regressive phase and should be discussed in one of the preceding paragraphs.
-"We have modified and added to paragraph 5.2 the regressive trend (line 520-522). However, it is written 'Meden' in geological maps (Bennacef et al., 1974; Bensalah et al., 1971) see above. Indeed, the transition from shales to sandstones correspond to a regressive trend (as proposed Wendt et al., 2006) but this paragraph only deal with the Argiles de Mehden Yahia shales interpreted as deeper environment the pattern corresponds to a MFS."

Line 410: not "Paleozoic" but "Devonian". Fig. 7 shows almost exclusively Devonian.
-"modified (line 476)."

Line 421: The major flooding surface is not MFS5 but MFS4 (Eifelian transgression).
-"We have added this comments (line 505-507). However, the MFS5 correspond to the transition of an important changes in geodynamic context but do not significate that it is an important MFS at the scale of the Devonian record. Besides, it is easily identified and extendable horizon because of a gamma ray peak. So, we have decided to horizontalized on it. Besides, outcrops (O1 – O9) from Wendt et al., (2006) are horizontalized on top Givetian."

Line 423: same error. Moreover, you have omitted the gap in the Emsian.
-"This gap is included in the pattern from D1 to D5 and is discussed in paragraph 6 (F) (line 585-591). Upper Emsian emergence is characterized by truncatures from satellites images (see fig. 6D and 6E) and well cross section (erosion and pinch out of upper Emsian to Eifelian series) (see fig. 10, 12 and 13)."

Lines 433-436: This is highly exaggerated. The facies variations between the Ahnet Basin and the adjacent ridges are very weak.
-"modified (line 500). We have moderated our purpose."

Line 442: MFS5 is not a major flooding surface. The corresponding black shales are diachronous (earliest ones in the Givetian, latest ones in the upper Frasnian), and their occurrence depends mainly on paleogeographic factors. It is true that there is an evident gap between the Givetian and the Frasnian, but this occurs only on the ridges, not in the basins, and it is caused by non-deposition, not by transgression.

-"The apparition of hot shales is observed during early Frasnian. This layer has been chosen as correlation layer as it was observed by GR in all the study core section and due to our biostratigraphical scale. These hot shales correspond to a flooding at the basin scale."

Line 451: not a maximum flooding but regression (see above).

-"D6 to D9 encompass the whole Frasnian to Famennian sedimentary succession interrupted by several sequence boundary and recording T-R trends (as shown in fig. 8)."

Line 514: an "early Eifelian" hiatus does not exist. Or do you mean the partitus Zone which in fact has not been documented? But I did not check the other references which appear to depend on palynomorph stratigraphy which, compared to conodont stratigraphy, is much less reliable.

-"corrected."

Line 660: Which are the "Three different periods of tectonic compressional pulses"? I am aware only of one, the Hercynian.

-"The Caledonian (i.e. Siluro-Devonian), Middle to Late Devonian and Pre-Hercynian events identified both in this study and in the literature see below."

Lines 668-1266: References: The reference list occupies almost the same space as the preceding text and should be drastically reduced, at least to one half. In order to avoid the impression that the article is nothing but a general review paper. Only articles referring to the study area should be included in the reference list. Unfortunately, the latter in its present length shows many incomplete citations (missing volume, missing pages, missing dots in abbreviations, missing editor, missing town (for books), missing capitalizing, wrong spelling), such as in lines 673, 676, 681, 685, 690, 696, 699, 740, 743, 745, 755, 762, 764, 765, 777, 814, 828, 830, 844, 863, 873, 893, 900, 902, 938, 957, 963, 979, 982, 1001, 1003, 1013, 1018, 1033, 1037, 1041, 1075, 1081, 1082, 1095, 1099, 1112, 1124, 1129, 1158, 1160, 1162, 1169, 1176, 1181, 1185, 1186, 1195, 1221, 1222, 1226, 1244, 1253, 1255, 1257, 1260. This list, however, is not complete. I did not check, if every reference in the text does also appear in the reference list and vice versa. This can be done much more accurately by a simple computer program (which I do not have). On the other hand, important local works are not cited.

-"We have modified references by using ZOTERO software and limited them in the text."

**Remarks to figures:**

Fig. 1: line 1275: 1: 200.000.

-"modified (line 1411)."

Line 1281: where are the supplementary data?

Map and reference of Monod (1931-1932) are missing.

"We have integrated supplementary data to the paper (cf. fig. 4, 11 and 13). Expect error on our part, we didn't' used map from Monod."

Fig. 2: Illizi Fm. Is missing in the Illizi column.

-"It is present in the Tassili column (see Fig. 3)."

Fig. 3: give exact coordinates for wells (W1- W21) and for outcrops (O1 – O9). What are the latter? Own data or previously published ones? Why there is no cross section along the O1-O9 line?

-"Coordinates of wells (W1- W21) are confidential they are put in a banalized format. O1-O9 outcrops data came from Wendt et al., (2006). However, you didn't had access to supplementary data. They are added to the paper (see fig. 11 and 13)."

Fig. 7: larger lettering is required. (I had to use a 3x magnifier to read it). What are the tiny arrows in the left gamma-ray-column?

-"We have done a bigger lettering. The tiny arrows were giving the trend of the gamma ray but they are not indispensable for understanding. They were deleted (see Fig. 8)."

Tab. 1: Please add a column with the equivalent individual formation names. In the present form this table is rather theoretical and shows no relation to the Devonian depositional areas.

-"added (see Table 1)."

Fig. 8: Because of its tiny lettering this figure is almost unreadable. Stages and formation names should be added for each sub-figure. The accompanying sections are unreadable. I could not check the source because the equivalent reference is incomplete. In the present form this figure appears rather useless. Gamma-ray-curves often do not correspond to their interpretation in the text (see above). It would make a certain sense, if there were a comparison with equivalent well logs in each sub-figure, but it would better to omit this figure completely.

-"We have done a bigger lettering cf. fig. 9 (size minimum 7). We can divide this figure by 2 to magnify for a better visibility (cf. Fig. a and b below)."

[Figure]

(a)

[Figure]

**Sedimentary structures**

| | | |
|---|---|---|
| Planar bedding | Hummocky cross bedding | Load & escape structures |
| Wave-ripple bedding | Swaley cross bedding | Slump |
| Convolute bedding | Cross bedding | Mud & pellets/clasts/drapes |
| Flaser bedding | Wave ripple bedding | Mud cracks |
| Bi-directional bedding | Combined flow ripples | Fining upwards |
| Trough cross stratification | Planar cross low angle stratification | |
| Asymmetrical ripples | Lenticular stratification & ripples | |
| Sigmoidal cross bedding | Flat lens | |

FG Firm ground
ScS Scoured surface
Syn Syneresis crack

Fe Iron
Si Silica
Py Pyrite

**Fossils & Faunal activity**

Bivalves
Fossil debris
Bryozoans
Crinoids
Dipl Ichnofacies
Bioturbations

(b)

Fig. 9: Needs larger lettering! In Fig. 2 the Emsian is a gap (which is correct), but in Fig. 9 this stage is represented by strata, which is an obvious contradiction.
-"We have done a bigger lettering. In Fig. 10, 12 and 13 upper Emsian series are truncated. An evidence of hiatus. On this representation format we cannot show the presence of hiatus (It is not a chronostratigraphic representation)."

Fig. 10: same as Fig. 9.
-"see above."

Fig. 11: "K" is missing on A and B. (line 1486).
-"modified (line 1658-1659)."

Fig. 12: larger lettering, the smallest ones are illegible.
-"We have enlarged the lettering (size minimum 7). However, enlarged the figure cannot be done without separating in two the figure. So, we have separated in two the figure (see Fig. 15 and 16)."

**Conclusion:**
As a whole the paper is well written, rather concise and accompanied by good illustrations (apart from the above remarks). It is an example of a modern interpretation of a basin and ridge paleogeography using all available techniques. An important contribution is the representation of well data which are difficult to obtain by non-oil geologists. Nevertheless, it cannot be overlooked that as a whole the paper appears to be based almost exclusively on pre-existing data. The personal contribution to the subject is difficult to distinguish Thus, in several aspects and conclusions the interpretations of the data are not or only poorly compatible with well-established field data. Some of them are highly speculative. It should also be made clear that the depositional units (basins and ridges) are nothing else than the southern prolongation of the same (but more accentuated) ones farther north. It should also be clearly expressed that the basin-and-ridge paleotopography in the Ahnet and Mouydir is of relatively short duration (early Eifelian to early Famennian). The depositional pattern of the late Famennian and the Carboniferous Is totally different from the Devonian one. A Devonian sea-level curve would be highly desirable. Absolutely necessary are several block diagrams to show the basin-and-ridge configuration at various stages. I recommend publication of the manuscript after major revision, but I would be glad to receive the revised manuscript once more before its final acceptance.

-"We have added a method figure (fig. 4) and better specified the original work in the method part. In our paper, we argue that the basement structures (at a lithospheric scale) are alternately reactivated (i.e. uplifted basement=>forced folds) during the Paleozoic. The basin-and-ridge feature was preserved since the Cambrian until the Carboniferous (as seen before) due to tectonic pulses (syn-sedimentary) and to the inherited basement features (i.e. mega shear zones and terranes rheologies). This particular zonation of the terranes (Archean, Paleoproterozoic, Proterozoic…) has constrained the basin-and-ridge architecture. However, we agree with the fact that the arches were consecutively levelled and flooded (i.e. eustatic control) during some major transgression periods. A similar block diagrams showing the evolution of basin-and-ridge configuration is published (Eschard et al., 2010), even if there are some minor differences."

Cited References:

Abdesselam-Rouighi, F.-F.: Biostratigraphie des spores, acritarches et chitinozoaires du Dévonien moyen et supérieur du bassin d'Illizi (Algèrie), Bull. Serv. Géologique Algèrie, 14(2), 97–117, 2003.

Baidder, L., Raddi, Y., Tahiri, M. and Michard, A.: Devonian extension of the Pan-African crust north of the West African craton, and its bearing on the Variscan foreland deformation: evidence from eastern Anti-Atlas (Morocco), Geol. Soc. Lond. Spec. Publ., 297(1), 453–465, doi:10.1144/SP297.21, 2008.

Belka, Z.: Early Devonian Kess-Kess carbonate mud mounds of the eastern Anti-Atlas (Morocco), and their relation to submarine hydrothermal venting, J. Sediment. Res., 68(3), 368–377, 1998.

Bellini, E. and Massa, D.: A stratigraphic contribution to the Palaeozoic of the southern basins of Libya, Geol. Libya, 1, 3–56, 1980.

[revised manuscript text omitted]

Lewis, M. M., Jackson, C. A.-L., Gawthorpe, R. L. and Whipp, P. S.: Early synrift reservoir development on the flanks of extensional forced folds: A seismic-scale outcrop analog from the

Hadahid fault system, Suez rift, Egypt, AAPG Bull., 99(06), 985–1012, doi:10.1306/12011414036, 2015.

MacEachern, J. A., Pemberton, S. G., Gingras, M. K. and Bann, K. L.: CHAPTER 4 - The Ichnofacies Paradigm: A Fifty-Year Retrospective, in Trace Fossils, edited by W. Miller, pp. 52–77, Elsevier, Amsterdam., 2007.

Massa, D.: Paléozoïque de Libye occidentale : stratigraphie et paléogéographie, Doctoral dissertation, Université de Nice, France., 1988.

Mckerrow, W. S., Niocaill, C. M. and Dewey, J. F.: The Caledonian Orogeny redefined, J. Geol. Soc., 157(6), 1149–1154, doi:10.1144/jgs.157.6.1149, 2000.

Michard, A., Hoepffner, C., Soulaimani, A. and Baidder, L.: The Variscan Belt, Earth Sci., 2008.

Moreau, J., Ghienne, J. F. and Hurst, A.: Kilometre-scale sand injectites in the intracratonic Murzuq Basin (South-west Libya): an igneous trigger?: Sand injectites in the Murzuq Basin: an igneous trigger?, Sedimentology, 59(4), 1321–1344, doi:10.1111/j.1365-3091.2011.01308.x, 2012.

Moreau-Benoit, A., Coquel, R. and Latreche, S.: Étude palynologique du dévoniendu bassin d'illizi (Sahara oriental algérien). Approche biostratigraphique, Geobios, 26(1), 3–31, doi:10.1016/S0016-6995(93)80002-9, 1993.

Ouanaimi, H. and Lazreq, N.: The 'Rich' group of the Drâa Basin (Lower Devonian, Anti-Atlas, Morocco): an integrated sedimentary and tectonic approach, Geol. Soc. Lond. Spec. Publ., 297(1), 467–482, doi:10.1144/SP297.22, 2008.

Reading, H. G. and Collinson, J. D.: Clastic coasts, in Sedimentary environments: processes, facies, and stratigraphy, pp. 154–231, John Wiley & Sons, Oxford ; Cambridge, Mass., 2009.

Seilacher, A.: Bathymetry of trace fossils, Mar. Geol., 5(5), 413–428, doi:10.1016/0025-3227(67)90051-5, 1967.

Sloss, L. L.: Sedimentary Cover—North American Craton: U.S., Geological Society of America., 1988.

Stampfli, G. M. and Borel, G. D.: A plate tectonic model for the Paleozoic and Mesozoic constrained by dynamic plate boundaries and restored synthetic oceanic isochrons, Earth Planet. Sci. Lett., 196(1), 17–33, 2002.

Stearns, D. W.: Faulting and forced folding in the Rocky Mountains foreland, Laramide Fold. Assoc. Basement Block Faulting West. U. S. Geol. Soc. Am. Mem., 151, 1–37, 1978.

Vinn, O. and Toom, U.: The trace fossil Zoophycos from the Silurian of Estonia, Est. J. Earth Sci., 64(4), 284, 2015.

Wendt, J.: Disintegration of the continental margin of northwestern Gondwana: Late Devonian of the eastern Anti-Atlas (Morocco), Geology, 13(11), 815–818, 1985.

Wendt, J., Belka, Z., Kaufmann, B., Kostrewa, R. and Hayer, J.: The world's most spectacular carbonate mud mounds (Middle Devonian, Algerian Sahara), J. Sediment. Res., 67(3), 424–436, doi:10.1306/D426858B-2B26-11D7-8648000102C1865D, 1997.

Wendt, J., Kaufmann, B., Belka, Z., Klug, C. and Lubeseder, S.: Sedimentary evolution of a Palaeozoic basin and ridge system: the Middle and Upper Devonian of the Ahnet and Mouydir (Algerian Sahara), Geol. Mag., 143(3), 269–299, doi:10.1017/S0016756806001737, 2006.

Wendt, J., Kaufmann, B. and Belka, Z.: Devonian stratigraphy and depositional environments in the southern Illizi Basin (Algerian Sahara), J. Afr. Earth Sci., 54(3–4), 85–96, doi:10.1016/j.jafrearsci.2009.03.006, 2009.

Withjack, M. O., Olson, J. and Peterson, E.: Experimental models of extensional forced folds, AAPG Bull., 74(7), 1038–1054, 1990.

Ziegler, P. A., Cloetingh, S. and van Wees, J.-D.: Dynamics of intra-plate compressional deformation: the Alpine foreland and other examples, Tectonophysics, 252(1), 7–59, doi:10.1016/0040-1951(95)00102-6, 1995.

---

## Author Comment (AC7) · 26 Sep 2018

Dear M. Lottaroli,

Thank you for your work and patience, we have tried to respond to all your comments. Your remarks have contributed to improve the paper. Please find our answers marked in red to your comments. The description of necessary changes in the revised manuscript are attached to this reply.

Sincerely,

Paul Perron, Michel Guiraud, Emmanuelle Vennin, Eric Portier, Isabelle Moretti, Moussa Konaté

Color legend in attached manuscript:

 + Green = moved sentences

 = deleted sentences

Red = added sentences

Beside the technical and scientific value of the paper I have collected some observations on the structure of the article. How data are presented and how much is clear which is the original work performed with respect of what has been re-digested from previous published work. On this respect there is to me some room of improvement. The reader needs a bit of help in being directed towards the key messages. As it is structured now I found it a little difficult, too much relevant observations in "brackets", too frequently reader is requested to look at more than one figures with reference to the same concept, jumping ahead and back-word. I would suggest some degree of simplification. At the end, even if is important to mark how big has been the effort of to give spatial relevance to info already published (e.g geochronology, etc: : :) the important thing is to convey the new and original message of the work. I Have never seen in my life Figures Captions as these. Captions represents basically another article. The relevance of illustrations must be stated in the article text. The Figure needs clear legends, but captions, if possible must be concise.

**Abstract Row** 17 – 40: It is lacking reference to your original work and the results of it. What make this paper one of original scientific content? Which is the new aspects of your approach with respect to what (a lot) have been already published on North Africa Paleozoic? Shorten the introductive remarks on Paleozoic Basins and expand the above.
-"We have reformulated the abstract axing on our work approach and results line 17-52)."

**Introduction** R47_ No need to cite specific Figure of Heine for a general remark like this, work citation is enough
-"We have modified (line 59)."

R48_ Non-conventional exploration has revived interest: : :: : :explain why and where
-"It is not the purpose of the paper."

R50,51,60,61_ incorporate in text all the (i.e. in brackets), too much. : : :.. Example: Share several common features, being generally from circular to : : :., being filled by continental to shallow marine: : :: : :: : :: : :etc: : :.etc
-"We have modified (line 61-65)."

R64_Depositional environment: : :.This is just the same you just stated in Row 51. Try to be more short in introductive remarks, these concepts are enough when explained once, you do it too frequently through the text. R82-86 In abstract too

-"We highlight the fact that this common feature of worldwide intracratonic basin is present in our case."

Geol Setting: : :.. R99_ sandwiching ! R105_Individualizing?

-"The inherited structural framework from ancient orogeny (i.e. Pan-African) will be reactivated during the Paleozoic individualizing the Saharan platform in arches and basins configurations."

**Data & meth** R122 – 129. The integrated work you describe is a standard one. You should underline more which are the new data you are presenting in this paper for the first time. Making use of GIS or looking at log and seismic doesn't represent itself something that is worth of note.

-"We have better specified the new data (especially satellite images, seismic lines and well-logs) which have been used in this study. We have integrated a figure presenting the method (cf. fig. 4 and line 146-147 in manuscript)."

R138 – 157. Describing data you analyze and methods you make reference to one table and Figs 3, 4, 5,6, 9,10. The reader would be supposed to jump ahead and backward to text to try and figure where these data are: : :: : :It would be possible to limitate this to wells, Outcrops, seismic, geol profiles and eventually have all of these on one or two Figures only?

-"We have limited citations. Regroup figures would be difficult because of the size."

R147_ supplementary data? What are they and where I found it in the paper?

-"We have integrated them to the paper for a better understanding (see Fig. 4, 11 and 13). They were put in supplementary because of their redundancy."

Structural framework 170_ Same is stated in the introduction: : :no need to repeat

-"We have corrected and modified (line 217-228)."

**4.1 & 4.2 Syn-sedimentary** Same comment for these two and is relevant to Figures citation. e.g. R187, R204, 231,242 To figure out one simple geological concept that comes from an original observation made by you the reader is supposed to jump omong 3 or more Figures, looking for, with a lens because is almost always un-readable, a fault F2: : :: : :a label DO1,DO2: : :..a part of Fig 5 that is 5AA' (R189) or 5A-A' (182). The reader is lost with all these citations. In fig 5 you have one map and one section, which is the need to label it differently if they are univocally designed by a 5A? Try if possible to simplify these two paragraphs because they are currently impossible to follow. Minimize the reference to the figures that display a concept, you do not need to cite all, just the more significative.

-"We have simplified the numeration (i.e. avoiding A-A') and captions of figures. We have also resized lettering. We also have limited citations for a same concept."

**5.1 Facies association**: : :: : : R268. Explain how the present study add knowledge to what stated above in definining the facies associations. Which are the new data? Which is the news with respect to the works cited?

-"We have better specified the new data (especially satellite images, seismic lines and well-logs) which have been used in this study. We have integrated a figure presenting the method (cf. fig. 4 in manuscript). Well correlation and stratigraphy sequence interpretation need

depositional environment electrofacies analysis. However, they come from previous work not always published. So, in order to go further, they were compiled and synthetized."

R454 6. An association: : :..refer the title to an observation not to a conclusion: : :.eg: subsidence and tectonic history
-"We have added (line 525)."

R532. Same as above
-"We have modified (line 620)."

R546: : :list of thermos orogenic events is complex, brackets inside brackets, cite name or age
-"We have modified (line 209-215). The age of different events are in the legend of fig. 1."

**Figures**
Fig.1 . Too full. Very difficult to read. I would simplify the Paleozoic series legend it is impossible to identify on Map different grades of colours within Cambrian or Carboniferous.
-"The legend is based on published geological maps. So, it is difficult to simply (loose of resolution). However, we have simplified the legend of the figure."

Too much writings in the AOI (fig3A) too small figure and too dense posting of the Geochronology data to appreciate their relationships with terrains
-"We have corrected and added (see fig. 2 previous fig. 3)."

Fig.3. W=well and O=Outcrops not in the legend.
-"We have added (line 1435-1436)."

Reference to location of sections Fig 5 & 6 difficult to read in Map.
-"We have increased the thickness of cadres and lines localizing the figures for a better visibility (cf. fig. 1 and 2)."

Capital letters of arches and basins confusing with letters that make reference to following figures.
-"Typography was homogenized (arches vs basins) between different figures."

Fig.4. I do not understand which is the extent of this area with respect to the previous Map (Fig.3A). The small writings (eg: Otj: : :: : :) are completely unreadable, remove it or enlarge. Fig.5. Same comment made above for the small writings in Figures. I do not understand the need to differentiate map and section with A, A'
-"We have simplified the numeration (i.e. avoiding A-A') and captions of figures. Captions are much concise. We have also resized lettering (Otj…) (see Fig. 5 and 6)."

Fig.7. symbols on core description section are too small to be understand.
-"We have enlarged the lettering. However, enlarged the figure cannot be done without separating in multiple part the figure (see Fig. 8)."

Fig11. K cited in legend but not in Figure
-"modified (line 1658-1659)."

Fig.12. A,B & C are too small, it should be enlarged. D &E are necessary?

-"We have enlarged the lettering and separated the figure in two (cf. Figs. 15 and 16 in revised manuscript). The D & E show the structural pattern of shear zones (i.e. SC sigmoidal geometries) which have constrained the tectonic framework of the Saharan platform. It is a typical structural style inherited in the area."